# DOES WEAK-TO-STRONG GENERALIZATION HAPPEN UNDER SPURIOUS CORRELATIONS?

**Chenruo Liu**,* **Yijun Dong**,* **Qi Lei**
New York University
New York, NY, USA
{cl7758,yd1319,ql518}@nyu.edu

## ABSTRACT

We initiate a unified theoretical and algorithmic study of a key problem in weak-to-strong (W2S) generalization: when fine-tuning a strong pre-trained student with pseudolabels from a weaker teacher on a downstream task with spurious correlations, does W2S happen, and how to improve it upon failures? We consider two sources of spurious correlations caused by group imbalance: (i) a weak teacher fine-tuned on group-imbalanced labeled data with a minority group of fraction $\eta_\ell$, and (ii) a group-imbalanced unlabeled set pseudolabeled by the teacher with a minority group of fraction $\eta_u$. Theoretically, a precise characterization of W2S gain at the proportional asymptotic limit shows that W2S always happens with sufficient pseudolabels when $\eta_u = \eta_\ell$ but may fail when $\eta_u \neq \eta_\ell$, where W2S gain diminishes as $(\eta_u - \eta_\ell)^2$ increases. Our theory is corroborated by extensive experiments on various spurious correlation benchmarks and teacher-student pairs. To boost W2S performance upon failures, we further propose a simple, effective algorithmic remedy that retrains the strong student on its high-confidence data subset after W2S fine-tuning. Our algorithm is group-label-free and achieves consistent, substantial improvements over vanilla W2S fine-tuning.

## 1 INTRODUCTION

Traditional learning paradigms like supervised learning and knowledge distillation (Hinton et al., 2015) learn from training data generated by strong teachers, *e.g.*, human experts. In contrast, contemporary foundation models encode encyclopedic knowledge through astronomical-scale pre-training, thereby achieving comparable or even superior performance to human experts in various domains via light post-training adaptation like fine-tuning (Brown et al., 2020; Achiam et al., 2023). This motivates the question on *superalignment* (Leike & Sutskever, 2023): can models with superhuman intelligence learn from weaker human supervision? *Weak-to-strong (W2S) generalization* (Burns et al., 2024) provides an encouraging answer for this question: a strong pre-trained student fine-tuned with pseudolabels generated by a weaker teacher can often outperform its teacher.

Since the first introduction of W2S by Burns et al. (2024), its mechanism has been extensively studied empirically (Guo et al., 2024; Liu & Alahi, 2024; Guo & Yang, 2024; Yang et al., 2024; 2025; Goel et al., 2025), and theoretically (Lang et al., 2024; Charikar et al., 2024; Wu & Sahai, 2025; Ildiz et al., 2025; Mulgund & Pabbaraju, 2025; Dong et al., 2025; Medvedev et al., 2025). While existing works on W2S generally assume access to clean downstream data, in practice, both the weak teacher and the unlabeled data for weak supervision often carry systematic biases, such as spurious correlations tied to demographic or acquisition factors (Arjovsky et al., 2019; Sagawa et al., 2020).

This challenge is especially relevant in the very settings that motivate W2S: a student broadly pre-trained on general data is fine-tuned for a specialized task where labeled samples are scarce and imperfect. In medicine, labels may be biased toward certain patient groups (Gupta et al., 2016) or imaging devices (Zech et al., 2018); in law, datasets may overrepresent particular jurisdictions or case types (Chalkidis et al., 2022); in autonomous driving, sensor data may be skewed toward specific weather or traffic conditions (Liu et al., 2024). For these specialized downstream tasks, one usually cannot interfere with the data acquisition process, nor obtain additional balanced data. It is therefore

---

*Equal contribution.

crucial to understand *whether W2S can remain effective under spurious correlations* caused by group imbalance—*when it succeeds, when it fails*, and *how its procedure can be improved.*

**Our contributions.** We initiate a systematic study of W2S under spurious correlations, providing (i) a theoretical analysis that answers the "when" question comprehensively by characterizing the impact of spurious correlations on W2S precisely in the proportional asymptotic limit, as well as (ii) a simple, effective remedy for the failure of W2S under spurious correlations inspired by our theory, toward answering the "how" question. Concretely, our contributions are as follows.

- **A theory of W2S under spurious correlations.** In Section 2, we conduct a systematic analysis in the ridgeless regression setting with zero approximation error, where W2S happens due to different estimation errors (*i.e.*, efficiency in utilizing data). At the proportional asymptotic limit, we provide *precise characterizations for the generalization errors of both teacher and student*. Consider using a weak teacher fine-tuned on labeled samples with a minority fraction $\eta_\ell$ to pseudolabel $N$ unlabeled samples with a minority fraction $\eta_u$ for W2S fine-tuning. We show that (i) *W2S always happens with sufficiently large $N$ when $\eta_\ell = \eta_u$* and *improves when the teacher and student have distinct representations*; whereas (ii) *when $\eta_\ell \neq \eta_u$, W2S can fail even with $N \to \infty$*, and *W2S gain tends to diminish as $(\eta_u - \eta_\ell)^2$ increases*. Our theory is validated with extensive experiments on synthetic regression problems and real classification tasks (Section 3).

- **An algorithmic enhancement for W2S when $\eta_\ell \neq \eta_u$.** In Section 4, we propose a simple, effective algorithm that retrains the strong student on its high-confidence data subset after W2S fine-tuning via the generalized cross-entropy loss (Zhang & Sabuncu, 2018). Our method requires no group annotations and improves W2S when the gap between $\eta_u$ and $\eta_\ell$ is large. We conduct extensive experiments on assorted spurious correlation benchmarks (e.g., Waterbirds (Sagawa et al., 2020), BFFHQ (Lee et al., 2021), and ImageNet-9 (Xiao et al., 2020)), across 10 different teacher–student model pairs. The results show that our algorithm achieves consistent and substantial gains over vanilla W2S.

## 1.1 RELATED WORKS

**W2S generalization.** Empirically, many methods have been developed to validate/enhance W2S across various vision and natural language modeling tasks, including adjustable loss functions (Guo et al., 2024), multi-teacher algorithms (Liu & Alahi, 2024), data refinement strategies (Guo & Yang, 2024; Yang et al., 2024), and the use of weak models for data filtering (Li et al., 2024). Notably, Jeon et al. (2025) specifically focuses on improving W2S performance under distribution shifts. Theoretical work on W2S is also rapidly expanding, offering various mechanistic explanations from first principles, including the perspectives of neighborhood expansion (Lang et al., 2024), data overlap density (Shin et al., 2025), transfer learning (Somerstep et al., 2024), teacher-student disagreement (Charikar et al., 2024; Mulgund & Pabbaraju, 2025; Yao et al., 2025; Xu et al., 2025), benign overfitting (Wu & Sahai, 2025; Xue et al., 2025), knowledge distillation (Ildiz et al., 2025), low intrinsic dimension of fine-tuning (Aghajanyan et al., 2021; Dong et al., 2025), regularization (Medvedev et al., 2025; Moniri & Hassani, 2025), and feature learning with different inductive biases (Oh et al., 2025). However, theoretical understanding of W2S under distribution shift, especially in the presence of spurious correlations, remains limited.

**Group robustness in knowledge distillation.** When transferring knowledge from a strong teacher to a weaker student, knowledge distillation (Hinton et al., 2015) has been shown to harm the minority group performance (Lukasik et al., 2021; Vilouras et al., 2023; Wang et al., 2023; Lee & Lee, 2023; Kenfack et al., 2024). To address this issue, different methods have been proposed, including adaptive mixing weights and per-class margins (Lukasik et al., 2021), distributionally robust optimization (Wang et al., 2023; Vilouras et al., 2023), last-layer transplantation (Lee & Lee, 2023), and gradient-based reweighting (Kenfack et al., 2024) [1]. Our work differs from these approaches in three key aspects: (a) W2S generalization, where a weak teacher supervises a stronger student, is fundamentally distinct from classical knowledge distillation, (b) we explicitly consider the impact of mismatched minority group proportions between teacher and student, and (c) our method for improving W2S performance does not require any auxiliary information such as group annotations.

---

[1] Group robustness under spurious correlations in supervised learning has been extensively studied and is out of scope of this work. We defer more discussions to Appendix B.

## 2 A THEORY OF W2S UNDER SPURIOUS CORRELATION

**Notations.** For any $p, q \in \mathbb{N}$, $p \geq q$, let $\mathrm{Stiefel}(p, q) = \{\mathbf{Q} \in \mathbb{R}^{p \times q} \mid \mathbf{Q}^\top \mathbf{Q} = \mathbf{I}_q\}$ be the Stiefel manifold. $\mathbf{A} \otimes \mathbf{B} \in \mathbb{R}^{mp \times nq}$ denotes the Kronecker product of $\mathbf{A} \in \mathbb{R}^{m \times n}$ and $\mathbf{B} \in \mathbb{R}^{p \times q}$; when $n = q$, let $[\mathbf{A}; \mathbf{B}] \in \mathbb{R}^{(m+p) \times n}$ be the vertical stack; when $m = p$, let $[\mathbf{A}, \mathbf{B}] \in \mathbb{R}^{m \times (n+q)}$ be the horizontal stack. For any $\mathbf{w} \in \mathbb{R}^d$ and $i \in [d]$ or $\mathcal{I} \subseteq [d]$, let $w_i$ and $[\mathbf{w}]_{\mathcal{I}}$ denote the $i$-th entry and the subvector of $\mathbf{w}$ indexed by $\mathcal{I}$. For any $\mathbf{A} \in \mathbb{R}^{m \times n}$ and $i \in [m], j \in [n]$, let $A_{i,j}$ denote the $(i, j)$-th entry; $[\mathbf{A}]_{i,:} \in \mathbb{R}^n$ denotes the $i$-th row; $[\mathbf{A}]_{:,j} \in \mathbb{R}^m$ denotes the $j$-th column; and index subsets $\mathcal{I} \subseteq [m], \mathcal{J} \subseteq [n]$ pick the corresponding submatrices.

### 2.1 PROBLEM SETUP: REGRESSION UNDER SPURIOUS CORRELATION

**Downstream task.** Consider a downstream regression task characterized by a distribution $\mathcal{D}(\eta) : \mathcal{X} \times \mathcal{Y} \times \mathcal{G} \to [0, 1]$ where $\mathcal{X}$ is the input space, $\mathcal{Y} = \mathbb{R}$ is the label space, and $\mathcal{G} = \{0, 1\}$ contains group labels (*i.e.*, 1 for minority and 0 for majority). The fraction of the minority group in the population is controlled by $\eta \in [0, \frac{1}{2}]$ such that $\Pr[g = 1] = 1 - \Pr[g = 0] = \eta$.

**Definition 1** (Regression under spurious correlations). *Let $\mathcal{D}_{\mathbf{x}}$ be the marginal distribution of $\mathbf{x} \in \mathcal{X}$; $\mathcal{D}_{\mathbf{x}|g}$ be the conditional distribution of $\mathbf{x}$ given $g$; and $\mathcal{D}_{y|\mathbf{x}}$ be the conditional distribution of $y$ given $\mathbf{x}$ satisfying $y = f_*(\mathbf{x}) + \epsilon$ for unknown $f_* : \mathcal{X} \to \mathbb{R}$ and i.i.d. label noise $\epsilon \sim \mathcal{N}(0, \sigma_y^2)$ independent of $\mathbf{x}$. Consider two feature maps: (i) the core feature $\mathbf{z} : \mathcal{X} \to \mathbb{R}^{d_z}$ determines the label $y$ through $\mathbf{z}(\mathbf{x}) \sim \mathcal{N}(\mathbf{0}_{d_z}, \mathbf{I}_{d_z})$ and $f_*(\mathbf{x}) = \mathbf{z}(\mathbf{x})^\top \boldsymbol{\beta}_*$ for fixed $\boldsymbol{\beta}_* \in \mathbb{R}^{d_z}$; while (ii) the group feature $\boldsymbol{\xi} : \mathcal{X} \to \mathbb{R}^p$ ($2 < p < \infty$) determines the group label $g$ through $\boldsymbol{\xi}(\mathbf{x}) \sim \mathcal{N}(g\boldsymbol{\mu}_\xi, \sigma_\xi^2 \mathbf{I}_p)$ for fixed $\boldsymbol{\mu}_\xi \in \mathbb{R}^p$ with dimension-independent $\|\boldsymbol{\mu}_\xi\|_2, \sigma_\xi^2 \asymp 1$.*

Here, $\mathbf{z}(\mathbf{x})$ encodes the core information for predicting $y$ that is invariant across groups, typically rich in semantics and therefore hard to learn (high-dimensional); while $\boldsymbol{\xi}(\mathbf{x})$ is a latent feature controlling which group $\mathbf{x}$ belongs to, typically simpler to represent and therefore low-dimensional.

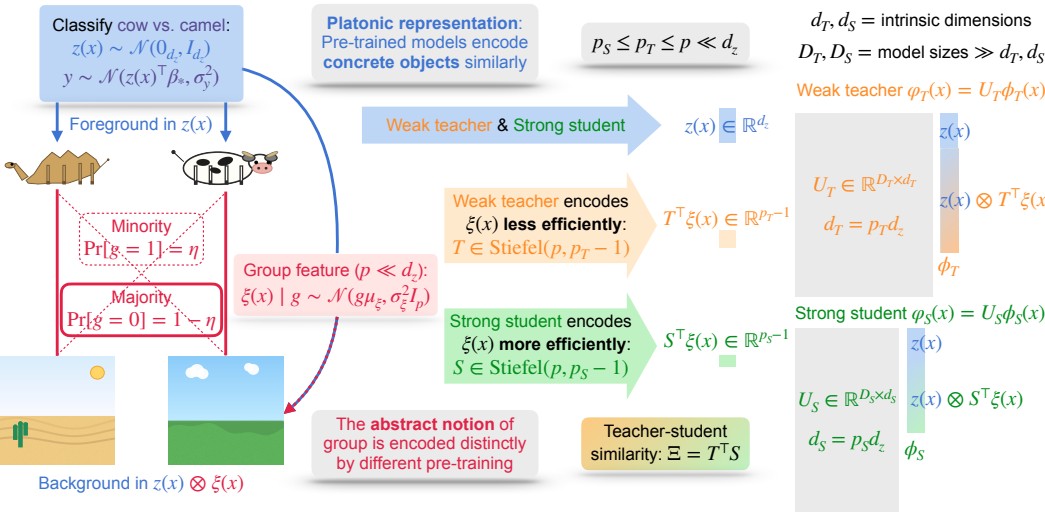

Figure 1: Visualization of the theoretical setup in Definitions 1 and 2 through Example 1.

**Weak vs. strong models.** We consider two pre-trained models that provide reasonably high-quality features for the downstream task: a weak teacher model $f_T : \mathcal{X} \to \mathbb{R}$ and a strong student model $f_S : \mathcal{X} \to \mathbb{R}$. Adapting the setting in Dong et al. (2025), we model fine-tuning in the kernel regime (Jacot et al., 2018; Malladi et al., 2023) with low intrinsic dimensions (Aghajanyan et al., 2021). In particular, we consider learning overparametrized linear layers $\boldsymbol{\theta}_T \in \mathbb{R}^{D_T}$ and $\boldsymbol{\theta}_S \in \mathbb{R}^{D_S}$ over high-dimensional pre-trained representations $\varphi_T : \mathcal{X} \to \mathbb{R}^{D_T}$ and $\varphi_S : \mathcal{X} \to \mathbb{R}^{D_S}$, respectively. When fine-tuning lies in the kernel regime, $\varphi_T, \varphi_S$ correspond to the gradients of the tunable parameters in $f_T, f_S$ at the pre-trained initialization, respectively, where $D_T, D_S$ stand for the large tunable parameter counts. The difference between $\varphi_T, \varphi_S$ that separates the weak and strong models on the downstream task with spurious correlations is pivotal in this setting:

**Definition 2** (Weak vs. strong models). *(i) The weak teacher representation $\varphi_T$ heavily entangles the core and group features: there exists $\mathbf{U}_T \in \mathrm{Stiefel}(D_T, d_T)$ ($d_T \ll D_T$) such that $\varphi_T(\mathbf{x}) =$*

$\mathbf{U}_T \phi_T(\mathbf{x})$ *and* $\phi_T(\mathbf{x}) = \mathbf{z}(\mathbf{x}) \otimes \mathbf{w}(\mathbf{x})$, *where* $\mathbf{w}(\mathbf{x}) = [1; \mathbf{T}^\top \boldsymbol{\xi}(\mathbf{x})] \in \mathbb{R}^{p_T}$ $(2 \leqslant p_T \leqslant p)$ *for a fixed* $\mathbf{T} \in \mathrm{Stiefel}(p, p_T - 1)$ *that projects* $\boldsymbol{\xi}(\mathbf{x})$ *to a lower dimension (i.e.,* $\phi_T(\mathbf{x}) = [\mathbf{z}(\mathbf{x}); \mathbf{z}(\mathbf{x}) \otimes (\mathbf{T}^\top \boldsymbol{\xi}(\mathbf{x}))] \in \mathbb{R}^{d_T}$). *We note that* $d_T = p_T d_z$. *Let* $\boldsymbol{\mu}_T = \mathbf{T}^\top \boldsymbol{\mu}_\xi \in \mathbb{R}^{p_T - 1}$.

*(ii) A strong student representation* $\varphi_S$ *partially disentangles the core and group features: there exists* $\mathbf{U}_S \in \mathrm{Stiefel}(D_S, d_S)$ $(d_S \ll D_S)$ *such that* $\varphi_S(\mathbf{x}) = \mathbf{U}_S \phi_S(\mathbf{x})$ *and* $\phi_S(\mathbf{x}) = \mathbf{z}(\mathbf{x}) \otimes \boldsymbol{\psi}(\mathbf{x})$, *where* $\boldsymbol{\psi}(\mathbf{x}) = [1; \mathbf{S}^\top \boldsymbol{\xi}(\mathbf{x})] \in \mathbb{R}^{p_S}$ $(2 \leqslant p_S \leqslant p_T)$ *for a fixed* $\mathbf{S} \in \mathrm{Stiefel}(p, p_S - 1)$ *that projects* $\boldsymbol{\xi}(\mathbf{x})$ *to a much lower dimension,* $p_S \ll p$ *(i.e.,* $\phi_S(\mathbf{x}) = [\mathbf{z}(\mathbf{x}); \mathbf{z}(\mathbf{x}) \otimes (\mathbf{S}^\top \boldsymbol{\xi}(\mathbf{x}))] \in \mathbb{R}^{d_S}$). *We note that* $d_S = p_S d_z$. *Let* $\boldsymbol{\mu}_S = \mathbf{S}^\top \boldsymbol{\mu}_\xi \in \mathbb{R}^{p_S - 1}$.[2]

Definition 2 formalizes the intuitions that compared to $\varphi_T$, the stronger $\varphi_S$ (i) represents the information required for the downstream task more efficiently ($d_S \leqslant d_T$) and (ii) partially disentangles the core and group features, bringing robustness to spurious correlations. Notice that with $\mathbf{z}(\mathbf{x})$ prepending in both $\varphi_T(\mathbf{x})$ and $\varphi_S(\mathbf{x})$, the teacher and student both have zero approximation error (*i.e.*, both pre-trained models are expressive enough for the downstream task), and W2S happens due to different estimation errors (*i.e.*, the student is more sample efficient than its teacher).

**Example 1.** *We take the well-known analogy of classifying cows (often in pastures) vs. camels (often in deserts) (Arjovsky et al., 2019) as an example (see Figure 1). With* $\mathbf{z}(\mathbf{x})$ *encoding the foreground of cows/camels,* $\boldsymbol{\xi}(\mathbf{x})$ *represents whether the background is typical or not, while* $\mathbf{z}(\mathbf{x}) \otimes \mathbf{w}(\mathbf{x})$ *and* $\mathbf{z}(\mathbf{x}) \otimes \boldsymbol{\psi}(\mathbf{x})$ *correspond to the representations of background from the weak and strong models.*

*While the Platonic representation hypothesis (Huh et al., 2024) suggests that different pre-trained models tend to represent similar concrete objects similarly (with the same* $\mathbf{z}(\mathbf{x})$*), different model capacities can lead to distinct representations of a "typical" group in* $\boldsymbol{\xi}(\mathbf{x})$*. For instance, a strong model that has learned the natural habitat of cows/camels during pre-training can encode typical samples as those with their respective backgrounds, leading to a simple, low-dimensional* $\boldsymbol{\psi}(\mathbf{x})$*; whereas a weaker model without such knowledge have to rely on more complicated mechanisms to represent typical samples (e.g., counting), resulting in a more complex, higher-dimensional* $\mathbf{w}(\mathbf{x})$*.*

Analogous to Dong et al. (2025), a key quantity that controls W2S gain is the similarity between the weak teacher and strong student representations, $\varphi_T$ and $\varphi_S$, as formalized in Definition 3.

**Definition 3** (Teacher-student similarity). *Under Definition 2, we define a similarity matrix* $\boldsymbol{\Xi} = \mathbf{T}^\top \mathbf{S} \in \mathbb{R}^{(p_T - 1) \times (p_S - 1)}$. *Notice that* $\|\boldsymbol{\Xi}\|_F^2 \leqslant p_S - 1$ *and* $\|\boldsymbol{\Xi}\|_2 \leqslant 1$.

$\boldsymbol{\Xi}$ measures the similarity of group features extracted by $\varphi_T, \varphi_S$, *e.g.*, $\|\boldsymbol{\Xi}\|_F^2 \to 0$ means $\mathbf{w}(\mathbf{x})$ and $\boldsymbol{\psi}(\mathbf{x})$ are orthogonal, while $\|\boldsymbol{\Xi}\|_F^2 \to p_S - 1$ means $\mathbf{w}(\mathbf{x})$ and $\boldsymbol{\psi}(\mathbf{x})$ are highly aligned.

**W2S fine-tuning pipeline.** We consider two training sets with *i.i.d.* samples: (i) a small labeled training set $\widetilde{\mathcal{S}} = \{(\widetilde{\mathbf{x}}_i, \widetilde{y}_i) \mid i \in [n]\} \sim \mathcal{D}(\eta_\ell)^n$ that is privately available only to the weak teacher, $\varphi_T$, and (ii) a large unlabeled training set $\mathcal{S}_x = \{\mathbf{x}_i \mid i \in [N]\}$ from $\mathcal{S} = \{(\mathbf{x}_i, y_i) \mid i \in [N]\} \sim \mathcal{D}(\eta_u)^N$ with hidden labels that is privately available only to the strong student, $\varphi_S$, where $\eta_\ell, \eta_u \in [0, \frac{1}{2}]$. The W2S fine-tuning pipeline consists of two stages: (i) Supervised fine-tuning (SFT) of $f_T(\cdot) = \varphi_T(\cdot)^\top \boldsymbol{\theta}_T$ on $\widetilde{\mathcal{S}}$ via ridgeless regression: assuming $n > d_T$,

$$\boldsymbol{\theta}_T = \operatorname*{argmin}_{\boldsymbol{\theta} \in \mathbb{R}^{D_T}} \|\boldsymbol{\theta}\|_2^2 \quad s.t. \quad \boldsymbol{\theta} \in \operatorname*{argmin}_{\boldsymbol{\theta}' \in \mathbb{R}^{D_T}} \frac{1}{n} \sum_{i=1}^n (\varphi_T(\widetilde{\mathbf{x}}_i)^\top \boldsymbol{\theta}' - \widetilde{y}_i)^2, \tag{1}$$

(ii) W2S fine-tuning of $f_S(\cdot) = \varphi_S(\cdot)^\top \boldsymbol{\theta}_S$ on $\mathcal{S}_x$ labeled by $f_T$ via ridgeless regression:

$$\boldsymbol{\theta}_S = \operatorname*{argmin}_{\boldsymbol{\theta} \in \mathbb{R}^{D_S}} \|\boldsymbol{\theta}\|_2^2 \quad s.t. \quad \boldsymbol{\theta} \in \operatorname*{argmin}_{\boldsymbol{\theta}' \in \mathbb{R}^{D_S}} \frac{1}{N} \sum_{i=1}^N (\varphi_S(\mathbf{x}_i)^\top \boldsymbol{\theta}' - f_T(\mathbf{x}_i))^2, \tag{2}$$

Following Burns et al. (2024), in this W2S fine-tuning pipeline, we assume the weak teacher after SFT is fixed and not trainable, accessible in the W2S fine-tuning stage only through inference. Moreover, the labeled training set, $\widetilde{\mathcal{S}}$, is only accessible in the first, SFT stage to the weak teacher, whereas the unlabeled set $\mathcal{S}_x$ is only accessible in the second, W2S fine-tuning stage to the strong student.

---

[2]For both $\mathbf{w}(\mathbf{x})$ and $\boldsymbol{\psi}(\mathbf{x})$, the first entry 1 effectively prepends the core feature $\mathbf{z}(\mathbf{x})$ in $\varphi_T(\mathbf{x})$ and $\varphi_S(\mathbf{x})$, which is essential to ensure that both teacher and student have negligible approximation error. Intuitively, pre-trained models have sufficient expressivity to learn the downstream task over population.

**Remark 1** (Why ridgeless regression provides sufficient regularization?). *We note that under Definition 2 where both $\varphi_T(\mathbf{x})$ and $\varphi_S(\mathbf{x})$ are constrained in low-dimensional subspaces,* $\mathrm{Range}(\mathbf{U}_T)$ *and* $\mathrm{Range}(\mathbf{U}_S)$, *ridgeless regression provides nearly optimal regularization to avoid overfitting (Wu & Xu, 2020; Hastie et al., 2022), which is essential for W2S generalization (Burns et al., 2024). When $\varphi_T(\mathbf{x})$ and $\varphi_S(\mathbf{x})$ are concentrated (in contrast to contrained) in low-dimensional subspaces with tails evenly distributed in the orthogonal complement, explicit regularization (Moniri & Hassani, 2025; Dong et al., 2025) or early stopping (Burns et al., 2024; Medvedev et al., 2025) becomes necessary to prevent the student from overfitting to noisy teacher labels. Nevertheless, analogous to Dong et al. (2025), extending our ridgeless analysis to ridge regression does not alter our key insights on spurious correlations. Therefore, we focus on the ridgeless setting for clarity of exposition.*

The generalization performance is evaluated over a test distribution $\mathcal{D}(\eta_t)$ for some $\eta_t \in [0,1]$: with the test risk $\mathcal{R}_{\eta_t}(f) := \mathbb{E}_{(\mathbf{x},y)\sim\mathcal{D}_{\mathbf{x},y}(\eta_t)}[(f(\mathbf{x})-y)^2]$, we consider the excess risk

$$\mathbf{ER}_{\eta_t}(f) := \mathcal{R}_{\eta_t}(f) - \mathcal{R}_{\eta_t}(f_*) = \mathcal{R}_{\eta_t}(f) - \sigma_y^2. \tag{3}$$

In particular, $\eta_t = \frac{1}{2}$ corresponds to the average test risk; $\eta_t = 0$ corresponds to the majority test risk; and $\eta_t = 1$ corresponds to the minority test risk.

## 2.2 W2S GENERALIZATION UNDER SPURIOUS CORRELATION

With the problem setup, we are ready to present our main theorems regarding the effect of spurious correlations on W2S generalization. First, to characterize the excess risks of $f_T$ and $f_S$ (and thereby the W2S generalization gain) precisely, we push the problem to the proportional asymptotic limit:

**Assumption 1** (Proportional asymptotic limit). *We consider $d_z, n, N \to \infty$ with $d_z/n \to \gamma_z \in (0, p_T^{-1})$ (i.e., $n > d_T$), $d_z/N \to \nu_z \in (0, p_S^{-1})$ (i.e., $N > d_S$), whereas $2 \leqslant p_S \leqslant p_T \leqslant p$ are fixed.*

We highlight that in practice, the unlabeled samples are typically much more affordable than the labeled ones, leading to $\nu_z \ll \gamma_z$. Now, we characterize the excess risks of the weak teacher after SFT and the strong student after W2S fine-tuning, respectively, in Theorems 1 and 2.

**Theorem 1** (SFT of weak teacher (Appendix D.1)). *Under Assumption 1, (1) satisfies*

$$\mathbb{E}_{\mathcal{D}(\eta_\ell)^n}\left[\mathbf{ER}_{\eta_t}(f_T)\right] \xrightarrow{\mathbb{P}} \sigma_y^2\,\gamma_z\Big(\quad \underbrace{\boxed{p_T}}_{\mathcal{V}_T^{(0)}\text{ from label noise}} \quad + \quad \underbrace{\boxed{\frac{\|(\eta_t - \eta_\ell)\,\boldsymbol{\mu}_T\|_2^2}{\sigma_\xi^2}}}_{\mathcal{V}_T^{(1)}\text{ from spurious correlations}}\quad\Big).$$

**Theorem 2** (W2S, formally in Theorem 3). *Under Assumption 1, (2) satisfies*

$$\mathbb{E}_{\mathcal{D}(\eta_u)^N,\mathcal{D}(\eta_\ell)^n}\left[\mathbf{ER}_{\eta_t}(f_S)\right] \xrightarrow{\mathbb{P}} \sigma_y^2\gamma_z\Big(\underbrace{\boxed{p_{T\wedge S}}}_{\mathcal{V}_S^{(0)}\leqslant\mathcal{V}_T^{(0)}} + \underbrace{\boxed{\frac{\|(\eta_u-\eta_\ell)\boldsymbol{\mu}_T + (\eta_t-\eta_u)\boldsymbol{\Xi}\boldsymbol{\mu}_S\|_2^2}{\sigma_\xi^2}}}_{\mathcal{V}_S^{(1)}\leqslant\mathcal{V}_T^{(1)}\text{ when }\eta_u=\eta_\ell} + \underbrace{\boxed{\Theta(\nu_z)}}_{\mathcal{E}_S\ll 1}\Big),$$

*where $p_{T\wedge S} = 1 + \|\boldsymbol{\Xi}\|_F^2 \in [1, p_S]$ is the effective group feature dimension learned by the strong student from the weak teacher controlled by the similarity between $\varphi_T$ and $\varphi_S$ in encoding group features (see Definition 3)—less similar teacher-student pairs enjoy lower $p_{T\wedge S}$; $\mathcal{V}_S^{(0)}$ and $\mathcal{V}_T^{(0)}$ are generalization errors of $f_S$ and $f_T$ from noisy labels; $\mathcal{V}_S^{(1)}$ and $\mathcal{V}_T^{(1)}$ are generalization errors of $f_S$ and $f_T$ induced by spurious correlations, $\eta_u, \eta_\ell \neq \eta_t$; and the higher-order term $\mathcal{E}_S$, formalized in Theorem 3, becomes negligible when $\nu_z \ll 1$.*

It is worth noting that the proportional asymptotic limit (Assumption 1) assumed in Theorems 1 and 2 can be relaxed to incorporate finite-sample cases via standard edge fluctuation analysis (see e.g., Hastie et al. (2022); Cheng & Montanari (2024)). We omit such extensions here since they do not bring additional insights to Theorems 1 and 2.

As a special case, without spurious correlations ($\eta_\ell = \eta_u = \eta_t$ or $\boldsymbol{\mu}_\xi = \mathbf{0}_p$), Theorems 1 and 2 exactly recover the results in Dong et al. (2025) at the proportional asymptotic limit: $\mathbb{E}[\mathbf{ER}_{\eta_t}(f_T)]\to\sigma_y^2\gamma_z p_T$ and $\mathbb{E}[\mathbf{ER}_{\eta_t}(f_S)]\to\sigma_y^2\gamma_z(p_{T\wedge S} + \Theta(\nu_z))$, where with a small $\nu_z \ll 1$, the W2S gain is larger when the teacher and student representations are less aligned (*i.e.*, lower $p_{T\wedge S}$). Meanwhile, Theorems 1 and 2 together reveal insights regarding the effect of spurious correlations on the W2S gain,

$$\Delta\mathcal{R}_{\eta_t} := \mathbb{E}_{\mathcal{D}(\eta_\ell)^n}\left[\mathbf{ER}_{\eta_t}(f_T)\right] - \mathbb{E}_{\mathcal{D}(\eta_u)^N,\mathcal{D}(\eta_\ell)^n}\left[\mathbf{ER}_{\eta_t}(f_S)\right], \tag{4}$$

as discussed in Remark 2, where W2S generalization happens whenever $\Delta\mathcal{R}_{\eta_t} > 0$.

**Remark 2** (Does W2S happen under spurious correlations?). *Theorems 1 and 2 provide a mixed answer to this question conditioned on various factors, including the teacher-student similarity, the separation between groups, and the choice of $\eta_u$ for given $\eta_\ell$[3], as summarized below:*

(a) **W2S happens whenever $\eta_u = \eta_\ell$ and $\nu_z$ is small**, *e.g., when $\|\Xi\|_F^2 \approx 0$ and $\nu_z \ll 1$, $\Delta\mathcal{R}_{\eta_t} > 0$ is optimized at $\eta_u \approx \eta_\ell$ (Figure 2). We highlight that when $\eta_u = \eta_\ell$, in addition to the W2S gain from variance reduction $\mathcal{V}_T^{(0)} - \mathcal{V}_S^{(0)} = p_T - p_{T \wedge S} \geqslant 0$, **the strong student improves upon its teacher in handling spurious correlations**: $\mathcal{V}_T^{(1)} - \mathcal{V}_S^{(1)} = ((\eta_t - \eta_\ell)^2/\sigma_\xi^2)(\|\boldsymbol{\mu}_T\|_2^2 - \|\Xi\boldsymbol{\mu}_S\|_2^2) \geqslant 0$, where the gain increases as the teacher-student similarity decreases.*

(b) *For fixed $\Xi$, assume $\boldsymbol{\mu}_T \neq \Xi\boldsymbol{\mu}_S$, when $\nu_z \ll 1$, the optimal $\eta_u$ that maximizes W2S gain is $\eta_u^\star = \frac{\eta_\ell\|\boldsymbol{\mu}_T\|_2^2 - (\eta_t + \eta_\ell)\boldsymbol{\mu}_T^\top\Xi\boldsymbol{\mu}_S + \eta_t\|\Xi\boldsymbol{\mu}_S\|_2^2}{\|\boldsymbol{\mu}_T - \Xi\boldsymbol{\mu}_S\|_2^2}$, e.g., when $\eta_\ell = \frac{1}{2}$, $\eta_u^\star = \frac{1}{2}$; when $\|\Xi\boldsymbol{\mu}_S\|_2 \ll \|\boldsymbol{\mu}_T\|_2$, $\eta_u^\star \approx \eta_\ell$; with $\|\Xi\boldsymbol{\mu}_S\|_2 \neq 0$, $\eta_u^\star$ tends to increase with $\|\boldsymbol{\mu}_S\|_2^2$ and deviate from $\eta_\ell$ when $\|\boldsymbol{\mu}_S\|_2^2 \approx \|\boldsymbol{\mu}_T\|_2^2$ (Figure 3 left).*

(c) *W2S gain increases as the teacher-student similarity $\|\Xi\|_F^2$ decreases (Figure 3 right).*

(d) **W2S may not happen if $\eta_u \neq \eta_\ell$, even when $\nu_z \ll 1$ and $\|\Xi\|_F^2 = 0$**, *e.g., when $\eta_\ell = 0.4$ but $\eta_u = 0.1$, with $\|\Xi\|_F^2 = 0$, W2S does not happen if the majority and minority groups are well separated: $\Delta\mathcal{R}_{1/2} < 0$ for any $\nu_z$ if $\|\boldsymbol{\mu}_T\|_2^2/\sigma_\xi^2 > 12.5(p_T - 1)$. More generally, for $\|\Xi\|_F^2 = 0$, $\mathcal{V}_S^{(1)}$ increases proportionally to $(\eta_u - \eta_\ell)^2$, and thus $\Delta\mathcal{R}_{\eta_t}$ diminishes as the gap increases.*

In Appendix F, we further discuss implications of Theorems 1 and 2 on the *fairness* of W2S.

## 2.3 SYNTHETIC EXPERIMENTS

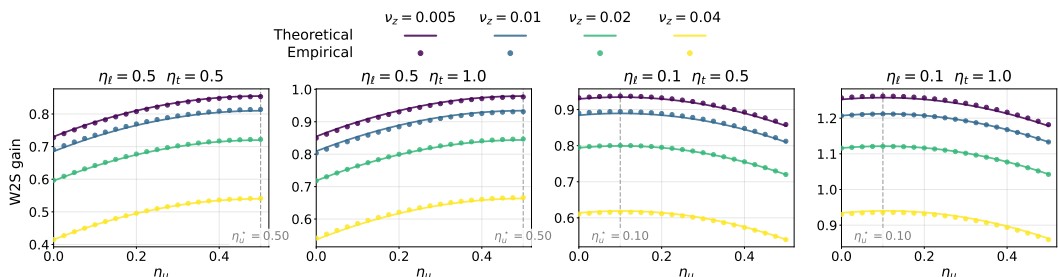

Figure 2: W2S gains across different combinations of $\eta_\ell$ and $\eta_t$. Each panel shows theoretical (solid lines) and empirical (circles) results for W2S gain as a function of $\eta_u$, across different $\nu_z$ values. Here we fix $\boldsymbol{\mu}_T$, $\boldsymbol{\mu}_S$, $\Xi$, and $d_z$ with $\|\boldsymbol{\mu}_T\|_2^2 = 10.0$, $\|\boldsymbol{\mu}_S\|_2^2 = 0.1$, $\|\Xi\|_F^2 = 0.1p_S$. Vertical dashed lines indicate the theoretical optimal $\eta_u^\star$ values that maximize W2S gain.

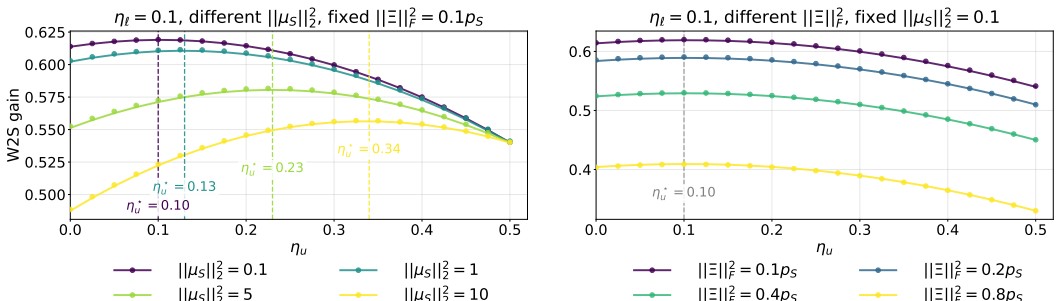

Figure 3: Impact of $\boldsymbol{\mu}_S$ and $\Xi$ on W2S gain. Both panels show theoretical (solid lines) and empirical (circles) results for W2S gain as a function of $\eta_u$. Fixed parameters: $\eta_\ell = 0.1$, $\eta_t = 0.5$, $\nu_z = 0.04$, $\|\boldsymbol{\mu}_T\|_2^2 = 10.0$. Left: varying $\|\boldsymbol{\mu}_S\|_2^2$ with fixed $\|\Xi\|_F^2 = 0.1p_S$. Right: varying $\|\Xi\|_F^2$ with fixed $\|\boldsymbol{\mu}_S\|_2^2 = 0.1$. Dashed lines indicate the theoretical optimal $\eta_u^\star$ values that maximize W2S gain.

---

[3]In practice, $\eta_\ell$ is typically fixed and known (*e.g.*, given a weak teacher fine-tuned on the Waterbirds training set), while $\eta_u$ can be controlled by the practitioner when collecting unlabeled data for W2S fine-tuning.

Figures 2 and 3 validate the theory in Section 2.2 through synthetic Gaussian experiments, with fixed $d_z = 2048$ in all experiments. We begin by examining how varying $\eta_u$ affects W2S gains under different values of $\eta_\ell$. As shown in Figure 2, when $\|\boldsymbol{\Xi}\|_F^2$ is small (a distinct teacher-student pair), W2S gains are maximized at $\eta_u \approx \eta_\ell$ for both balanced ($\eta_\ell = 0.5$) and highly spurious ($\eta_\ell = 0.1$) unlabeled data. This holds for both the average test risk and the minority test risk, consistent with Remark 2(a). Moreover, the magnitude of the W2S gain decreases as $\nu_z$ increases, reflecting the role of $\mathcal{E}_S$ in Theorem 2. Figure 3 left shows that as $\|\boldsymbol{\mu}_S\|_2^2$ increases so that $\|\boldsymbol{\Xi}\boldsymbol{\mu}_S\|_2^2$ becomes non-negligible compared to $\|\boldsymbol{\mu}_T\|_2^2$, the optimal value $\eta_u^\star$ gradually shifts away from $\eta_\ell$. This indicates that in some special cases $\eta_u^\star$ may not lie near $\eta_\ell$, consistent with Remark 2(b). Figure 3 right illustrates that the W2S gain decreases as the teacher-student similarity $\|\boldsymbol{\Xi}\|_F^2$ increases, consistent with Remark 2(c).

# 3 REAL-WORLD EVALUATION

Now we extend our theoretical understanding of W2S under spurious correlation to real-world tasks. We first leverage the theoretical framework to interpret our findings on how spurious correlations affect W2S performance across real-world benchmarks.

## 3.1 MODEL AND DATASET SETUP

**Pre-trained models.** Our weak teachers and strong students are drawn from a diverse set of pre-trained vision backbones that differ in architecture and training paradigm. Specifically, we consider ResNet-18 (ResNet18) (He et al., 2016), CLIP ViT-B/32 (Clipb32) (Radford et al., 2021), ConvNeXt-L (ConvNeXt) (Liu et al., 2022), DINOv2 ViT-L/14 (DINOv2) (Oquab et al., 2023), and MAE ViT-B/16 (MAE) (He et al., 2022). For each experiment on a given dataset, we include all teacher—student pairs whose relative strength (measured by accuracy) remains stable when we vary parameters including $\eta_\ell, \eta_u, N$, or $n$. We freeze all backbone parameters, view the pre-trained feature for the teacher and the student as $\varphi_T$ and $\varphi_S$, and only finetune the classification head.

**Datasets.** From both theoretical and practical perspectives, effective W2S requires that the pre-trained weak teacher and strong student have learned feature representations that are useful for the downstream task. Therefore, we evaluate W2S performance on widely used spurious correlation benchmarks that are relatively close to the pre-training distribution. These include Waterbirds (Sagawa et al., 2020), BFFHQ (Lee et al., 2021), and ImageNet-9 (Xiao et al., 2020), which contain spurious correlations between background and bird labels, age and gender labels, and background and object labels, respectively. We further provide a self-generated dataset, BG-COCO, by creating spurious correlations between cats/dogs from COCO (Lin et al., 2014) and indoor/outdoor scenes from Places (Zhou et al., 2017). We note that in all four datasets, the spurious correlation arises from highly imbalanced group proportions between the majority and minority groups. We denote the minority group proportion in the original training set of each dataset as $\eta_o$, which equals 0.05, 0.005, 0, and 0.05 for Waterbirds, BFFHQ, ImageNet-9, and BG-COCO, respectively. Detailed configurations of the dataset splits are provided in Appendix E.1.

## 3.2 INTERPRETING W2S UNDER SPURIOUS CORRELATIONS

We investigate how the proportion of the minority group in the unlabeled data affects W2S performance when the labeled data are either group-balanced or group-imbalanced. Specifically, we fix $\eta_\ell = 0.5$ and $\eta_\ell = \eta_o$, respectively, and vary $\eta_u$ while recording the change in W2S gain.[4] Figure 4 presents the average W2S gain across all teacher–student pairs on all four datasets. More comprehensive results are provided in Appendix E.2.

Our results show that, for both average accuracy ($\eta_t = 0.5$) and worst group accuracy ($\eta_t = 1$), increasing the minority proportion in the unlabeled data improves W2S performance when the weak teacher is free of spurious correlation ($\eta_\ell = 0.5$). Moreover, when the weak teacher itself encodes spurious correlation ($\eta_\ell = \eta_o$), the W2S gain is consistently positive across all four datasets at $\eta_u = \eta_o$, but surprisingly decreases for more balanced data as $\eta_u$ increases from $\eta_o$ to 0.5. Overall, W2S gain is negatively affected as the gap between $\eta_u$ and $\eta_\ell$ increases. These observations echo our theory and synthetic experiments (see Theorems 1 and 2, Remark 2, and Figure 2), showing that our theoretical findings on regression extends natually to broader, real-world classification problems.

---

[4]For classification tasks, W2S gain refers to the improvement in test accuracy achieved by the strong student after W2S fine-tuning over its weak teacher.

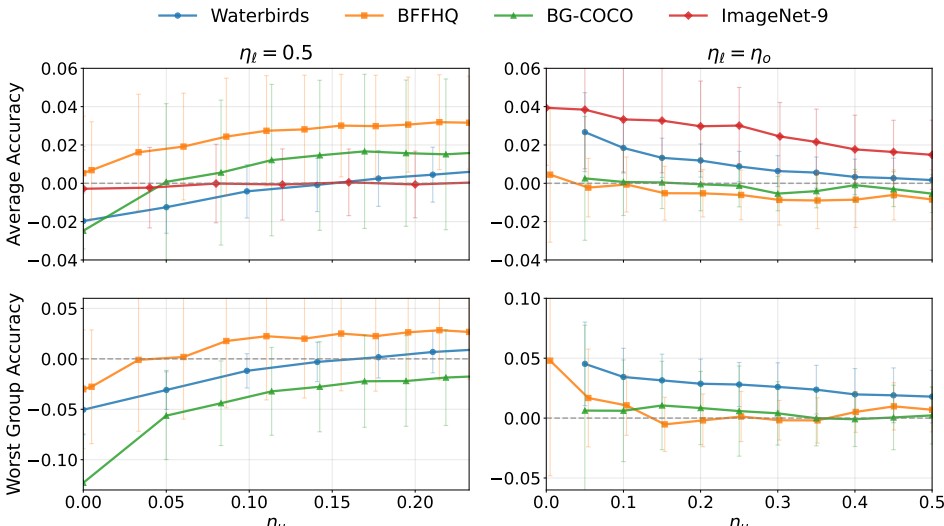

Figure 4: Average W2S gain across all teacher-student pairs as a function of $\eta_u$ on all four datasets. Top row: average accuracy; bottom row: worst group accuracy. Left column fixes $\eta_\ell = 0.5$; right column fixes $\eta_\ell = \eta_o$. For $\eta_\ell = 0.5$, curves are plotted over a shared $\eta_u$ interval aligned across datasets (bounded by minority group sample availability) to enable direct comparability. For $\eta_\ell = \eta_o$, each dataset is plotted from its own $\eta_o$ (0.05, 0.005, 0.05, and 0 for Waterbirds, BFFHQ, BG-COCO, and ImageNet-9, respectively) up to 0.5. ImageNet-9 does not have a clearly defined worst group and is therefore omitted from the bottom panels.

## 4 ENHANCED-W2S METHOD

Inspired by the theory and observations in Sections 2 and 3, we introduce a simple retraining method based on student confidence and generalized cross-entropy to strengthen W2S under spurious correlations. We show that this approach remarkably outperforms vanilla W2S across multiple datasets and large pre-trained backbones, without requiring any group annotations.

**Method.** Both our theoretical results and empirical findings indicate that W2S gain is noticeably reduced when there is a large discrepancy between the minority proportions of the unlabeled data and the labeled data. Therefore, we propose a simple method that requires no group label annotations and is capable of improving W2S gain in two particularly important settings: one where the labeled data are heavily affected by spurious correlation while the unlabeled data are free of it ($\eta_\ell = \eta_o, \eta_u = 0.5$), and the other where the unlabeled data suffer from spurious correlation while the labeled data are balanced ($\eta_\ell = 0.5, \eta_u = \eta_o$).

Formally, let the unlabeled data be $\hat{\mathcal{S}} = \{(\mathbf{x}_i, \hat{y}_i) \mid i \in [N]\}$, where $\hat{y}_i$ is the pseudolabel given by the weak teacher on $\mathbf{x}_i \in \mathcal{S}_x$. Our method enhances W2S gain by retraining the strong student after W2S fine-tuning, based on two components: (i) selecting a fraction $p \in (0, 1]$ of $\hat{\mathcal{S}}$ consisting of the samples with the highest student confidence (equivalently, the lowest entropy), and (ii) applying the generalized cross-entropy (GCE) loss (Zhang & Sabuncu, 2018) with parameter $q \in (0, 1]$ to each $(\mathbf{x}_i, \hat{y}_i)$ in this subset:

$$\mathcal{L}_{\text{GCE}}(\mathbf{x}_i, \hat{y}_i; q) = \frac{1 - \mathbf{p}_{\hat{y}_i}(\mathbf{x}_i)^q}{q},$$

where $\mathbf{p}_{\hat{y}_i}(\mathbf{x}_i)$ is the softmax probability that the student assigns to the pseudolabel $\hat{y}_i$ for $\mathbf{x}_i$.

In both settings ($\eta_\ell = \eta_o, \eta_u = 0.5$ and $\eta_\ell = 0.5, \eta_u = \eta_o$), selecting a high-confidence subset of the student's predictions filters for samples where all relevant features are clearly expressed and effectively used during prediction, thus preventing the strong student from over-relying on any single (potentially spurious) feature. Moreover, unlike the CE loss which imposes overly strong penalties on high-confidence but incorrect pseudolabels, applying the GCE loss to the selected subset mitigates

the negative impact of pseudolabel noise from the weak teacher.[5] More importantly, for the case $\eta_\ell = \eta_o, \eta_u = 0.5$, confidence-based selection further provides a crucial benefit. As shown in Appendix E.3, the high-confidence subset tends to filter out a larger fraction of minority samples to effectively reduce the new $\eta_u$ during retraining. This observation aligns with our theoretical prediction that when $\eta_\ell = \eta_o$, decreasing $\eta_u$ from $0.5$ leads to improved W2S gain.

| Dataset | $\eta_\ell$ | $\eta_u$ | Teacher–Student pair | | | | | | | | | |
|---|---|---|---|---|---|---|---|---|---|---|---|---|
| | | | DINOv2 ConvNeXt | DINOv2 Clipb32 | DINOv2 ResNet18 | DINOv2 MAE | ConvNeXt Clipb32 | ConvNeXt ResNet18 | ConvNeXt MAE | Clipb32 ResNet18 | Clipb32 MAE | ResNet18 MAE |
| Waterbirds | 0.5 | $\eta_o$ | 6.60 | 11.29 | 7.34 | 16.68 | 3.79 | 2.05 | 6.28 | — | 2.07 | 0.77 |
| | $\eta_o$ | 0.5 | 7.19 | 13.86 | 11.73 | 11.62 | 2.85 | 2.02 | 4.33 | — | 1.32 | 14.54 |
| BFFHQ | 0.5 | $\eta_o$ | 6.85 | 2.75 | 8.42 | 4.93 | 4.05 | — | — | 6.54 | 5.12 | — |
| | $\eta_o$ | 0.5 | 3.92 | 8.53 | 2.02 | 4.56 | 2.09 | — | — | 2.06 | -1.37 | — |
| BG-COCO | 0.5 | $\eta_o$ | 5.38 | 13.40 | 12.88 | 24.01 | 9.82 | 6.49 | 15.25 | 3.39 | 12.43 | 2.05 |
| | $\eta_o$ | 0.5 | 10.21 | 16.99 | 12.25 | -3.52 | 3.41 | 1.21 | -3.07 | 3.48 | 0.31 | 3.70 |
| ImageNet-9 | 0.5 | $\eta_o$ | — | 6.03 | 7.45 | 24.11 | 4.74 | 5.30 | 18.49 | 4.22 | 21.73 | 17.98 |
| | $\eta_o$ | 0.5 | — | 8.21 | 11.28 | 22.00 | 3.77 | 1.81 | 10.50 | 4.51 | 23.24 | 15.76 |

Table 1: Relative improvement of Enhanced-W2S over vanilla W2S (%, measured by average accuracy) across all datasets and teacher–student pairs. Each entry reports the mean improvement over all $N, n$ combinations. For each model pair in the table header, the assignment of weak teacher and strong student depends on the dataset. We report for each dataset only those model pairs whose relative strength relationship remains consistent across different $(\eta_\ell, \eta_u)$ settings within that dataset.

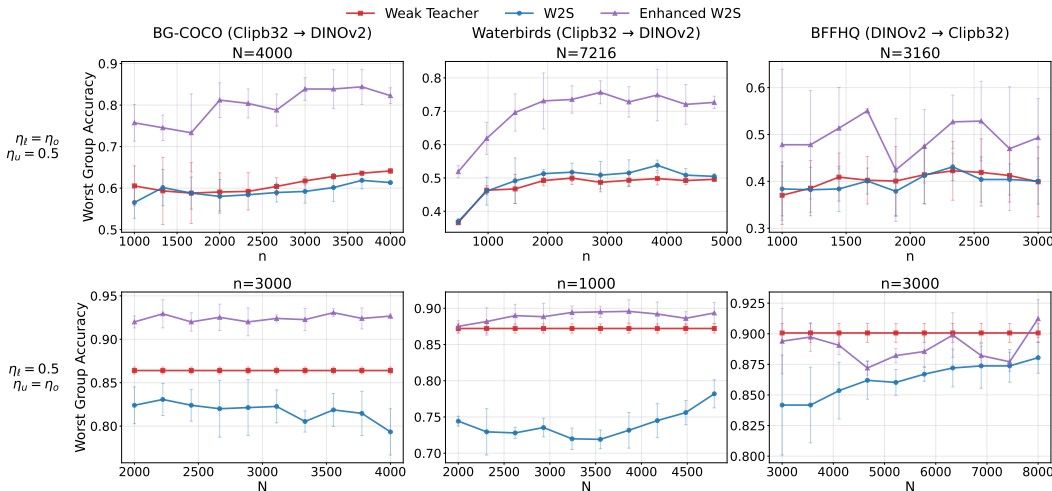

Figure 5: Comparison of Enhanced-W2S and original W2S for the (Clipb32, DINOv2) pair on BG-COCO, Waterbirds, and BFFHQ. Top row: worst group accuracy with $\eta_\ell = \eta_o$, $\eta_u = 0.5$ (fixed $N$, varying $n$). Bottom row: worst group accuracy with $\eta_\ell = 0.5$, $\eta_u = \eta_o$ (fixed $n$, varying $N$).

**Main results.** We evaluate our Enhanced-W2S method across all four datasets. Table 1 reports the relative improvement of Enhanced-W2S over vanilla W2S for each teacher–student pair. Figure 5 further visualizes the performance of Enhanced-W2S versus vanilla W2S for a representative model pair. Overall, for both average accuracy and worst group accuracy, Enhanced-W2S achieves consistent and substantial improvements over vanilla W2S under both $(\eta_\ell, \eta_u)$ settings. Specifically, Table 1 shows that 67 out of 70 model pairs exhibit a positive gain (measured by average accuracy), with the

---

[5]In our Enhanced-W2S method, the role of GCE loss is analogous to its original use in Zhang & Sabuncu (2018) for handling noisy labels. Different from the setting studied in Nam et al. (2020), where GCE loss on ground-truth labeled datasets with spurious correlations was observed to amplify bias, in our method GCE loss is applied to a pseudolabeled dataset restricted to a high-confidence subset, and thus serves a fundamentally different role.

mean relative improvements across all pairs reaching 7.02% (Waterbirds), 4.32% (BFFHQ), 7.50% (BG-COCO), and 11.73% (ImageNet-9). In addition, the relative improvement of Enhanced-W2S in terms of worst group accuracy across all pairs is 21.14% (Waterbirds), 3.81% (BFFHQ), and 7.73% (BG-COCO). Further details are provided in Appendix E.3.

| Comparison | Metric ($\times 10^{-2}$) | Waterbirds $(\eta_\ell, \eta_u)$ | | BFFHQ $(\eta_\ell, \eta_u)$ | | BG-COCO $(\eta_\ell, \eta_u)$ | | ImageNet-9 $(\eta_\ell, \eta_u)$ | |
|---|---|---|---|---|---|---|---|---|---|
| | | $(0.5, \eta_o)$ | $(\eta_o, 0.5)$ | $(0.5, \eta_o)$ | $(\eta_o, 0.5)$ | $(0.5, \eta_o)$ | $(\eta_o, 0.5)$ | $(0.5, \eta_o)$ | $(\eta_o, 0.5)$ |
| Enhanced W2S $-$ Enhanced W2S ($q \to 0$) | Average | 3.27 | 1.00 | 3.51 | 0.46 | 3.41 | 0.52 | 1.00 | 0.51 |
| | Worst | 5.15 | 2.39 | 4.34 | 1.90 | 3.79 | 1.46 | — | — |
| Enhanced W2S $-$ Enhanced W2S ($p = 1$) | Average | 2.79 | 4.84 | 2.77 | 2.69 | 4.97 | 3.37 | 4.51 | 5.08 |
| | Worst | 3.57 | 8.58 | 2.75 | 3.73 | 6.26 | 5.44 | — | — |

Table 2: Ablation across four datasets: improvements of Enhanced-W2S over two baselines, using only the CE loss (i.e., the $q \to 0$ limit of GCE) and using all unlabeled data ($p = 1$), in terms of either average accuracy or worst group accuracy. For each dataset, improvements are computed as the mean across all model pairs whose relative strength relationship remains consistent under different $(\eta_\ell, \eta_u)$ settings. ImageNet-9 has no well-defined worst group, so those entries are omitted.

**Ablation study.** We conduct controlled ablations to examine the contribution of the two key components of our Enhanced-W2S methods, namely the use of the GCE loss and confidence-based selection. Specifically, we conduct two separate ablations: (i) replacing the GCE loss with the standard CE loss, and (ii) replacing confidence-based selection with using the full unlabeled dataset. We then retrain the model under each variant and compare the results with the original Enhanced-W2S method. Table 2 shows that under both $(\eta_\ell, \eta_u)$ settings, the GCE loss and confidence-based selection each play a positive role in improving W2S gain. When $\eta_\ell = \eta_o, \eta_u = 0.5$, the impact of using CE loss is consistently smaller than that of using the full unlabeled dataset; whereas under $\eta_\ell = 0.5, \eta_u = \eta_o$, the effects of the two ablations are more comparable. This suggests that filtering out minority group samples via high-confidence selection plays a more critical role in the former setting, while in the latter setting the contributions of GCE and high-confidence selection are comparable.

## 5 CONCLUSIONS, DISCUSSIONS, AND FUTURE DIRECTIONS

In this work, we start with a theoretical framework that models W2S generalization under spurious correlations induced by group imbalance. Within this framework, we precisely characterize how different factors, such as the proportions of minority groups in labeled and unlabeled data and the teacher-student similarity, affect W2S, which is validated through extensive synthetic experiments and on diverse real-world tasks. Inspired by our analysis, we proposed Enhanced-W2S, a confidence-based retraining algorithm that does not require any group labels and substantially improves W2S gains when the labeled or unlabeled data are highly group-imbalanced. The effectiveness of this approach is demonstrated across assorted real-world datasets.

It is important to emphasize that spurious correlations in W2S constitute a critical issue that deserves closer attention. Beyond standard benchmarks, such correlations can pose substantial risks: in socially sensitive domains they may reflect demographic biases, and in safety-critical applications they can degrade reliability under rare but high-stakes conditions. While our experiments focus on public computer vision benchmarks, the mechanisms we analyze are broadly relevant. Our algorithm provides the first attempt to improve W2S in this setting, and we hope this work will inspire further efforts toward more reliable and efficient W2S methods.

Meanwhile, from a more technical perspective, another exciting future direction is to investigate W2S generation (with or without spurious correlation) beyond the kernel regime by taking the training dynamics of the teacher and student models, conditioned on their pre-trained initializations, into consideration.

ACKNOWLEDGMENTS

The authors would like to thank Jason D. Lee and Denny Wu for insightful discussions. YD acknowledges support of the NYU Courant Instructorship. QL acknowledges support of NSF DMS-2523382 and DOE Office of Science under Award #DE-SC0024721.

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

# Appendices

## A    USAGE OF LARGE LANGUAGE MODELS

Large language models were used in a limited manner to (i) search for related literature, (ii) check grammar/phrasing, and (iii) make stylistic adjustments.

## B    ADDITIONAL RELATED WORKS

**Knowledge distillation.** Knowledge distillation (KD) (Hinton et al., 2015) is closely related to W2S but with the roles reversed: KD transfers knowledge from a larger teacher model to a smaller student model. A series of works has analyzed when and why a distilled student generalizes (Phuong & Lampert, 2019; Stanton et al., 2021; Ojha et al., 2023; Nagarajan et al., 2023; Dong et al., 2024; Ildiz et al., 2025). Analytically, W2S departs from traditional KD because "weak" vs. "strong" is defined relative to pretraining, so W2S is naturally studied as fine-tuning on pseudolabeled data.

**Group robustness and spurious correlation.** Extensive efforts have been devoted to mitigating spurious correlation for robust and safe generalization to unseen test domains (Arjovsky et al., 2019; Sagawa et al., 2020; Krueger et al., 2021; Deng et al., 2023; Phan et al., 2024; Wen et al., 2025; Liu et al., 2025b; Bombari & Mondelli, 2025). Among these studies, a subset of work specifically targets spurious correlation arising from group imbalance. When group labels are available, canonical approaches include reweighting minority groups (Sagawa et al., 2020), downsampling majority groups (Deng et al., 2023), distributionally robust optimization (Sagawa et al., 2020; Zhang et al., 2020), and progressive data expansion (Deng et al., 2023). Since obtaining group annotations in training data can be costly or even infeasible, alternative strategies aim to identify biased samples without explicit group supervision (Nam et al., 2020; Liu et al., 2021; Zhang et al., 2022; Yenamandra et al., 2023; Han & Zou, 2024), or leverage auxiliary signals such as knowledge of spurious attributes (Puli et al., 2021), class annotations (LaBonte et al., 2024), and superclass-level information (Liu et al., 2025a).

**Multi-round retraining and confidence-based selection.** Multi-round retraining and confidence-based data selection are widely adopted ideas that have been leveraged, independently, to improve model performance for W2S generalization (Burns et al., 2024; Liu & Alahi, 2024), mitigate spurious correlation (Liu et al., 2021; Nam et al., 2020), and in the broader literature on self-training (Xie et al., 2020; Yu et al., 2021) and co-training (Zhang & Zhou, 2011; Ma et al., 2020). Our theory provides a

principled motivation to combine these practical techniques, bringing an effective algorithmic remedy for the failures of W2S under spurious correlation.

## C   ADDITIONAL NOTATIONS

For any $n \in \mathbb{N}$, let $[n] = \{1, 2, \ldots, n\}$. $\mathbf{e}_i$ is the $i$-th canonical basis of a conformable dimension. We adapt the standard big-O notations for functions of multiple variables: for two functions $f, g : \mathbb{N}^k \to \mathbb{R}_{\geqslant 0}$, $f(\mathbf{n}) = O(g(\mathbf{n}))$ means that there exists a constant $C > 0$ such that $f(\mathbf{n}) \leqslant Cg(\mathbf{n})$ for all $\mathbf{n} \in \mathbb{N}^k$; $f(\mathbf{n}) = \Omega(g(\mathbf{n}))$ means that there exists a constant $c > 0$ such that $f(\mathbf{n}) \geqslant cg(\mathbf{n})$ for all $\mathbf{n} \in \mathbb{N}^k$; $f(\mathbf{n}) = \Theta(g(\mathbf{n}))$ means that $f(\mathbf{n}) = O(g(\mathbf{n}))$ and $f(\mathbf{n}) = \Omega(g(\mathbf{n}))$; $f(\mathbf{n}) = o(g(\mathbf{n}))$ means that $\lim_{\min_i n_i \to \infty} f(\mathbf{n})/g(\mathbf{n}) = 0$. For a scalar quantity $f(n) \geqslant 0$ depending on $n \in \mathbb{N}$, $f(n) = O_{\mathbb{P}}(g(n))$ means that for any $\delta \in (0, 1)$, there exists a constant $C(\delta) > 0$, independent of $n$, and a natural number $N(\delta) \in \mathbb{N}$ such that $\Pr[f(n) \leqslant C(\delta)g(n)] \geqslant 1 - \delta$ for all $n \geqslant N(\delta)$.

## D   PROOFS FOR SECTION 2.2

We first observe that when $n > d_T$, (1) can be equivalently written as $\boldsymbol{\theta}_T = \mathbf{U}_T \boldsymbol{\beta}_T$ where

$$\boldsymbol{\beta}_T = \operatorname*{argmin}_{\boldsymbol{\beta} \in \mathbb{R}^{d_T}} \frac{1}{n} \sum_{i=1}^{n} (\phi_T(\widetilde{\mathbf{x}}_i)^\top \boldsymbol{\beta} - \widetilde{y}_i)^2. \tag{5}$$

Analogously, (2) can be equivalently written as $\boldsymbol{\theta}_S = \mathbf{U}_S \boldsymbol{\beta}_S$ where

$$\boldsymbol{\beta}_S = \operatorname*{argmin}_{\boldsymbol{\beta} \in \mathbb{R}^{d_S}} \frac{1}{N} \sum_{i=1}^{N} (\phi_S(\mathbf{x}_i)^\top \boldsymbol{\beta} - f_T(\mathbf{x}_i))^2. \tag{6}$$

### D.1   SFT OF WEAK TEACHER

We start by considering the population-optimal linear predictor over the weak teacher representation, $\phi_T(\cdot)$: as $n \to \infty$, (5) converges to

$$\boldsymbol{\beta}_T^\infty = \operatorname*{argmin}_{\boldsymbol{\beta} \in \mathbb{R}^{d_T}} \mathbb{E}_{(\mathbf{x}, y) \sim \mathcal{D}_{\mathbf{x}, y}(\eta_\ell)}[(\phi_T(\mathbf{x})^\top \boldsymbol{\beta} - y)^2]. \tag{7}$$

**Lemma 1** (Population SFT of weak teacher). *When supervisedly fine-tuned over the population, the weak teacher from (7) satisfies $f_T^\infty(\mathbf{x}) = \phi_T(\mathbf{x})^\top \boldsymbol{\beta}_T^\infty = \mathbf{z}(\mathbf{x})^\top \boldsymbol{\beta}_* = f_*$.*

*Proof of Lemma 1.* Notice that (7) admits a closed-form solution

$$\boldsymbol{\beta}_T^\infty = \boldsymbol{\Sigma}_{\phi_T, \eta_\ell}^{-1} \boldsymbol{\mu}_{\phi_T, \eta_\ell}, \quad \boldsymbol{\Sigma}_{\phi_T, \eta_\ell} = \mathbb{E}_{\mathcal{D}(\eta_\ell)}[\phi_T(\mathbf{x})\phi_T(\mathbf{x})^\top], \quad \boldsymbol{\mu}_{\phi_T, \eta_\ell} = \mathbb{E}_{\mathcal{D}(\eta_\ell)}[\phi_T(\mathbf{x})y].$$

Since $\mathbf{z}(\mathbf{x})$ and $\mathbf{w}(\mathbf{x})$ are independent, we have

$$
\begin{aligned}
\boldsymbol{\Sigma}_{\phi_T, \eta_\ell} &= \mathbb{E}_{\mathcal{D}(\eta_\ell)}\left[(\mathbf{z}(\mathbf{x}) \otimes \mathbf{w}(\mathbf{x}))(\mathbf{z}(\mathbf{x}) \otimes \mathbf{w}(\mathbf{x}))^\top\right] \\
&= \mathbb{E}_{\mathcal{D}(\eta_\ell)}\left[\mathbf{z}(\mathbf{x})\mathbf{z}(\mathbf{x})^\top\right] \otimes \mathbb{E}_{\mathcal{D}(\eta_\ell)}\left[\mathbf{w}(\mathbf{x})\mathbf{w}(\mathbf{x})^\top\right] \\
&= \mathbf{I}_{d_z} \otimes \mathbf{C}_T(\eta_\ell),
\end{aligned}
\tag{8}
$$

where

$$\mathbf{C}_T(\eta_\ell) = \begin{bmatrix} 1 & \eta_\ell \boldsymbol{\mu}_T^\top \\ \eta_\ell \boldsymbol{\mu}_T & \sigma_\xi^2 \mathbf{I}_{p_T-1} + \eta_\ell^2 \boldsymbol{\mu}_T \boldsymbol{\mu}_T^\top \end{bmatrix} = \operatorname{diag}\left(0, \sigma_\xi^2 \mathbf{I}_{p_T-1}\right) + \begin{bmatrix} 1 \\ \eta_\ell \boldsymbol{\mu}_T \end{bmatrix} \begin{bmatrix} 1 & \eta_\ell \boldsymbol{\mu}_T \end{bmatrix}, \tag{9}$$

whose inverse can be computed via block matrix inversion as

$$\mathbf{C}_T(\eta_\ell)^{-1} = \begin{bmatrix} 1 + \sigma_\xi^{-2} \|\eta_\ell \boldsymbol{\mu}_T\|_2^2 & -\sigma_\xi^{-2} \eta_\ell \boldsymbol{\mu}_T^\top \\ -\sigma_\xi^{-2} \eta_\ell \boldsymbol{\mu}_T & \sigma_\xi^{-2} \mathbf{I}_{p_T-1} \end{bmatrix}. \tag{10}$$

Meanwhile, by the independence of $\mathbf{z}(\mathbf{x})$ and $\mathbf{w}(\mathbf{x})$, we have

$$
\begin{aligned}
\boldsymbol{\mu}_{\phi_T, \eta_\ell} &= \mathbb{E}_{\mathcal{D}(\eta_\ell)}\left[\mathbf{z}(\mathbf{x}) \otimes \mathbf{w}(\mathbf{x})(\mathbf{z}(\mathbf{x})^\top \boldsymbol{\beta}_* + \epsilon)\right] \\
&= \left(\mathbb{E}_{\mathcal{D}(\eta_\ell)}\left[\mathbf{z}(\mathbf{x})\mathbf{z}(\mathbf{x})^\top\right] \boldsymbol{\beta}_*\right) \otimes \mathbb{E}_{\mathcal{D}(\eta_\ell)}\left[\mathbf{w}(\mathbf{x})\right] \\
&= \boldsymbol{\beta}_* \otimes \begin{bmatrix} 1 \\ \eta_\ell \boldsymbol{\mu}_T \end{bmatrix}.
\end{aligned}
$$

Therefore, the population-optimal linear predictor over $\phi_T$ is given by

$$
\begin{aligned}
\boldsymbol{\beta}_T^\infty =& \boldsymbol{\Sigma}_{\phi_T,\eta_\ell}^{-1} \boldsymbol{\mu}_{\phi_T,\eta_\ell} = (\mathbf{I}_{d_z} \otimes \mathbf{C}_T(\eta_\ell))^{-1} \left( \boldsymbol{\beta}_* \otimes \begin{bmatrix} 1 \\ \eta_\ell \boldsymbol{\mu}_T \end{bmatrix} \right) = \boldsymbol{\beta}_* \otimes \left( \mathbf{C}_T(\eta_\ell)^{-1} \begin{bmatrix} 1 \\ \eta_\ell \boldsymbol{\mu}_T \end{bmatrix} \right) \\
=& \boldsymbol{\beta}_* \otimes \left( \begin{bmatrix} 1 + \sigma_\xi^{-2} \|\eta_\ell \boldsymbol{\mu}_T\|_2^2 & -\sigma_\xi^{-2} \eta_\ell \boldsymbol{\mu}_T^\top \\ -\sigma_\xi^{-2} \eta_\ell \boldsymbol{\mu}_T & \sigma_\xi^{-2} \mathbf{I}_{p_T - 1} \end{bmatrix} \begin{bmatrix} 1 \\ \eta_\ell \boldsymbol{\mu}_T \end{bmatrix} \right) = \boldsymbol{\beta}_* \otimes \mathbf{e}_1,
\end{aligned}
$$

where $\mathbf{e}_1 \in \mathbb{R}^{p_T}$ is the first canonical basis. $\qquad \square$

While the $f_T^\infty = f_*$ achieves the optimal population risk $\mathcal{R}(f_T^\infty) = \mathcal{R}(f_*) = \sigma_y^2$, the inefficient representation ($d_T = d_z p_T \gg d_z$) and the entangled features of $\phi_T$ make the finite-sample generalization challenging, especially under spurious correlations, as we will show next.

**Theorem 1** (SFT of weak teacher (Appendix D.1)). *Under Assumption 1, (1) satisfies*

$$
\mathbb{E}_{\mathcal{D}(\eta_\ell)^n} \left[ \mathbf{ER}_{\eta_t}(f_T) \right] \xrightarrow{\mathbb{P}} \sigma_y^2 \gamma_z \Big( \underbrace{\boxed{p_T}}_{\mathcal{V}_T^{(0)} \text{ from label noise}} + \underbrace{\boxed{\frac{\|(\eta_t - \eta_\ell) \boldsymbol{\mu}_T\|_2^2}{\sigma_\xi^2}}}_{\mathcal{V}_T^{(1)} \text{ from spurious correlations}} \Big).
$$

*Proof of Theorem 1.* For a small labeled set $\widetilde{\mathcal{S}} = \{(\widetilde{\mathbf{x}}_i, \widetilde{y}_i) \mid i \in [n]\} \sim \mathcal{D}(\eta_\ell)^n$, the SFT in (1) admits a closed-form solution

$$
\boldsymbol{\beta}_T = (\widetilde{\boldsymbol{\Phi}}_T^\top \widetilde{\boldsymbol{\Phi}}_T)^{-1} \widetilde{\boldsymbol{\Phi}}_T^\top \widetilde{\mathbf{y}}, \tag{11}
$$

where $\widetilde{\boldsymbol{\Phi}}_T = [\phi_T(\widetilde{\mathbf{x}}_1), \cdots, \phi_T(\widetilde{\mathbf{x}}_n)]^\top \in \mathbb{R}^{n \times d_T}$ and $\widetilde{\mathbf{y}} = [\widetilde{y}_1, \cdots, \widetilde{y}_n]^\top \in \mathbb{R}^n$. Since Lemma 1 shows that the population-optimal linear predictor over $\phi_T$ is $f_T^\infty(\mathbf{x}) = \phi_T(\mathbf{x})^\top \boldsymbol{\beta}_T^\infty = f_*(\mathbf{x})$, we have $\widetilde{\mathbf{y}} = \widetilde{\boldsymbol{\Phi}}_T \boldsymbol{\beta}_T^\infty + \widetilde{\boldsymbol{\epsilon}}$ where $\widetilde{\boldsymbol{\epsilon}} \sim \mathcal{N}(\mathbf{0}_n, \sigma_y^2 \mathbf{I}_n)$. Therefore, we observe that

$$
\boldsymbol{\beta}_T - \boldsymbol{\beta}_T^\infty = (\widetilde{\boldsymbol{\Phi}}_T^\top \widetilde{\boldsymbol{\Phi}}_T)^{-1} \widetilde{\boldsymbol{\Phi}}_T^\top \widetilde{\boldsymbol{\epsilon}}.
$$

Since the excess risk over the test distribution $\mathcal{D}(\eta_t)$ is given by

$$
\begin{aligned}
\mathbf{ER}_{\eta_t}(f_T) =& \mathbb{E}_{\mathcal{D}(\eta_t)}[(f_T(\mathbf{x}) - f_*(\mathbf{x}))^2] = \mathbb{E}_{\mathcal{D}(\eta_t)}[(\phi_T(\mathbf{x})^\top (\boldsymbol{\beta}_T - \boldsymbol{\beta}_T^\infty))^2] \\
=& \|\boldsymbol{\beta}_T - \boldsymbol{\beta}_T^\infty\|_{\boldsymbol{\Sigma}_{\phi_T, \eta_t}}^2,
\end{aligned}
$$

where $\boldsymbol{\Sigma}_{\phi_T, \eta_t} = \mathbb{E}_{\mathcal{D}(\eta_t)}[\phi_T(\mathbf{x})\phi_T(\mathbf{x})^\top]$, let $\widetilde{\boldsymbol{\Sigma}}_n = \frac{1}{n} \widetilde{\boldsymbol{\Phi}}_T^\top \widetilde{\boldsymbol{\Phi}}_T \in \mathbb{R}^{d_T \times d_T}$ be the sample covariance matrix of $\phi_T(\widetilde{\mathbf{x}})$ over $\widetilde{\mathbf{x}} \sim \mathcal{D}_{\mathbf{x}}(\eta_\ell)$, we have

$$
\begin{aligned}
\mathbb{E}_{\widetilde{\mathcal{S}} \sim \mathcal{D}(\eta_\ell)^n} \left[ \mathbf{ER}_{\eta_t}(f_T) \right] =& \mathrm{tr} \left( \mathbb{E}_{\widetilde{\mathcal{S}} \sim \mathcal{D}(\eta_\ell)^n} \left[ \boldsymbol{\Sigma}_{\phi_T, \eta_t} (\boldsymbol{\beta}_T - \boldsymbol{\beta}_T^\infty)(\boldsymbol{\beta}_T - \boldsymbol{\beta}_T^\infty)^\top \right] \right) \\
=& \sigma_y^2 \, \mathrm{tr} \left( \boldsymbol{\Sigma}_{\phi_T, \eta_t} \mathbb{E}_{\widetilde{\mathcal{S}} \sim \mathcal{D}(\eta_\ell)^n} \left[ (\widetilde{\boldsymbol{\Phi}}_T^\top \widetilde{\boldsymbol{\Phi}}_T)^{-1} \right] \right) \\
=& \frac{\sigma_y^2}{n} \, \mathrm{tr} \left( \boldsymbol{\Sigma}_{\phi_T, \eta_t} \widetilde{\boldsymbol{\Sigma}}_n^{-1} \right).
\end{aligned} \tag{12}
$$

Recall $\phi_T(\mathbf{x}) = \mathbf{z}(\mathbf{x}) \otimes \mathbf{w}(\mathbf{x})$ and notice that for any $\eta \in [0, 1]$,

$$
\mathbb{E}_{\mathcal{D}(\eta)}[\phi_T(\mathbf{x})] = \mathbb{E}_{\mathcal{D}(\eta)}[\mathbf{z}(\mathbf{x})] \otimes \mathbb{E}_{\mathcal{D}(\eta)}[\mathbf{w}(\mathbf{x})] = \mathbf{0}_{d_T},
$$

while the derivation of (8) suggests that for any $\eta \in [0, 1]$,

$$
\boldsymbol{\Sigma}_{\phi_T, \eta} = \mathbb{E}_{\mathcal{D}(\eta)}[\phi_T(\mathbf{x})\phi_T(\mathbf{x})^\top] = \mathbf{I}_{d_z} \otimes \mathbf{C}_T(\eta). \tag{13}
$$

However, we notice that $\phi_T(\mathbf{x})$ is not multivariate Gaussian due to the non-Gaussianity of products of independent Gaussian variables and the dependence of entries in $\mathbf{z}(\mathbf{x}) \otimes \mathbf{w}(\mathbf{x})$. Therefore, $\widetilde{\boldsymbol{\Sigma}}_n$ cannot be directly computed using inverse Wishart. Instead, we leverage the concentration of $\widetilde{\boldsymbol{\Sigma}}_n$ in the proportional asymptotic limit (Assumption 1). In particular, Lemma 2 and (13) implies that as $d_z, n \to \infty$ with $d_z/n \to \gamma_z \in (0, p_T^{-1})$,

$$
\frac{\sigma_y^2}{n} \, \mathrm{tr} \left( \boldsymbol{\Sigma}_{\phi_T, \eta_t} \widetilde{\boldsymbol{\Sigma}}_n^{-1} \right) \xrightarrow{\mathbb{P}} \sigma_y^2 \gamma_z \, \mathrm{tr} \left( \mathbf{C}_T(\eta_t) \mathbf{C}_T(\eta_\ell)^{-1} \right). \tag{14}
$$

Leveraging the derivation of (9) and (10), we observe that

$$\text{tr}\left(\mathbf{C}_T(\eta_t)\mathbf{C}_T(\eta_\ell)^{-1}\right) = p_T + \sigma_\xi^{-2}\eta_\ell^2\|\boldsymbol{\mu}_T\|_2^2 - 2\sigma_\xi^{-2}\eta_t\eta_\ell\|\boldsymbol{\mu}_T\|_2^2 + \sigma_\xi^{-2}\eta_t^2\|\boldsymbol{\mu}_T\|_2^2$$

$$= p_T + \sigma_\xi^{-2}(\eta_t - \eta_\ell)^2\|\boldsymbol{\mu}_T\|_2^2 = p_T + (\eta_t - \eta_\ell)^2\frac{\|\boldsymbol{\mu}_T\|_2^2}{\sigma_\xi^2}, \quad (15)$$

and therefore, (14) becomes

$$\frac{\sigma_y^2}{n}\,\text{tr}\left(\boldsymbol{\Sigma}_{\phi_T,\eta_t}\widetilde{\boldsymbol{\Sigma}}_n^{-1}\right) \xrightarrow{\mathbb{P}} \sigma_y^2\frac{d_z}{n}\left(p_T + (\eta_t - \eta_\ell)^2\frac{\|\boldsymbol{\mu}_T\|_2^2}{\sigma_\xi^2}\right)$$

$$= \sigma_y^2\,\gamma_z\left(p_T + (\eta_t - \eta_\ell)^2\frac{\|\boldsymbol{\mu}_T\|_2^2}{\sigma_\xi^2}\right).$$

Plugging the above into (12) completes the proof. $\qquad\square$

**Lemma 2.** *For fixed $p_T \geqslant 2$, let $\mathbf{C} \in \mathbb{R}^{p_T \times p_T}$ be any fixed symmetric matrix with $\|\mathbf{C}\|_2 < \infty$. Recall $\widetilde{\boldsymbol{\Sigma}}_n = \frac{1}{n}\sum_{i=1}^n \phi_T(\widetilde{\mathbf{x}}_i)\phi_T(\widetilde{\mathbf{x}}_i)^\top$ where $\widetilde{\mathbf{x}}_i \sim \mathcal{D}_{\mathbf{x}}(\eta_\ell)$ i.i.d. for all $i \in [n]$. As $d_z, n \to \infty$ with $d_z/n \to \gamma_z \in (0, p_T^{-1})$,*

$$\frac{1}{n}\text{tr}\left((\mathbf{I}_{d_z} \otimes \mathbf{C})\widetilde{\boldsymbol{\Sigma}}_n^{-1}\right) \xrightarrow{\mathbb{P}} \gamma_z\,\text{tr}\left(\mathbf{C}\mathbf{C}_T(\eta_\ell)^{-1}\right).$$

*Proof of Lemma 2.* We observe that $\phi_T(\mathbf{x})\phi_T(\mathbf{x})^\top = (\mathbf{z}(\mathbf{x})\mathbf{z}(\mathbf{x})^\top) \otimes (\mathbf{w}(\mathbf{x})\mathbf{w}(\mathbf{x})^\top)$, and the sample covariance matrix $\widetilde{\boldsymbol{\Sigma}}_n$ can be partitioned into $d_z \times d_z$ blocks of size $p_T \times p_T$:

$$\widetilde{\boldsymbol{\Sigma}}_n = \left[\widetilde{\boldsymbol{\Sigma}}_n^{(k,l)}\right]_{k,l\in[d_z]} \quad \text{where} \quad \widetilde{\boldsymbol{\Sigma}}_n^{(k,l)} = \frac{1}{n}\sum_{i=1}^n z_k(\widetilde{\mathbf{x}}_i)z_l(\widetilde{\mathbf{x}}_i) \cdot \mathbf{w}(\widetilde{\mathbf{x}}_i)\mathbf{w}(\widetilde{\mathbf{x}}_i)^\top \in \mathbb{R}^{p_T \times p_T},$$

where for any $\mathbf{x} \in \mathcal{X}$, $z_k(\mathbf{x})$ is the $k$-th entry of $\mathbf{z}(\mathbf{x})$. Notice that $\mathbb{E}[\widetilde{\boldsymbol{\Sigma}}_n^{(k,l)}] = \delta_{k,l}\mathbf{C}_T(\eta_\ell)$ where $\delta_{k,l}$ is the Kronecker delta since $\mathbf{z}(\mathbf{x})$ and $\mathbf{w}(\mathbf{x})$ are independent, and $\mathbb{E}[z_k(\mathbf{x})z_l(\mathbf{x})] = \delta_{k,l}$ given $\mathbf{z}(\mathbf{x}) \sim \mathcal{N}(\mathbf{0}_{d_z}, \mathbf{I}_{d_z})$.

**Off-diagonal blocks are negligible.** For any $k, l \in [d_z]$ with $k \neq l$, we define $p_T \times p_T$ self-adjoint matrices,

$$\mathbf{Y}_i^{(k,l)} := \frac{1}{n}z_k(\widetilde{\mathbf{x}}_i)z_l(\widetilde{\mathbf{x}}_i)\left(\mathbf{w}(\widetilde{\mathbf{x}}_i)\mathbf{w}(\widetilde{\mathbf{x}}_i)^\top - \mathbf{C}_T(\eta_\ell)\right) \quad \text{and} \quad \mathbf{R}_i^{(k,l)} := \frac{1}{n}z_k(\widetilde{\mathbf{x}}_i)z_l(\widetilde{\mathbf{x}}_i)\mathbf{C}_T(\eta_\ell),$$

where recall from the derivation of (8) that $\mathbf{C}_T(\eta_\ell) = \mathbb{E}_{\mathcal{D}(\eta_\ell)}[\mathbf{w}(\mathbf{x})\mathbf{w}(\mathbf{x})^\top]$. Since $\mathbb{E}[z_k(\mathbf{x})z_l(\mathbf{x})] = \delta_{k,l}$, we have $\mathbb{E}[\mathbf{Y}_i^{(k,l)}] = \mathbb{E}[\mathbf{R}_i^{(k,l)}] = \mathbf{0}_{p_T \times p_T}$ for $k \neq l$, and

$$\widetilde{\boldsymbol{\Sigma}}_n^{(k,l)} = \sum_{i=1}^n \mathbf{Y}_i^{(k,l)} + \mathbf{R}_i^{(k,l)}, \quad \mathbb{E}[\widetilde{\boldsymbol{\Sigma}}_n^{(k,l)}] = \mathbf{0}_{p_T \times p_T}.$$

Let $L_n := 4\sqrt{\log(n)}$ and consider the event

$$E_n := \left\{\max_{i\in[n]}\|\mathbf{z}(\widetilde{\mathbf{x}}_i)\|_\infty^2 \leqslant L_n^2\right\} \wedge \left\{\max_{i\in[n]}\|\mathbf{w}(\widetilde{\mathbf{x}}_i)\|_2^2 \leqslant L_n^2\right\}.$$

First, for $\mathbf{z}(\widetilde{\mathbf{x}}_i) \sim \mathcal{N}(\mathbf{0}_{d_z}, \mathbf{I}_{d_z})$, the union bound and Gaussian tail bound imply that

$$\Pr\left[\max_{i\in[n]}\|\mathbf{z}(\widetilde{\mathbf{x}}_i)\|_\infty > \frac{L_n}{\sqrt{2}}\right] = \Pr\left[\max_{i\in[n],\,k\in[d_z]}|z_k(\widetilde{\mathbf{x}}_i)| > \frac{L_n}{\sqrt{2}}\right]$$

$$\leqslant 2nd_z\exp\left(-\frac{L_n^2}{4}\right) = \frac{2d_z}{n}\cdot n^{-2} = o(n^{-1}).$$

Meanwhile, we observe that for $\mathbf{g}_i \sim \mathcal{N}(\mathbf{0}_{p_T-1}, \mathbf{I}_{p_T-1})$,

$$\|\mathbf{w}(\widetilde{\mathbf{x}}_i)\|_2^2 = 1 + \left\|[\mathbf{w}(\widetilde{\mathbf{x}}_i)]_{2:p_T}\right\|_2^2 \leqslant 2\eta_\ell^2\|\boldsymbol{\mu}_T\|_2^2 + 2\sigma_\xi^2\|\mathbf{g}_i\|_2^2,$$

while the Laurent-Massart $\chi^2$ tail bound (Laurent & Massart, 2000) implies that

$$\Pr\left[\|\mathbf{g}_i\|_2^2 > p_T - 1 + 2\sqrt{(p_T - 1)t} + 2t\right] \leqslant e^{-t}, \quad \forall t > 0.$$

Then, for fixed and finite $p_T$ and $\|\boldsymbol{\mu}_T\|_2$, with a sufficiently large $n$, there exists a constant $a_n > 1/8$ such that applying the union bound with $t = a_n L_n^2$ yields that

$$\Pr\left[\max_{i\in[n]}\|\mathbf{w}(\widetilde{\mathbf{x}}_i)\|_2^2 > L_n^2\right] \leqslant n\exp\left(-a_n L_n^2\right) = o(n^{-1}).$$

Applying the union bound again, we get $\Pr[E_n^c] = o(n^{-1})$ for sufficiently large $n$. Meanwhile, conditioned on $E_n$, we have for any $i \in [n]$,

$$\left\|\mathbf{Y}_i^{(k,l)}\right\|_2 \leqslant \frac{1}{n}\|\mathbf{z}(\widetilde{\mathbf{x}}_i)\|_\infty^2\|\mathbf{w}(\widetilde{\mathbf{x}}_i)\|_2^2 \leqslant \frac{L_n^4}{n}, \quad \left\|\mathbf{R}_i^{(k,l)}\right\|_2 \leqslant \frac{1}{n}\|\mathbf{z}(\widetilde{\mathbf{x}}_i)\|_\infty^2\|\mathbf{C}_T(\eta_\ell)\|_2 \lesssim \frac{L_n^2}{n},$$

which implies that

$$\left\|\sum_{i=1}^n \mathbb{E}\left[(\mathbf{Y}_i^{(k,l)})^2 \mid E_n\right]\right\|_2 \leqslant \sum_{i=1}^n \mathbb{E}\left[\left\|\mathbf{Y}_i^{(k,l)}\right\|_2^2 \mid E_n\right] \leqslant n \cdot \frac{L_n^8}{n^2} = \frac{L_n^8}{n},$$

$$\left\|\sum_{i=1}^n \mathbb{E}\left[(\mathbf{R}_i^{(k,l)})^2 \mid E_n\right]\right\|_2 \leqslant \sum_{i=1}^n \mathbb{E}\left[\left\|\mathbf{R}_i^{(k,l)}\right\|_2^2 \mid E_n\right] \lesssim n \cdot \frac{L_n^4}{n^2} = \frac{L_n^4}{n}.$$

Then, applying the matrix Bernstein inequality (Tropp, 2012, Theorem 1.6) to $\sum_{i=1}^n \mathbf{Y}_i^{(k,l)}$ and $\sum_{i=1}^n \mathbf{R}_i^{(k,l)}$ and a union bound over all $k, l \in [d_z]$ off-diagonal blocks with $k \neq l$ yields that

$$\Pr\left[\max_{k\neq l}\left\|\widetilde{\boldsymbol{\Sigma}}_n^{(k,l)}\right\|_2 \gtrsim \sqrt{\frac{\log(n)}{n}}\right] \leqslant \Pr\left[E_n^c\right] + \Pr\left[\max_{k\neq l}\left\|\widetilde{\boldsymbol{\Sigma}}_n^{(k,l)}\right\|_2 \gtrsim \sqrt{\frac{\log(n)}{n}} \,\middle|\, E_n\right]$$

$$\leqslant o(n^{-1}) + 2p_T n^{-\Omega\left(\frac{n^2}{\log^4(n)}\right)} \cdot d_z^2 = o(1),$$

and therefore, the off-diagonal blocks are negligible:

$$\max_{k\neq l}\left\|\widetilde{\boldsymbol{\Sigma}}_n^{(k,l)}\right\|_2 = O_{\mathbb{P}}\left(\sqrt{\frac{\log(n)}{n}}\right).$$

**Diagonal blocks are concentrated.** Consider the $k$-th diagonal block $\widetilde{\boldsymbol{\Sigma}}_n^{(k,k)}$ for any fixed $k \in [d_z]$:

$$\widetilde{\boldsymbol{\Sigma}}_n^{(k,k)} = \frac{1}{n}\sum_{i=1}^n z_k(\widetilde{\mathbf{x}}_i)^2 \cdot \mathbf{w}(\widetilde{\mathbf{x}}_i)\mathbf{w}(\widetilde{\mathbf{x}}_i)^\top$$

$$= \frac{1}{n}\sum_{i=1}^n \mathbf{w}(\widetilde{\mathbf{x}}_i)\mathbf{w}(\widetilde{\mathbf{x}}_i)^\top + \frac{1}{n}\sum_{i=1}^n (z_k(\widetilde{\mathbf{x}}_i)^2 - 1) \cdot \mathbf{w}(\widetilde{\mathbf{x}}_i)\mathbf{w}(\widetilde{\mathbf{x}}_i)^\top,$$

where we denote $\widehat{\mathbf{C}}_{T,n} = \frac{1}{n}\sum_{i=1}^n \mathbf{w}(\widetilde{\mathbf{x}}_i)\mathbf{w}(\widetilde{\mathbf{x}}_i)^\top$. Let

$$s_k := \frac{1}{n}\sum_{i=1}^n z_k(\widetilde{\mathbf{x}}_i)^2. \tag{16}$$

Then,

$$\widetilde{\boldsymbol{\Sigma}}_n^{(k,k)} - s_k\mathbf{C}_T(\eta_\ell) = (\widehat{\mathbf{C}}_{T,n} - \mathbf{C}_T(\eta_\ell)) + \frac{1}{n}\sum_{i=1}^n (z_k(\widetilde{\mathbf{x}}_i)^2 - 1) \cdot (\mathbf{w}(\widetilde{\mathbf{x}}_i)\mathbf{w}(\widetilde{\mathbf{x}}_i)^\top - \mathbf{C}_T(\eta_\ell)),$$

where both terms are sums of independent random matrices with zero mean. Leveraging the same argument as for the off-diagonal blocks using the same event $E_n$, the matrix Bernstein inequality (Tropp, 2012, Theorem 1.6), and a union bound over all $k \in [d_z]$, we have for sufficiently large $n$,

$$\max_{k\in[d_z]}\left\|\widetilde{\boldsymbol{\Sigma}}_n^{(k,k)} - s_k\mathbf{C}_T(\eta_\ell)\right\|_2 = O_{\mathbb{P}}\left(\sqrt{\frac{\log(n)}{n}}\right). \tag{17}$$

Also, the $\chi^2$ concentration (Laurent & Massart, 2000) implies that for any fixed $\epsilon \in (0, 1/2)$, as $d_z, n \to \infty$ with $d_z/n \to \gamma_z \in (0, p_T^{-1})$,

$$
\begin{aligned}
\Pr\left[\min_{k \in [d_z]} s_k < 1 - \epsilon\right] &\leqslant d_z \Pr\left[s_k < 1 - \epsilon\right] \leqslant d_z \exp\left(-\Theta(n\epsilon^2)\right) = o(1), \\
\Pr\left[\max_{k \in [d_z]} s_k > 1 + \epsilon\right] &\leqslant d_z \Pr\left[s_k > 1 + \epsilon\right] \leqslant d_z \exp\left(-\Theta(n\epsilon^2)\right) = o(1),
\end{aligned}
\tag{18}
$$

so that all $s_k$'s are close to 1 with high probability.

**Concentration of $\widetilde{\boldsymbol{\Sigma}}_n^{-1}$.** Let $\mathbf{D}_n = \mathrm{diag}\left(s_1 \mathbf{C}_T(\eta_\ell), \cdots, s_{d_z} \mathbf{C}_T(\eta_\ell)\right)$ be the block-diagonal matrix with $k$-th diagonal block $s_k \mathbf{C}_T(\eta_\ell)$ for all $k \in [d_z]$; and $\mathbf{E}_n = \widetilde{\boldsymbol{\Sigma}}_n - \mathbf{D}_n$ be the fluctuations around $\mathbf{D}_n$. Since $\mathbf{C}_T(\eta_\ell)$ is positive definite, (18) implies that $\left\|\mathbf{D}_n^{-1}\right\|_2 < \infty$ with high probability for sufficiently large $n$. Then, the resolvent identity implies that

$$
\widetilde{\boldsymbol{\Sigma}}_n^{-1} = (\mathbf{D}_n + \mathbf{E}_n)^{-1} = \mathbf{D}_n^{-1} - \mathbf{D}_n^{-1} \mathbf{E}_n (\mathbf{D}_n + \mathbf{E}_n)^{-1}.
\tag{19}
$$

In particular, the block matrix inversion formula implies that for any $k \in [d_z]$, the $k$-th diagonal block of $\widetilde{\boldsymbol{\Sigma}}_n^{-1}$, denoted as $(\widetilde{\boldsymbol{\Sigma}}_n^{-1})^{(k,k)} \in \mathbb{R}^{p_T \times p_T}$ is concentrated around the $k$-th diagonal block of $\mathbf{D}_n^{-1}$, $(s_k \mathbf{C}_T(\eta_\ell))^{-1}$:

$$
\left\|(\widetilde{\boldsymbol{\Sigma}}_n^{-1})^{(k,k)} - (s_k \mathbf{C}_T(\eta_\ell))^{-1}\right\|_2 \lesssim \|\mathbf{E}_n\|_2 = O_{\mathbb{P}}\left(\sqrt{\frac{\log(n)}{n}}\right),
\tag{20}
$$

**Concentration of the trace.** Finally, notice that the trace of interest, $\frac{1}{n} \mathrm{tr}\left((\mathbf{I}_{d_z} \otimes \mathbf{C})\widetilde{\boldsymbol{\Sigma}}_n^{-1}\right)$, depends only on the diagonal blocks of $\widetilde{\boldsymbol{\Sigma}}_n^{-1}$. Then, (20) implies that

$$
\begin{aligned}
\frac{1}{n} \mathrm{tr}\left((\mathbf{I}_{d_z} \otimes \mathbf{C})\widetilde{\boldsymbol{\Sigma}}_n^{-1}\right) &= \frac{1}{n} \sum_{k=1}^{d_z} \mathrm{tr}\left(\mathbf{C}(\widetilde{\boldsymbol{\Sigma}}_n^{-1})^{(k,k)}\right) \\
&= \left(\frac{1}{n} \sum_{k=1}^{d_z} \frac{1}{s_k}\right) \mathrm{tr}\left(\mathbf{C}\mathbf{C}_T(\eta_\ell)^{-1}\right) + O_{\mathbb{P}}\left(\sqrt{\frac{\log(n)}{n}}\right) \\
&= \left(\frac{1}{d_z} \sum_{k=1}^{d_z} \frac{1}{s_k}\right) \cdot \frac{d_z}{n} \mathrm{tr}\left(\mathbf{C}\mathbf{C}_T(\eta_\ell)^{-1}\right) + o_{\mathbb{P}}(1).
\end{aligned}
$$

Since $\{ns_k\}_{k=1}^{d_z}$ are independent and $\chi_n^2$ distributed, for any fixed $n > 2$, $\mathbb{E}[s_k^{-1}] = \frac{n}{n-2}$. Then, the weak law of large numbers implies that as $d_z, n \to \infty$,

$$
\frac{1}{d_z} \sum_{k=1}^{d_z} \frac{1}{s_k} \xrightarrow{\mathbb{P}} \frac{n}{n-2} \xrightarrow[n \to \infty]{} 1.
$$

Putting everything together with $d_z/n \to \gamma_z \in (0, p_T^{-1})$ completes the proof. $\square$

## D.2 W2S FINE-TUNING OF STRONG STUDENT

**Theorem 3** (W2S fine-tuning of strong student (formal restatement of Theorem 2)). *Under Assumption 1, as $d_z, n, N \to \infty$ with $d_z/n \to \gamma_z \in (0, p_T^{-1})$ and $d_z/N \to \nu_z \in (0, p_S^{-1})$,*

$f_S(\mathbf{x}) = \varphi_S(\mathbf{x})^\top \boldsymbol{\theta}_S = \phi_S(\mathbf{x})^\top \boldsymbol{\beta}_S$ *from* (2) *satisfies*

$$\mathbb{E}_{\mathcal{S}_x \sim \mathcal{D}(\eta_u)^N, \widetilde{\mathcal{S}} \sim \mathcal{D}(\eta_\ell)^n} [\mathbf{ER}_{\eta_t}(f_S)] \xrightarrow{\mathbb{P}} \sigma_y^2 \gamma_z \Big( \boxed{p_{T \wedge S}} +$$

$$\underbrace{\boxed{\frac{\|(\eta_u - \eta_\ell)\boldsymbol{\mu}_T + (\eta_t - \eta_u)\boldsymbol{\Xi}\boldsymbol{\mu}_S\|_2^2}{\sigma_\xi^2}}}_{\mathcal{V}_S^{(1)} \leqslant \mathcal{V}_T^{(1)} \text{ when } \eta_u = \eta_\ell} +$$

$$\underbrace{\boxed{\nu_z(p_T - p_{T \wedge S})\left(p_S + (\eta_t - \eta_u)^2 \frac{\|\boldsymbol{\mu}_S\|_2^2}{\sigma_\xi^2}\right)}}_{\mathcal{E}_S = \Theta(\nu_z) \ll 1 \text{ negligible when } \nu_z \ll 1} \Big),$$

where the red annotation $\mathcal{V}_S^{(0)} \leqslant \mathcal{V}_T^{(0)}$ appears under $p_{T \wedge S}$.

*where* $p_{T \wedge S} = 1 + \|\boldsymbol{\Xi}\|_F^2 \in [1, p_S]$ *quantifies the effective dimension of group features learned by the strong student from the weak teacher.*

Notice that in Theorem 3, $\mathcal{V}_S^{(0)} + \mathcal{V}_S^{(1)}$ is dominant and will be small if $\mathbf{T}, \mathbf{S}$ are nearly orthogonal (*i.e.,* $\|\boldsymbol{\Xi}\|_2 \approx 0$) and $\eta_u \approx \eta_t$; whereas $\mathcal{E}_S$ tends to be much smaller than $\mathcal{V}_S^{(0)} + \mathcal{V}_S^{(1)}$, especially since unlabeled data is usually abundant compared to labeled data (*i.e.,* $\nu_z \ll \gamma_z$).

*Proof of Theorems 2 and 3.* We first introduce some helpful notions for the proof. Recall $\phi_S(\mathbf{x}) = \mathbf{z}(\mathbf{x}) \otimes \boldsymbol{\psi}(\mathbf{x})$. Let $\boldsymbol{\Sigma}_{\phi_S, \eta} = \mathbb{E}_{\mathcal{D}(\eta)}[\phi_S(\mathbf{x})\phi_S(\mathbf{x})^\top]$ for any $\eta \in [0, 1]$ and observe that

$$\boldsymbol{\Sigma}_{\phi_S, \eta} = \mathbf{I}_{d_z} \otimes \mathbf{C}_S(\eta), \quad \mathbf{C}_S(\eta) = \mathbb{E}_{\mathcal{D}(\eta)}[\boldsymbol{\psi}(\mathbf{x})\boldsymbol{\psi}(\mathbf{x})^\top] = \begin{bmatrix} 1 & \eta\boldsymbol{\mu}_S^\top \\ \eta\boldsymbol{\mu}_S & \sigma_\xi^2\mathbf{I}_{p_S-1} + \eta^2\boldsymbol{\mu}_S\boldsymbol{\mu}_S^\top \end{bmatrix}. \quad (21)$$

The block matrix inversion formula implies that

$$\mathbf{C}_S(\eta)^{-1} = \begin{bmatrix} 1 + \sigma_\xi^{-2}\eta^2\|\boldsymbol{\mu}_S\|_2^2 & -\sigma_\xi^{-2}\eta\boldsymbol{\mu}_S^\top \\ -\sigma_\xi^{-2}\eta\boldsymbol{\mu}_S & \sigma_\xi^{-2}\mathbf{I}_{p_S-1} \end{bmatrix}. \quad (22)$$

Meanwhile, the cross covariance of the student-teacher representations under $\mathcal{D}(\eta)$ is given by

$$\boldsymbol{\Sigma}_{\phi_S, \phi_T, \eta} = \mathbb{E}_{\mathcal{D}(\eta)}[\phi_S(\mathbf{x})\phi_T(\mathbf{x})^\top] = \mathbb{E}_{\mathcal{D}(\eta)}[(\mathbf{z}(\mathbf{x}) \otimes \boldsymbol{\psi}(\mathbf{x}))(\mathbf{z}(\mathbf{x}) \otimes \mathbf{w}(\mathbf{x}))^\top]$$
$$= \mathbb{E}_{\mathcal{D}(\eta)}[\mathbf{z}(\mathbf{x})\mathbf{z}(\mathbf{x})^\top] \otimes \mathbb{E}_{\mathcal{D}(\eta)}[\boldsymbol{\psi}(\mathbf{x})\mathbf{w}(\mathbf{x})^\top] = \mathbf{I}_{d_z} \otimes \mathbf{A}(\eta),$$

where

$$\mathbf{A}(\eta) = \mathbb{E}_{\mathcal{D}(\eta)}\left[\boldsymbol{\psi}(\mathbf{x})\mathbf{w}(\mathbf{x})^\top\right] = \mathbb{E}_{\mathcal{D}(\eta)}\left[\begin{bmatrix} 1 \\ \mathbf{S}^\top\boldsymbol{\xi}(\mathbf{x}) \end{bmatrix} \begin{bmatrix} 1 & \boldsymbol{\xi}(\mathbf{x})^\top\mathbf{T} \end{bmatrix}\right] \quad (23)$$

$$= \begin{bmatrix} 1 & \eta\boldsymbol{\mu}_T^\top \\ \eta\boldsymbol{\mu}_S & \sigma_\xi^2\mathbf{S}^\top\mathbf{T} + \eta^2\boldsymbol{\mu}_S\boldsymbol{\mu}_T^\top \end{bmatrix} \in \mathbb{R}^{p_S \times p_T}. \quad (24)$$

**Close-form solution and population-optimal predictor of W2S fine-tuning.** Given the equivalence between (2) and (6), we consider the latter throughout the proof. Adapting the notion from the proof of Theorem 1, given the labeled set $\widetilde{\mathcal{S}} = \{(\widetilde{\mathbf{x}}_i, \widetilde{y}_i) \mid i \in [n]\} \sim \mathcal{D}(\eta_\ell)^n$ and the unlabeled set $\mathcal{S} = \{(\mathbf{x}_i, y_i) \mid i \in [N]\} \sim \mathcal{D}_\mathbf{x}(\eta_u)^N$ with unknown $y_i$'s, we denote

$$\widetilde{\boldsymbol{\Phi}}_T = [\phi_T(\widetilde{\mathbf{x}}_1), \cdots, \phi_T(\widetilde{\mathbf{x}}_n)]^\top \in \mathbb{R}^{n \times d_T}, \quad \widetilde{\mathbf{y}} = [\widetilde{y}_1, \cdots, \widetilde{y}_n]^\top \in \mathbb{R}^n,$$
$$\boldsymbol{\Phi}_S = [\phi_S(\mathbf{x}_1), \cdots, \phi_S(\mathbf{x}_N)]^\top \in \mathbb{R}^{N \times d_S}, \quad \boldsymbol{\Phi}_T = [\phi_T(\mathbf{x}_1), \cdots, \phi_T(\mathbf{x}_N)]^\top \in \mathbb{R}^{N \times d_T}.$$

Then, since $n > d_T$ and $N > d_S$ by Assumption 1, (6) admits a unique closed-form solution

$$\boldsymbol{\beta}_S = \left(\boldsymbol{\Phi}_S^\top\boldsymbol{\Phi}_S\right)^{-1}\boldsymbol{\Phi}_S^\top\boldsymbol{\Phi}_T\boldsymbol{\beta}_T \quad \text{where} \quad \boldsymbol{\beta}_T = (\widetilde{\boldsymbol{\Phi}}_T^\top\widetilde{\boldsymbol{\Phi}}_T)^{-1}\widetilde{\boldsymbol{\Phi}}_T^\top\widetilde{\mathbf{y}}$$

from (11). Recall from Lemma 1 that the population-optimal linear predictor over $\phi_T$ in Lemma 1 is $f_T^\infty(\mathbf{x}) = \phi_T(\mathbf{x})^\top \boldsymbol{\beta}_T^\infty = f_*(\mathbf{x})$ with $\boldsymbol{\beta}_T^\infty = \boldsymbol{\beta}_* \otimes \mathbf{e}_1$. Conditioned on $f_T^\infty(\mathbf{x})$, the population-optimal linear predictor over $\phi_S$ is given by

$$
\begin{aligned}
\boldsymbol{\beta}_S^\infty &= \mathbb{E}_{\mathcal{D}(\eta_u)} \left[ \phi_S(\mathbf{x}) \phi_S(\mathbf{x})^\top \right]^{-1} \mathbb{E}_{\mathcal{D}(\eta_u)} \left[ \phi_S(\mathbf{x}) \phi_T(\mathbf{x})^\top \right] \boldsymbol{\beta}_T^\infty \\
&= (\mathbf{I}_{d_z} \otimes \mathbf{C}_S(\eta_u))^{-1} (\mathbf{I}_{d_z} \otimes \mathbf{A}(\eta_u)) \boldsymbol{\beta}_T^\infty \\
&= (\mathbf{I}_{d_z} \otimes (\mathbf{C}_S(\eta_u)^{-1} \mathbf{A}(\eta_u)))(\boldsymbol{\beta}_* \otimes \mathbf{e}_1) \\
&= \boldsymbol{\beta}_* \otimes (\mathbf{C}_S(\eta_u)^{-1} \mathbf{A}(\eta_u) \mathbf{e}_1) = \boldsymbol{\beta}_* \otimes \mathbf{e}_1,
\end{aligned}
$$

which implies that $f_S^\infty(\mathbf{x}) = \phi_S(\mathbf{x})^\top \boldsymbol{\beta}_S^\infty = f_*(\mathbf{x})$, *i.e.*, a strong student W2S fine-tuned with pseudolabels from the Bayes-optimal weak teacher over the population is also Bayes-optimal. Therefore, the student estimator in (6) differs from $\boldsymbol{\beta}_S^\infty$ by

$$
\begin{aligned}
\boldsymbol{\beta}_S - \boldsymbol{\beta}_S^\infty &= \left( \boldsymbol{\Phi}_S^\top \boldsymbol{\Phi}_S \right)^{-1} \boldsymbol{\Phi}_S^\top \boldsymbol{\Phi}_T (\boldsymbol{\beta}_T - \boldsymbol{\beta}_T^\infty) \\
&= \left( \boldsymbol{\Phi}_S^\top \boldsymbol{\Phi}_S \right)^{-1} \boldsymbol{\Phi}_S^\top \boldsymbol{\Phi}_T (\widetilde{\boldsymbol{\Phi}}_T^\top \widetilde{\boldsymbol{\Phi}}_T)^{-1} \widetilde{\boldsymbol{\Phi}}_T^\top \widetilde{\boldsymbol{\epsilon}},
\end{aligned}
$$

and the estimation error of W2S fine-tuning is given by

$$
\mathbf{ER}_{\eta_t}(f_S) = \mathbb{E}_{\mathcal{D}(\eta_t)}[(f_S(\mathbf{x}) - f_*(\mathbf{x}))^2] = \mathbb{E}_{\mathcal{D}(\eta_t)}[(\phi_S(\mathbf{x})^\top (\boldsymbol{\beta}_S - \boldsymbol{\beta}_S^\infty))^2] = \| \boldsymbol{\beta}_S - \boldsymbol{\beta}_S^\infty \|_{\boldsymbol{\Sigma}_{\phi_S, \eta_t}}^2.
$$

Then, conditioned on $\widetilde{\boldsymbol{\Phi}}_T$ and $\boldsymbol{\Phi}_S, \boldsymbol{\Phi}_T$, the excess risk can be expressed as

$$
\begin{aligned}
&\mathbb{E}_{\widetilde{\boldsymbol{\epsilon}}} \left[ \mathbf{ER}_{\eta_t}(f_S) \mid \widetilde{\boldsymbol{\Phi}}_T, \boldsymbol{\Phi}_S, \boldsymbol{\Phi}_T \right] \\
&= \mathbb{E}_{\widetilde{\boldsymbol{\epsilon}}} \left[ \left\| \left( \boldsymbol{\Phi}_S^\top \boldsymbol{\Phi}_S \right)^{-1} \boldsymbol{\Phi}_S^\top \boldsymbol{\Phi}_T (\widetilde{\boldsymbol{\Phi}}_T^\top \widetilde{\boldsymbol{\Phi}}_T)^{-1} \widetilde{\boldsymbol{\Phi}}_T^\top \widetilde{\boldsymbol{\epsilon}} \right\|_{\boldsymbol{\Sigma}_{\phi_S, \eta_t}}^2 \mid \widetilde{\boldsymbol{\Phi}}_T, \boldsymbol{\Phi}_S, \boldsymbol{\Phi}_T \right] \qquad (25) \\
&= \sigma_y^2 \ \mathrm{tr} \left( \boldsymbol{\Sigma}_{\phi_S, \eta_t} \left( \boldsymbol{\Phi}_S^\top \boldsymbol{\Phi}_S \right)^{-1} \boldsymbol{\Phi}_S^\top \boldsymbol{\Phi}_T (\widetilde{\boldsymbol{\Phi}}_T^\top \widetilde{\boldsymbol{\Phi}}_T)^{-1} \boldsymbol{\Phi}_T^\top \boldsymbol{\Phi}_S \left( \boldsymbol{\Phi}_S^\top \boldsymbol{\Phi}_S \right)^{-1} \right).
\end{aligned}
$$

**Concentration of sample covariance matrices.** Define the sample (cross) covariance matrices

$$
\widehat{\boldsymbol{\Sigma}}_{S,N} = \frac{1}{N} \boldsymbol{\Phi}_S^\top \boldsymbol{\Phi}_S, \quad \widehat{\boldsymbol{\Sigma}}_{S,T,N} = \frac{1}{N} \boldsymbol{\Phi}_S^\top \boldsymbol{\Phi}_T, \quad \widetilde{\boldsymbol{\Sigma}}_{T,n} = \frac{1}{n} \widetilde{\boldsymbol{\Phi}}_T^\top \widetilde{\boldsymbol{\Phi}}_T.
$$

Then, taking the expectation of (25) over $\widetilde{\mathcal{S}}$ and $\mathcal{S}_x$ yields

$$
\mathbb{E}_{\widetilde{\mathcal{S}}, \mathcal{S}_x}[\mathbf{ER}_{\eta_t}(f_S)] = \frac{\sigma_y^2}{n} \mathrm{tr} \left( \mathbb{E}_{\mathcal{S}_x} \left[ \widehat{\boldsymbol{\Sigma}}_{S,T,N}^\top \widehat{\boldsymbol{\Sigma}}_{S,N}^{-1} \boldsymbol{\Sigma}_{\phi_S, \eta_t} \widehat{\boldsymbol{\Sigma}}_{S,N}^{-1} \widehat{\boldsymbol{\Sigma}}_{S,T,N} \right] \mathbb{E}_{\widetilde{\mathcal{S}}} \left[ \widetilde{\boldsymbol{\Sigma}}_{T,n}^{-1} \right] \right). \qquad (26)
$$

At the proportional asymptotic limit, Lemma 3 below shows that

$$
\begin{aligned}
&\frac{1}{n} \mathrm{tr} \left( \mathbb{E}_{\mathcal{S}_x} \left[ \widehat{\boldsymbol{\Sigma}}_{S,T,N}^\top \widehat{\boldsymbol{\Sigma}}_{S,N}^{-1} \boldsymbol{\Sigma}_{\phi_S, \eta_t} \widehat{\boldsymbol{\Sigma}}_{S,N}^{-1} \widehat{\boldsymbol{\Sigma}}_{S,T,N} \right] \mathbb{E}_{\widetilde{\mathcal{S}}} \left[ \widetilde{\boldsymbol{\Sigma}}_{T,n}^{-1} \right] \right) \\
&\xrightarrow{\mathbb{P}} \gamma_z \, \mathrm{tr} \left( \mathbf{C}_{T,S}(\eta_t, \eta_u) \mathbf{C}_T(\eta_\ell)^{-1} \right) + \gamma_z \nu_z \, (p_T - p_{T \wedge S}) \, \mathrm{tr} \left( \mathbf{C}_S(\eta_t) \mathbf{C}_S(\eta_u)^{-1} \right),
\end{aligned}
$$

Leveraging (21), (22), and (23), we have

$$
\begin{aligned}
\mathrm{tr} \left( \mathbf{C}_{T,S}(\eta_t, \eta_u) \mathbf{C}_T(\eta_\ell)^{-1} \right) &= \mathrm{tr} \left( \mathbf{A}(\eta_u)^\top \mathbf{C}_S(\eta_u)^{-1} \mathbf{C}_S(\eta_t) \mathbf{C}_S(\eta_u)^{-1} \mathbf{A}(\eta_u) \mathbf{C}_T(\eta_\ell)^{-1} \right) \\
&= 1 + \| \boldsymbol{\Xi} \|_F^2 + \frac{\| (\eta_u - \eta_\ell) \boldsymbol{\mu}_T + (\eta_t - \eta_u) \boldsymbol{\Xi} \boldsymbol{\mu}_S \|_2^2}{\sigma_\xi^2} \\
&= p_{T \wedge S} + \frac{\| (\eta_u - \eta_\ell) \boldsymbol{\mu}_T + (\eta_t - \eta_u) \boldsymbol{\Xi} \boldsymbol{\mu}_S \|_2^2}{\sigma_\xi^2},
\end{aligned}
$$

while an analogous derivation as in (15) implies that

$$
\mathrm{tr} \left( \mathbf{C}_S(\eta_t) \mathbf{C}_S(\eta_u)^{-1} \right) = p_S + (\eta_t - \eta_u)^2 \frac{\| \boldsymbol{\mu}_S \|_2^2}{\sigma_\xi^2}.
$$

Overall, plugging everything back to (26) yields

$$
\mathbb{E}_{\widehat{\mathcal{S}}, \mathcal{S}_x} \left[ \mathbf{ER}_{\eta_t}(f_S) \right] \overset{\mathbb{P}}{\to} \sigma_y^2 \gamma_z \left( p_{T \wedge S} + \frac{\|(\eta_u - \eta_\ell)\boldsymbol{\mu}_T + (\eta_t - \eta_u)\boldsymbol{\Xi}\boldsymbol{\mu}_S\|_2^2}{\sigma_\xi^2} \right)
$$
$$
+ \sigma_y^2 \gamma_z \nu_z (p_T - p_{T \wedge S}) \left( p_S + (\eta_t - \eta_u)^2 \frac{\|\boldsymbol{\mu}_S\|_2^2}{\sigma_\xi^2} \right),
$$

$\square$

**Lemma 3.** *In the proof of Theorem 2, at the proportional asymptotic limit,*

$$
\frac{1}{n} \operatorname{tr} \left( \mathbb{E}_{\mathcal{S}_x} \left[ \widehat{\boldsymbol{\Sigma}}_{S,T,N}^\top \widehat{\boldsymbol{\Sigma}}_{S,N}^{-1} \boldsymbol{\Sigma}_{\phi_S, \eta_t} \widehat{\boldsymbol{\Sigma}}_{S,N}^{-1} \widehat{\boldsymbol{\Sigma}}_{S,T,N} \right] \mathbb{E}_{\widehat{\mathcal{S}}} \left[ \widetilde{\boldsymbol{\Sigma}}_{T,n}^{-1} \right] \right)
$$
$$
\overset{\mathbb{P}}{\to} \gamma_z \operatorname{tr} \left( \mathbf{C}_{T,S}(\eta_t, \eta_u) \mathbf{C}_T(\eta_\ell)^{-1} \right) + \gamma_z \nu_z (p_T - p_{T \wedge S}) \operatorname{tr} \left( \mathbf{C}_S(\eta_t) \mathbf{C}_S(\eta_u)^{-1} \right),
$$

*where* $\mathbf{C}_{T,S}(\eta_t, \eta_u) \in \mathbb{R}^{p_T \times p_T}$ *is defined as*

$$
\mathbf{C}_{T,S}(\eta_t, \eta_u) = \mathbf{A}(\eta_u)^\top \mathbf{C}_S(\eta_u)^{-1} \mathbf{C}_S(\eta_t) \mathbf{C}_S(\eta_u)^{-1} \mathbf{A}(\eta_u),
$$

*Proof of Lemma 3.* The proof mostly follows the same argument as in Lemma 2, with the key difference being a careful treatment of the off-diagonal blocks in the sample (cross) covariance matrices $\widehat{\boldsymbol{\Sigma}}_{S,N}$ and $\widehat{\boldsymbol{\Sigma}}_{S,T,N}$, which are still small but with an additional non-negligible higher-order moment in the proportional asymptotic limit.

Following the proof of Lemma 2, "Separation of block diagonals and off-diagonals", we first partition $\widehat{\boldsymbol{\Sigma}}_{S,N}$ and $\widehat{\boldsymbol{\Sigma}}_{S,T,N}$ into $d_z \times d_z$ blocks:

$$
\widehat{\boldsymbol{\Sigma}}_{S,N} = \left[ \widehat{\boldsymbol{\Sigma}}_{S,N}^{(k,l)} \right]_{k,l=1}^{d_z}, \quad \widehat{\boldsymbol{\Sigma}}_{S,T,N} = \left[ \widehat{\boldsymbol{\Sigma}}_{S,T,N}^{(k,l)} \right]_{k,l=1}^{d_z},
$$

where $\widehat{\boldsymbol{\Sigma}}_{S,N}^{(k,l)} \in \mathbb{R}^{p_S \times p_S}$ and $\widehat{\boldsymbol{\Sigma}}_{S,T,N}^{(k,l)} \in \mathbb{R}^{p_S \times p_T}$ are given by

$$
\widehat{\boldsymbol{\Sigma}}_{S,N}^{(k,l)} = \frac{1}{N} \sum_{i=1}^N z_k(\mathbf{x}_i) z_l(\mathbf{x}_i) \cdot \boldsymbol{\psi}(\mathbf{x}_i)\boldsymbol{\psi}(\mathbf{x}_i)^\top,
$$
$$
\widehat{\boldsymbol{\Sigma}}_{S,T,N}^{(k,l)} = \frac{1}{N} \sum_{i=1}^N z_k(\mathbf{x}_i) z_l(\mathbf{x}_i) \cdot \boldsymbol{\psi}(\mathbf{x}_i)\mathbf{w}(\mathbf{x}_i)^\top.
$$

We denote

$$
s_{kl} = \frac{1}{N} \sum_{i=1}^N z_k(\mathbf{x}_i) z_l(\mathbf{x}_i) \quad \text{for all } k, l \in [d_z],
$$

and observe that for $k \neq l$, $\mathbb{E}[s_{kl}] = 0$, and for $k = l$, $\mathbb{E}[s_{kk}] = 1$. We therefore observe and denote that

$$
\mathbf{D}_S := \mathbb{E}_{\mathcal{D}(\eta_u)} \left[ \widehat{\boldsymbol{\Sigma}}_{S,N} \right] = \mathbf{I}_{d_z} \otimes \mathbf{C}_S(\eta_u), \quad \mathbf{D}_{S,T} := \mathbb{E}_{\mathcal{D}(\eta_u)} \left[ \widehat{\boldsymbol{\Sigma}}_{S,T,N} \right] = \mathbf{I}_{d_z} \otimes \mathbf{A}(\eta_u). \quad (27)
$$

We further define the reminder fluctuation matrices around $\mathbf{D}_S$ and $\mathbf{D}_{S,T}$:

$$
\mathbf{E}_S = \widehat{\boldsymbol{\Sigma}}_{S,N} - \mathbf{D}_S, \quad \mathbf{E}_{S,T} = \widehat{\boldsymbol{\Sigma}}_{S,T,N} - \mathbf{D}_{S,T}, \quad (28)
$$

where

$$
\mathbf{E}_S = \left[ \mathbf{E}_S^{(k,l)} \right]_{k,l=1}^{d_z} = \begin{bmatrix} \widehat{\boldsymbol{\Sigma}}_{S,N}^{(1,1)} - \mathbf{C}_S(\eta_u) & \widehat{\boldsymbol{\Sigma}}_{S,N}^{(1,2)} & \cdots & \widehat{\boldsymbol{\Sigma}}_{S,N}^{(1,d_z)} \\ \widehat{\boldsymbol{\Sigma}}_{S,N}^{(2,1)} & \widehat{\boldsymbol{\Sigma}}_{S,N}^{(2,2)} - \mathbf{C}_S(\eta_u) & \cdots & \widehat{\boldsymbol{\Sigma}}_{S,N}^{(2,d_z)} \\ \vdots & \vdots & \ddots & \vdots \\ \widehat{\boldsymbol{\Sigma}}_{S,N}^{(d_z,1)} & \widehat{\boldsymbol{\Sigma}}_{S,N}^{(d_z,2)} & \cdots & \widehat{\boldsymbol{\Sigma}}_{S,N}^{(d_z,d_z)} - \mathbf{C}_S(\eta_u) \end{bmatrix},
$$

$$
\mathbf{E}_{S,T} = \left[ \mathbf{E}_{S,T}^{(k,l)} \right]_{k,l=1}^{d_z} = \begin{bmatrix} \widehat{\boldsymbol{\Sigma}}_{S,T,N}^{(1,1)} - \mathbf{A}(\eta_u) & \widehat{\boldsymbol{\Sigma}}_{S,T,N}^{(1,2)} & \cdots & \widehat{\boldsymbol{\Sigma}}_{S,T,N}^{(1,d_z)} \\ \widehat{\boldsymbol{\Sigma}}_{S,T,N}^{(2,1)} & \widehat{\boldsymbol{\Sigma}}_{S,T,N}^{(2,2)} - \mathbf{A}(\eta_u) & \cdots & \widehat{\boldsymbol{\Sigma}}_{S,T,N}^{(2,d_z)} \\ \vdots & \vdots & \ddots & \vdots \\ \widehat{\boldsymbol{\Sigma}}_{S,T,N}^{(d_z,1)} & \widehat{\boldsymbol{\Sigma}}_{S,T,N}^{(d_z,2)} & \cdots & \widehat{\boldsymbol{\Sigma}}_{S,T,N}^{(d_z,d_z)} - \mathbf{A}(\eta_u) \end{bmatrix}.
$$

$$(29)$$

Again, following the proof of Lemma 2, the resolvent identity implies that

$$\widehat{\boldsymbol{\Sigma}}_{S,N}^{-1} = (\mathbf{D}_S + \mathbf{E}_S)^{-1} = \mathbf{D}_S^{-1} - \mathbf{D}_S^{-1}\mathbf{E}_S(\mathbf{D}_S + \mathbf{E}_S)^{-1}$$
$$= \mathbf{D}_S^{-1} - \mathbf{D}_S^{-1}\mathbf{E}_S\mathbf{D}_S^{-1} + \mathbf{D}_S^{-1}\mathbf{E}_S\mathbf{D}_S^{-1}\mathbf{E}_S(\mathbf{D}_S + \mathbf{E}_S)^{-1}. \tag{30}$$

Then, since $\mathbb{E}[\mathbf{E}_S] = \mathbf{0}_{d_S \times d_S}$ and $\mathbb{E}[\mathbf{E}_{S,T}] = \mathbf{0}_{d_S \times d_T}$, we have

$$\mathbb{E}_{\mathcal{S}_x}\left[\widehat{\boldsymbol{\Sigma}}_{S,T,N}^{\top}\widehat{\boldsymbol{\Sigma}}_{S,N}^{-1}\boldsymbol{\Sigma}_{\phi_S,\eta_t}\widehat{\boldsymbol{\Sigma}}_{S,N}^{-1}\widehat{\boldsymbol{\Sigma}}_{S,T,N}\right]$$
$$= \mathbb{E}\left[(\mathbf{D}_{S,T} + \mathbf{E}_{S,T})^{\top}(\mathbf{D}_S + \mathbf{E}_S)^{-1}(\mathbf{I}_{d_z} \otimes \mathbf{C}_S(\eta_t))(\mathbf{D}_S + \mathbf{E}_S)^{-1}(\mathbf{D}_{S,T} + \mathbf{E}_{S,T})\right]$$
$$= \mathbf{D}_{S,T}^{\top}\mathbf{D}_S^{-1}(\mathbf{I}_{d_z} \otimes \mathbf{C}_S(\eta_t))\mathbf{D}_S^{-1}\mathbf{D}_{S,T} + \mathbf{R}_N$$
$$\qquad + \mathbb{E}\left[\mathbf{E}_{S,T}^{\top}\mathbf{D}_S^{-1}(\mathbf{I}_{d_z} \otimes \mathbf{C}_S(\eta_t))\mathbf{D}_S^{-1}\mathbf{E}_{S,T}\right] \quad (=: \mathbf{R}_{S,T})$$
$$\qquad + \mathbb{E}\left[\mathbf{D}_{S,T}^{\top}\mathbf{D}_S^{-1}\mathbf{E}_S\mathbf{D}_S^{-1}(\mathbf{I}_{d_z} \otimes \mathbf{C}_S(\eta_t))\mathbf{D}_S^{-1}\mathbf{E}_S\mathbf{D}_S^{-1}\mathbf{D}_{S,T}\right] \quad (=: \mathbf{R}_{S,S}) \tag{31}$$
$$\qquad - \mathbb{E}\left[\mathbf{D}_{S,T}^{\top}\mathbf{D}_S^{-1}\mathbf{E}_S\mathbf{D}_S^{-1}(\mathbf{I}_{d_z} \otimes \mathbf{C}_S(\eta_t))\mathbf{D}_S^{-1}\mathbf{E}_{S,T}\right] \quad (=: \mathbf{R}_{S,S,T})$$
$$\qquad - \mathbb{E}\left[\mathbf{E}_{S,T}^{\top}\mathbf{D}_S^{-1}(\mathbf{I}_{d_z} \otimes \mathbf{C}_S(\eta_t))\mathbf{D}_S^{-1}\mathbf{E}_S\mathbf{D}_S^{-1}\mathbf{D}_{S,T}\right], \quad (=: \mathbf{R}_{S,T,S})$$

where $\|\mathbf{R}_N\|_2 = o_{\mathbb{P}}(1)$ for sufficiently large $N$; $\mathbf{E}_S$ and $\mathbf{E}_{S,T}$ are averages over $N$ i.i.d. random matrices with $d_z \times d_z$ independent blocks. Therefore, when taking expectation for the second moments of $\mathbf{E}_S$ and $\mathbf{E}_{S,T}$, the off-diagonal blocks in $\mathbf{R}_{S,T}, \mathbf{R}_{S,S}, \mathbf{R}_{S,S,T}, \mathbf{R}_{S,T,S} \in \mathbb{R}^{d_T \times d_T}$ vanish due to independence, and only the diagonal blocks remain, which are i.i.d. across $k \in [d_z]$. Notice that (27) implies that

$$\mathbf{D}_{S,T}^{\top}\mathbf{D}_S^{-1}(\mathbf{I}_{d_z} \otimes \mathbf{C}_S(\eta_t))\mathbf{D}_S^{-1}\mathbf{D}_{S,T} = \mathbf{I}_{d_z} \otimes \mathbf{C}_{T,S}(\eta_t, \eta_u).$$

Also, we recall that the fourth moment of any Gaussian random vector $\mathbf{g} \sim \mathcal{N}(\mathbf{0}, \mathbf{I}_d)$ satisfies for any fixed matrix $\mathbf{M} \in \mathbb{R}^{d \times d}$,

$$\mathbb{E}\left[\left(\mathbf{g}\mathbf{g}^{\top}\right)^2\right] = (d+2)\mathbf{I}_d, \quad \mathbb{E}\left[\left(\mathbf{g}\mathbf{g}^{\top}\right)\mathbf{M}\left(\mathbf{g}\mathbf{g}^{\top}\right)\right] = \mathrm{tr}(\mathbf{M})\mathbf{I}_d + \mathbf{M} + \mathbf{M}^{\top}. \tag{32}$$

Define a function $g : \mathbb{R}^{d_T \times d_T} \to \mathbb{R}$ as

$$g(\mathbf{A}) = \frac{1}{n} \mathrm{tr}\left(\mathbf{A}\mathbb{E}_{\widetilde{\mathcal{S}}}\left[\widetilde{\boldsymbol{\Sigma}}_{T,n}^{-1}\right]\right).$$

Then, we have

$$\frac{1}{n}\mathrm{tr}\left(\mathbb{E}_{\mathcal{S}_x}\left[\widehat{\boldsymbol{\Sigma}}_{S,T,N}^{\top}\widehat{\boldsymbol{\Sigma}}_{S,N}^{-1}\boldsymbol{\Sigma}_{\phi_S,\eta_t}\widehat{\boldsymbol{\Sigma}}_{S,N}^{-1}\widehat{\boldsymbol{\Sigma}}_{S,T,N}\right]\mathbb{E}_{\widetilde{\mathcal{S}}}\left[\widetilde{\boldsymbol{\Sigma}}_{T,n}^{-1}\right]\right)$$
$$= g\left(\mathbf{I}_{d_z} \otimes \mathbf{C}_{T,S}(\eta_t, \eta_u)\right) + g(\mathbf{R}_{S,T}) + g(\mathbf{R}_{S,S}) - g(\mathbf{R}_{S,S,T}) - g(\mathbf{R}_{S,T,S}) + o_{\mathbb{P}}(1),$$

where given the $\mathbf{I}_{d_z} \otimes \mathbf{C}_{T,S}(\eta_t, \eta_u)$ structure, Lemma 2 then implies that under the proportional asymptotic limit,

$$g\left(\mathbf{I}_{d_z} \otimes \mathbf{C}_{T,S}(\eta_t, \eta_u)\right) \xrightarrow{\mathbb{P}} \gamma_z \mathrm{tr}\left(\mathbf{C}_{T,S}(\eta_t, \eta_u)\mathbf{C}_T(\eta_\ell)^{-1}\right).$$

Let $\mathbf{M} := \mathbf{C}_S(\eta_t)\mathbf{C}_S(\eta_u)^{-1} \in \mathbb{R}^{p_S \times p_S}$ and $\mathbf{M}' := [\mathbf{M}]_{2:p_S, 2:p_S}$. Recall from (31) that $\mathbf{R}_{S,T} = \mathbb{E}\left[\mathbf{E}_{S,T}^{\top}\mathbf{D}_S^{-1}(\mathbf{I}_{d_z} \otimes \mathbf{C}_S(\eta_t))\mathbf{D}_S^{-1}\mathbf{E}_{S,T}\right]$, (32) and (15), along with the proof of Lemma 2 imply that

$$g(\mathbf{R}_{S,T}) \xrightarrow{\mathbb{P}} \gamma_z \nu_z\left(p_T \mathrm{tr}(\mathbf{M}) + \frac{2}{\sigma_\xi^2}\mathrm{tr}\left(\mathbf{M}'\boldsymbol{\Xi}^{\top}\boldsymbol{\Xi}\right) + C_S\frac{(\eta_u - \eta_\ell)^2\|\boldsymbol{\mu}_T\|_2^2}{\sigma_\xi^2}\right),$$

for some constant $C_S > 0$ independent of $d_z, N$. Analogously, we have

$$g(\mathbf{R}_{S,S}) \xrightarrow{\mathbb{P}} \gamma_z \nu_z\left(p_{T \wedge S}\mathrm{tr}(\mathbf{M}) + \frac{2}{\sigma_\xi^2}\mathrm{tr}\left(\mathbf{M}'\boldsymbol{\Xi}^{\top}\boldsymbol{\Xi}\right) + C_S\frac{(\eta_u - \eta_\ell)^2\|\boldsymbol{\mu}_T\|_2^2}{\sigma_\xi^2}\right),$$

$$g((\mathbf{R}_{S,S,T})) \xrightarrow{\mathbb{P}} \gamma_z \nu_z\left(p_{T \wedge S}\mathrm{tr}(\mathbf{M}) + \frac{2}{\sigma_\xi^2}\mathrm{tr}\left(\mathbf{M}'\boldsymbol{\Xi}^{\top}\boldsymbol{\Xi}\right) + C_S\frac{(\eta_u - \eta_\ell)^2\|\boldsymbol{\mu}_T\|_2^2}{\sigma_\xi^2}\right)$$

$$g((\mathbf{R}_{S,T,S})) = g((\mathbf{R}_{S,S,T})).$$

Overall, at the proportional asymptotic limit,

$$\frac{1}{n} \operatorname{tr} \left( \mathbb{E}_{\mathcal{S}_x} \left[ \widehat{\boldsymbol{\Sigma}}_{S,T,N}^{\top} \widehat{\boldsymbol{\Sigma}}_{S,N}^{-1} \boldsymbol{\Sigma}_{\phi_S, \eta_t} \widehat{\boldsymbol{\Sigma}}_{S,N}^{-1} \widehat{\boldsymbol{\Sigma}}_{S,T,N} \right] \mathbb{E}_{\widetilde{\mathcal{S}}} \left[ \widetilde{\boldsymbol{\Sigma}}_{T,n}^{-1} \right] \right)$$

$$\xrightarrow{\mathbb{P}} \gamma_z \operatorname{tr} \left( \mathbf{C}_{T,S}(\eta_t, \eta_u) \mathbf{C}_T(\eta_\ell)^{-1} \right) + \gamma_z \nu_z \left( p_T - p_{T \wedge S} \right) \operatorname{tr} \left( \mathbf{C}_S(\eta_t) \mathbf{C}_S(\eta_u)^{-1} \right).$$

$\square$

# E  ADDITIONAL EXPERIMENTAL DETAILS

## E.1  DATASET STATISTICS

In this work, we construct three distinct splits for each of the four datasets, Waterbirds (Sagawa et al., 2020), BFFHQ (Lee et al., 2021), ImageNet-9 (Xiao et al., 2020), and BG-COCO. Specifically, each dataset is partitioned into a group-imbalanced training set $\mathcal{D}_1$, a group-balanced training set $\mathcal{D}_2$, and a group-balanced test set $\mathcal{D}_3$. The minority group proportion in $\mathcal{D}_1$, $\mathcal{D}_2$, and $\mathcal{D}_3$ is $\eta_o$, 0.5, and 0.5, respectively. Across different real-world experiments in the paper, we vary the group proportions ($\eta_\ell$ and $\eta_u$) as well as the sample sizes ($n$ and $N$). We first summarize the dataset statistics for each benchmark, and then describe how $\mathcal{D}_1$ to $\mathcal{D}_3$ are utilized in different experimental setups.

**Waterbirds statistics.**   The Waterbirds (Sagawa et al., 2020) dataset is designed to capture spurious correlations between natural backgrounds and bird labels, with $\eta_o = 0.05$. Table 3 reports the detailed group distributions across $\mathcal{D}_1$ to $\mathcal{D}_3$. Following Sagawa et al. (2020), we supplement additional samples for the minority groups (waterbird, land) and (landbird, water) in the same manner as the original dataset, due to the limited size of the raw data.

| Split | (waterbird, water) | (waterbird, land) | (landbird, water) | (landbird, land) | Total |
|-------|-------------------|-------------------|-------------------|------------------|-------|
| $\mathcal{D}_1$ | 1,057 | 56 | 184 | 3,498 | 4,795 |
| $\mathcal{D}_2$ | 1,804 | 1,804 | 1,804 | 1,804 | 7,216 |
| $\mathcal{D}_3$ | 451 | 451 | 451 | 451 | 1,804 |

Table 3: Dataset statistics for Waterbirds. Each column corresponds to a group, and the last column gives the total sample count.

**BFFHQ statistics.**   The BFFHQ (Lee et al., 2021) dataset is designed to capture spurious correlations between age and gender labels, with $\eta_o = 0.005$. Table 4 reports the detailed group distributions across $\mathcal{D}_1$ to $\mathcal{D}_3$. Due to the limited size of the minority groups in the raw data, our splits are constructed from de-duplicated samples across multiple BFFHQ subsets.

| Split | (young, female) | (young, male) | (old, female) | (old, male) | Total |
|-------|-----------------|---------------|---------------|-------------|-------|
| $\mathcal{D}_1$ | 9,552 | 48 | 48 | 9,552 | 19,200 |
| $\mathcal{D}_2$ | 790 | 790 | 790 | 790 | 3,160 |
| $\mathcal{D}_3$ | 198 | 198 | 198 | 198 | 792 |

Table 4: Dataset statistics for BFFHQ. Each column corresponds to a group, and the last column gives the total sample count.

**ImageNet-9 statistics.**   The ImageNet-9 (Xiao et al., 2020) dataset is designed to capture spurious correlations between object and background labels. Different from Waterbirds and BFFHQ, ImageNet-9 is a 9-class classification task over categories dog, bird, wheeled vehicle, reptile, carnivore, insect, musical instrument, primate, and fish. The original dataset provides two variants, mixed-same and mixed-rand. In the mixed-same version, each image background is replaced with a background from an image of the same class, thus preserving spurious correlations; in the mixed-rand version, the background is randomized and contains no information about the true label. These two variants correspond to minority group proportions of 0 and 0.5, respectively. Table 5 reports the dataset statistics across $\mathcal{D}_1$ to $\mathcal{D}_3$. Based on this table, we set $\eta_o = 0$. Note that ImageNet-9 does not have a well-defined group structure under either the mixed-same or mixed-rand settings. Therefore, we do not report worst-group accuracy for this dataset.

| Split | mixed-same | mixed-rand | Total | Per-class |
|-------|-----------|-----------|-------|-----------|
| $\mathcal{D}_1$ | 4,050 | 0 | 4,050 | 450 |
| $\mathcal{D}_2$ | 0 | 3,240 | 3,240 | 360 |
| $\mathcal{D}_3$ | 0 | 810 | 810 | 90 |

Table 5: Dataset statistics for ImageNet-9. Within each split, the nine classes have identical counts.

**BG-COCO statistics.** The BG-COCO dataset is a self-generated benchmark designed to capture spurious correlations between cats/dogs from COCO (Lin et al., 2014) and indoor/outdoor scenes from Places (Zhou et al., 2017). Specifically, we define the indoor/outdoor split as living room, dining room (indoor) and park (outdoor). By construction, cats are aligned with indoor scenes and dogs with outdoor scenes. Table 6 reports the detailed group distributions across $\mathcal{D}_1$ to $\mathcal{D}_3$. Based on this table, we set $\eta_o = 0.05$.

| Split | (cat, indoor) | (cat, outdoor) | (dog, indoor) | (dog, outdoor) | Total |
|-------|--------------|---------------|--------------|---------------|-------|
| $\mathcal{D}_1$ | 1,900 | 100 | 100 | 1,900 | 4,000 |
| $\mathcal{D}_2$ | 1,000 | 1,000 | 1,000 | 1,000 | 4,000 |
| $\mathcal{D}_3$ | 250 | 250 | 250 | 250 | 1,000 |

Table 6: Dataset statistics for BG-COCO. Each column corresponds to a group, and the last column gives the total sample count.

Across all four datasets, we construct training and evaluation splits as follows. When either $\eta_\ell$ or $\eta_u$ is fixed $\eta_o$, samples are drawn from $\mathcal{D}_1$ with the desired size $N$ (for unlabeled data) or $n$ (for labeled data). When either $\eta_\ell$ or $\eta_u$ is fixed to 0.5, balanced samples are instead drawn from $\mathcal{D}_2$. Several experiments involve fixing $\eta_\ell$ while varying $\eta_u$. In this setting, if necessary, we keep the labeled data unchanged and supplement the unlabeled data with additional samples independently drawn from $\mathcal{D}_1$ or $\mathcal{D}_2$, while ensuring that the total unlabeled sample size $N$ remains constant across different $\eta_u$. The balanced dataset $\mathcal{D}_3$ is reserved for testing, and when required, we further split 20% of $\mathcal{D}_3$ as a separate validation set.

### E.2 RESULTS FOR INTERPRETING W2S UNDER SPURIOUS CORRELATIONS

| Dataset | # of model pairs with increased W2S gain | |
|---------|------------------|---------------------|
| | Average accuracy | Worst group accuracy |
| Waterbirds | 10/10 | 10/10 |
| BFFHQ | 10/10 | 9/10 |
| BG-COCO | 9/10 | 10/10 |
| ImageNet-9 | 7/10 | — |

Table 7: Proportion of teacher-student pairs that exhibit an increase in W2S gain as $\eta_u$ increases from 0 to the maximum feasible value of $\eta_u$ (Waterbirds: 0.5, BFFHQ: 0.23, BG-COCO: 0.5, ImageNet-9: 0.4) when $\eta_\ell = 0.5$, summarized across all datasets. ImageNet-9 has no well-defined worst group, so only average accuracy is reported.

In Section 3.2, we primarily presented how the average W2S gain across all teacher-–student pairs varies with increasing $\eta_u$ on each dataset. Here, we further provide results for individual model pairs. Specifically, Figure 6 compares the difference in W2S gain between the group-balanced ($\eta_\ell = 0.5$) and group-imbalanced ($\eta_\ell = \eta_o$) settings on selected datasets. Table 7 summarizes, for $\eta_\ell = 0.5$, the proportion of model pairs that exhibit an increase in W2S gain as $\eta_u$ increases from 0 across all datasets. Table 8 summarizes, for $\eta_\ell = \eta_o$, the proportion of model pairs that exhibit a decrease in W2S gain as $\eta_u$ increases from $\eta_o$ across all datasets. These results further validate our theoretical analysis in Section 2.2, which predicts that in most cases the larger the gap between $\eta_u$ and $\eta_\ell$, the smaller the resulting W2S gain.

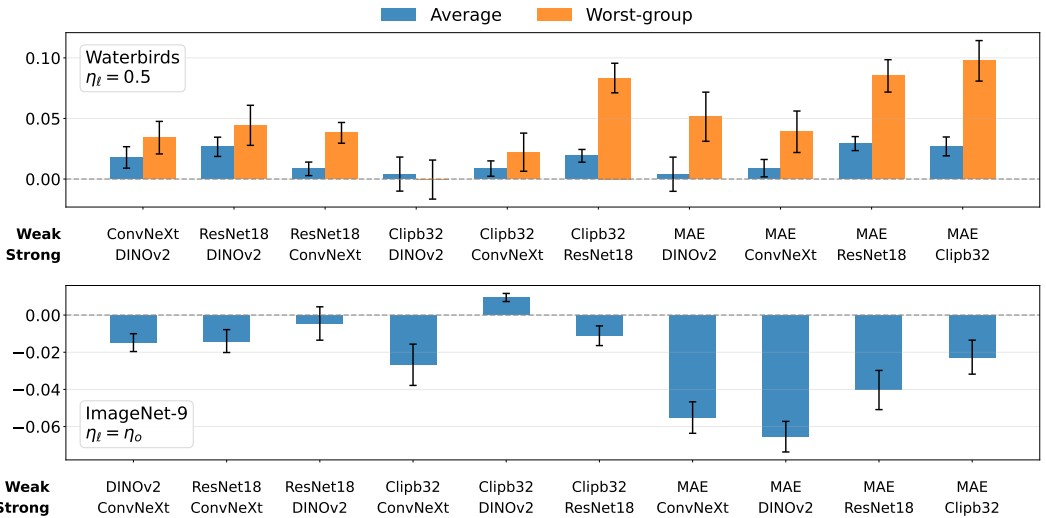

Figure 6: Top: On the Waterbirds dataset, the change in W2S gain (value at $\eta_u = 0.5$ minus value at $\eta_u = 0$) across all teacher–student pairs with fixed $\eta_\ell = 0.5$. Bottom: On the ImageNet-9 dataset, the change in W2S gain (value at $\eta_u = 0.5$ minus value at $\eta_u = \eta_o$) across all teacher–student pairs with fixed $\eta_\ell = \eta_o$. ImageNet-9 does not have a clearly defined worst group and is therefore omitted from the bottom panel.

| Dataset | # of model pairs with decreased W2S gain | |
|---|---|---|
| | Average accuracy | Worst group accuracy |
| Waterbirds | 7/10 | 8/10 |
| BFFHQ | 8/10 | 7/10 |
| BG-COCO | 8/10 | 7/10 |
| ImageNet-9 | 9/10 | — |

Table 8: Proportion of teacher-student pairs that exhibit a decrease in W2S gain as $\eta_u$ increases from $\eta_o$ to 0.5 when $\eta_\ell = \eta_o$, summarized across all datasets. ImageNet-9 has no well-defined worst group, so only average accuracy is reported.

### E.3 RESULTS FOR ENHANCED W2S

**Model training.** Enhanced-W2S improves upon vanilla W2S by retraining the strong student after the initial W2S fine-tuning. First, we select a fraction $p \in (0, 1]$ of $\hat{S}$ consisting of those samples for which the student exhibits the lowest prediction entropy. Second, we apply the GCE loss $\mathcal{L}_{\text{GCE}}(\mathbf{x}_i, \hat{y}_i; q)$ with parameter $q \in (0, 1]$ to each selected sample $(\mathbf{x}_i, \hat{y}_i)$. We tune the hyperparameters by grid search over $p \in \{0.2, 0.4, 0.6, 0.8, 1.0\}$ and $q \in \{0, 0.2, 0.7\}$, where $q = 0$ corresponds to the CE loss (i.e., the $q \to 0$ limit of GCE). To avoid a trivial overlap with the vanilla W2S baseline, $(p, q) = (1, 0)$ is excluded from the Enhanced-W2S search space. In the case of $(\eta_\ell, \eta_u) = (\eta_o, 0.5)$, we further restrict the subset ratio to $p \in \{0.2, 0.4, 0.6\}$ to emphasize the role of high-confidence subsets in filtering for the majority group. Each run of Enhanced-W2S is repeated with multiple random seeds, and the reported results are obtained by averaging across seeds.

**Role of confidence-based selection.** When $(\eta_\ell, \eta_u) = (\eta_o, 0.5)$, Figure 7 shows that samples with high student confidence (i.e., low predictive entropy) after W2S fine-tuning are almost exclusively drawn from the majority group, and furthermore are nearly always assigned the correct pseudolabels by both the weak teacher and the strong student. At the same time, Theorem 2 predicts that reducing $\eta_u$ from 0.5 directly increases the W2S gain. These two observations together suggest that confidence-based selection provides significant benefits for improving W2S performance in the setting $(\eta_\ell, \eta_u) = (\eta_o, 0.5)$.

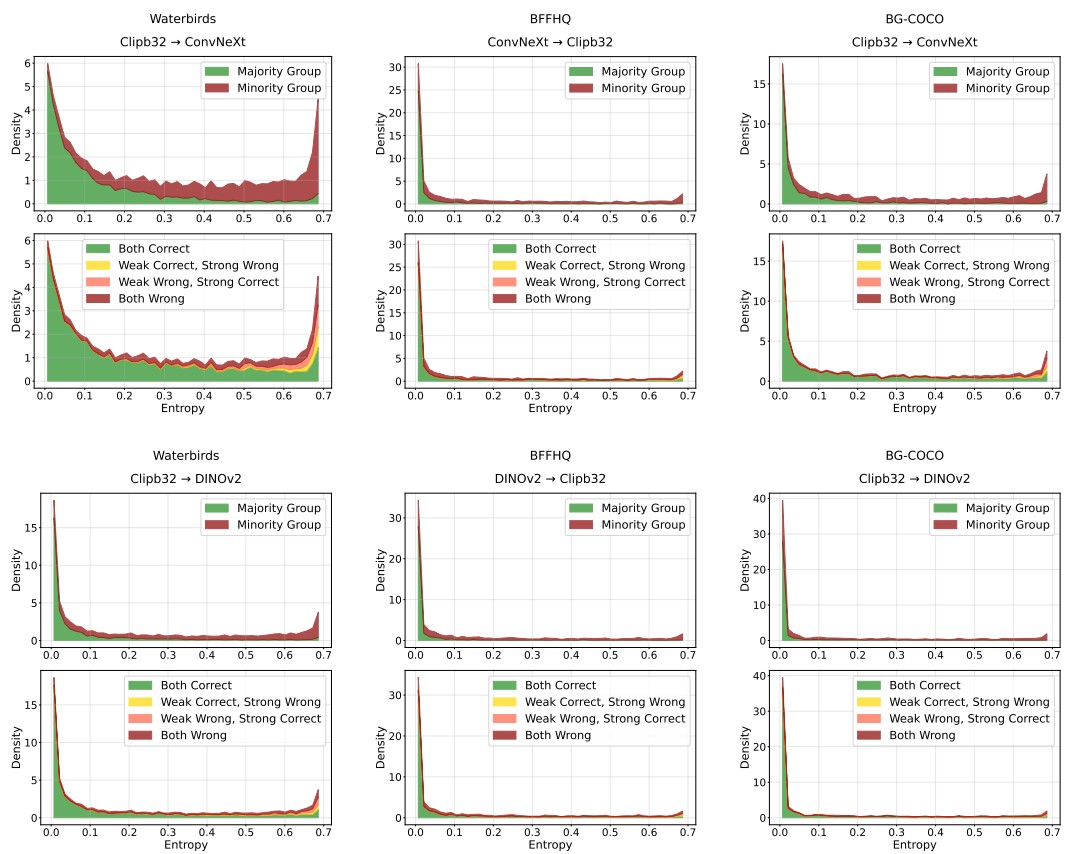

Figure 7: Student confidence on unlabeled data as stacked density plots of predictive entropy ($\eta_\ell = \eta_o, \eta_u = 0.5$). Each panel shows the student's predictive entropy (after W2S fine-tuning), visualized as two stacked density plots: (top) split by group (majority vs. minority) and (bottom) split by prediction correctness of the weak teacher and the strong student. Columns correspond to datasets (Waterbirds, BFFHQ, BG-COCO). Rows correspond to model pairs: (ConvNeXt, Clipb32) and (Clipb32, DINOv2).

| Dataset | $\eta_\ell$ | $\eta_u$ | Mean relative improvement (%) | |
|---|---|---|---|---|
| | | | Average Accuracy | Worst Group Accuracy |
| Waterbirds | 0.5 | $\eta_o$ | 6.32 | 10.12 |
| | $\eta_o$ | 0.5 | 7.72 | 32.15 |
| BFFHQ | 0.5 | $\eta_o$ | 5.52 | 4.06 |
| | $\eta_o$ | 0.5 | 3.12 | 3.57 |
| BG-COCO | 0.5 | $\eta_o$ | 10.51 | 11.76 |
| | $\eta_o$ | 0.5 | 4.50 | 3.71 |
| ImageNet-9 | 0.5 | $\eta_o$ | 12.23 | — |
| | $\eta_o$ | 0.5 | 11.23 | — |

Table 9: Mean relative improvement (%) of Enhanced-W2S over vanilla W2S, averaged across selected teacher–student pairs, for both average accuracy and worst group accuracy. For each dataset, we select all model pairs whose relative strength relationship remains consistent across different $(\eta_\ell, \eta_u)$ settings.

**Mean relative gains.** Table 9 summarizes the mean relative improvement of Enhanced-W2S over vanilla W2S, averaged across all teacher–student pairs. Consistent with the main text, our method achieves clear gains under both average accuracy and worst group accuracy. On the Waterbirds dataset,

we further compare the performance of Enhanced-W2S with the auxiliary confidence loss proposed in (Burns et al., 2024), which was also designed to improve the generalization ability of W2S. Specifically, we perform a grid search over the auxiliary confidence loss weight $\alpha \in \{0.2, 0.4, 0.6, 0.8\}$, and Table 10 reports, for each $(\eta_\ell, \eta_u)$ configuration, the mean relative improvement of Enhanced-W2S minus the mean relative improvement obtained with the auxiliary confidence loss. Our method yields larger gains in all cases, confirming that it is motivated by our theoretical analysis (see Section 4) and is specifically tailored to address W2S under spurious correlation.

| $\eta_\ell$ | $\eta_u$ | Difference in mean relative improvement (%) | |
| --- | --- | --- | --- |
| | | Average Accuracy | Worst Group Accuracy |
| 0.5 | $\eta_o$ | 5.22 | 3.88 |
| $\eta_o$ | 0.5 | 5.90 | 3.41 |

Table 10: Difference in mean relative improvement (%) Waterbirds, computed as Enhanced-W2S minus the auxiliary confidence loss baseline, averaged across selected teacher–student pairs for each $(\eta_\ell, \eta_u)$ configuration.

## F  GROUP FAIRNESS IN W2S GENERALIZATION

Ensuring that algorithmic decisions do not exhibit systematic bias against certain attributes (e.g., race, gender, age) has long been a central objective in fair machine learning (Liu et al., 2019; Oneto & Chiappa, 2020; Mehrabi et al., 2021). At the same time, when the data contains spurious correlations caused by group imbalance, unfairness across different groups is likely to arise, as the model tends to rely on spurious features when making predictions (Izmailov et al., 2022). This lack of group fairness is particularly concerning when groups are defined by sensitive attributes. Therefore, a line of work on mitigating spurious correlation has explicitly targeted group robustness, and the evaluation metrics adopted in this literature (e.g., worst group accuracy) can be interpreted as a measure of fairness. In parallel, several works have more directly studied the relationship between spurious correlations and formal notions of group fairness (Veitch et al., 2021; Schrouff et al., 2024).

In this section, we extend our analysis of W2S under spurious correlation to incorporate the notion of group fairness. Under W2S, the minority group proportions in both the labeled dataset ($\eta_\ell$) and the unlabeled dataset ($\eta_u$) jointly influence the extent to which the strong student preserves group-level parity after the W2S process.

**Definition 4** (Group risk disparity). *Under Definitions 1 and 2, we define the group risk disparity of the strong student after W2S fine-tuning as*

$$\Delta_{\mathrm{grp}}(f_S) := \left| \mathbb{E}_{\mathcal{D}(\eta_u)^N, \, \mathcal{D}(\eta_\ell)^n} \left[ \mathbf{ER}_0(f_S) \right] - \mathbb{E}_{\mathcal{D}(\eta_u)^N, \, \mathcal{D}(\eta_\ell)^n} \left[ \mathbf{ER}_1(f_S) \right] \right|, \qquad (33)$$

*where $\mathbf{ER}_0(f_S)$ and $\mathbf{ER}_1(f_S)$ denote the excess risks of the student on the majority ($\eta_t = 0$) and minority ($\eta_t = 1$) groups, respectively.*

In Definition 4, we quantify the group fairness through the absolute difference between the student's excess risk on the majority group and the minority group. It is important to note that our definition of group risk disparity is directly aligned with the notion of perfect fairness (also referred to as risk parity) in the group-fairness literature (Williamson & Menon, 2019; Liu et al., 2025b). In particular, the condition $\Delta_{\mathrm{grp}}(f_S) = 0$ is equivalent to achieving perfect fairness (risk parity).

**Corollary 1** (Group risk disparity of W2S). *Under Definitions 1 and 2 and assumption 1, the group risk disparity of the strong student after W2S fine-tuning satisfies*

$$\Delta_{\mathrm{grp}}(f_S) \xrightarrow{\mathbb{P}} \frac{\sigma_y^2 \gamma_z}{\sigma_\xi^2} \left| 2(\eta_\ell - \eta_u) \boldsymbol{\mu}_T^\top \boldsymbol{\Xi} \boldsymbol{\mu}_S - (1 - 2\eta_u) \left( \|\boldsymbol{\Xi}\boldsymbol{\mu}_S\|_2^2 + \nu_z (p_T - p_{T \wedge S}) \|\boldsymbol{\mu}_S\|_2^2 \right) \right|$$

Corollary 1 follows directly from the precise asymptotic characterization of the strong student excess risk in Theorem 3, providing a precise quantification of the group risk disparity in the proportional asymptotic limit.

We outline several key insights from Corollary 1 below:

(a) **Low teacher-student similarity ($p_{T \wedge S} = 1$) brings robustness of group fairness $\Delta_{\mathrm{grp}}(f_S)$ to teacher bias $\eta_\ell < 0.5$, where W2S is fair if the unlabeled training set is balanced $\eta_u = 0.5$.**

Notably, when $p_{T \wedge S} = 1$ (i.e., $\|\boldsymbol{\Xi}\|_F^2 = 0$), $\Delta_{\text{grp}}(f_S)$ becomes independent of $\eta_\ell$ and is only affected by $\eta_u$ through the $(1 - 2\eta_u)$ factor. When $\eta_u = 0.5$ further, we have $1 - 2\eta_u = 0$, and therefore $\Delta_{\text{grp}}(f_S) \xrightarrow{\mathbb{P}} 0$, i.e., the strong student from W2S fine-tuning is fair, even though a biased weak teacher fine-tuned with $\eta_\ell < \eta_u$ can still hurt the generalization of the strong student.

(b) **Low teacher-student similarity ($p_{T \wedge S} = 1$) induces group fairness of W2S, $\Delta_{\text{grp}}(f_S) \to 0$, as $\nu_z \to 0$.** While $p_{T \wedge S} = 1$ (i.e., $\|\boldsymbol{\Xi}\|_F^2 = 0$) alone does not guarantee fairness, it provides

$$\Delta_{\text{grp}} \xrightarrow{\mathbb{P}} (1 - 2\eta_u)\gamma_z\nu_z\frac{\sigma_y^2}{\sigma_\xi^2}(p_T - p_{T \wedge S})\|\boldsymbol{\mu}_S\|_2^2 \asymp (1 - 2\eta_u)\gamma_z\nu_z,$$

where we have $\Delta_{\text{grp}}(f_S) \xrightarrow{\mathbb{P}} 0$ as $d_z, n, N \to \infty$ if $\nu_z \to 0$, i.e., when $N$ is large enough compared to $d_z$, low teacher-student similarity induces fairness of W2S.

(c) **For high teacher-student similarity ($p_{T \wedge S} \to p_S$), group fairness of the student $\Delta_{\text{grp}}(f_S)$ is influenced by the fairness of the teacher $\eta_\ell$, with the dependence determined by $\boldsymbol{\mu}_T^\top\boldsymbol{\Xi}\boldsymbol{\mu}_S$.** In particular, when $p_{T \wedge S} \to p_S$ so that $\left|\boldsymbol{\mu}_T^\top\boldsymbol{\Xi}\boldsymbol{\mu}_S\right|$ is non-negligible, W2S is fair (i.e., $\Delta_{\text{grp}}(f_S) \xrightarrow{\mathbb{P}} 0$) when

$$\eta_\ell^{\text{fair}} = \eta_u + (1 - 2\eta_u)\frac{\|\boldsymbol{\Xi}\boldsymbol{\mu}_S\|_2^2 + \nu_z(p_T - p_{T \wedge S})\|\boldsymbol{\mu}_S\|_2^2}{\boldsymbol{\mu}_T^\top\boldsymbol{\Xi}\boldsymbol{\mu}_S},$$

assuming it falls in the range $\eta_\ell^{\text{fair}} \in [0, 0.5]$, while the group fairness gets worse (i.e., $\Delta_{\text{grp}}(f_S)$ increases) as $\eta_\ell$ deviates from $\eta_\ell^{\text{fair}}$. Notably,

(1) if $\eta_\ell^{\text{fair}} < 0$, the group fairness gets worse as $\eta_\ell$ increases, best when $\eta_\ell = 0$; while

(2) if $\eta_\ell^{\text{fair}} > 0.5$ the group fairness gets worse as $\eta_\ell$ decreases, best when $\eta_\ell = 0.5$.

