# OpenReview forum: "Does Weak-to-strong Generalization Happen under Spurious Correlations?"
_ICLR.cc/2026/Conference — ICLR 2026 Poster_

### Official Review · Reviewer_rtLS · 2025-10-17

**Soundness:** 3
**Presentation:** 3
**Contribution:** 3
**Rating:** 6
**Confidence:** 3

**Summary:**

This paper presents a unified theoretical and algorithmic investigation into the phenomenon of Weak-to-Strong (W2S) generalization under the presence of spurious correlations caused by group imbalance. The authors model two key sources of imbalance: the minority fraction in the labeled data ($\eta_l$) used to train the teacher, and the minority fraction in the unlabeled data ($\eta_u$) used for W2S fine-tuning.

The main theoretical contribution, derived within the proportional asymptotic limit of a linear regression setting, precisely characterizes the W2S gain. The theory predicts that W2S always occurs when $\eta_l = \eta_u$, but critically, W2S generalization can fail when $\eta_l \neq \eta_u$, with the gain diminishing as the squared difference $(\eta_u - \eta_l)^2$ increases.

Based on these findings, the paper proposes Enhanced-W2S, a simple yet effective algorithmic remedy. This method involves post-W2S retraining of the strong student on its high-confidence data subset using the Generalized Cross-Entropy (GCE) loss. The method is entirely group-label-free. Extensive experiments across four spurious correlation benchmarks and ten different teacher-student pairs confirm the theoretical predictions and demonstrate consistent, substantial gains achieved by Enhanced-W2S over vanilla W2S fine-tuning.

**Strengths:**

1. The paper addresses a highly critical, yet previously unexplored, aspect of W2S generalization: its robustness to real-world biases like spurious correlations. The idea attracts me a lot and I'd like to see the core idea published in the conference. To my best knowledge, this issue has not been investigated in previous W2SG papers, both empirically and theoretically.
2. The key theoretical conclusion that W2S gain increases as the teacher-student similarity decreases is intuitive and aligns with teacher-student disagreement literature (Charikar et al., 2024; Mulgund & Pabbaraju, 2025; Yao et al., 2025; Xu et al., 2025).
3. To my knowledge, the paper includes a thorough review of relevant literature concerning W2S generalization and group robustness.
4. The result appears correct since it can recover Dong et al. (2025), though I have not verified the proof details.

**Weaknesses:**

## Weaknesses
1. The paper is generally well-written, but it could be significantly improved by providing more explicit and intuitive explanations for every mathematical choice and definition (e.g., the precise geometric meaning of $\left\\|\Xi\right\\|_F^2$ and the role of $p_T$ and $p_S$) alongside the formal derivations. This would greatly enhance the accessibility and understanding of the theoretical model.
2. The paper's theory successfully demonstrates that under certain ideal conditions, such as when the minority group proportions in the labeled and pseudo-labeled data match ($\eta_l = \eta_u$), W2SG is more likely to occur. However, the conclusion doesn't seem to fully elucidate the precise conditions under which W2SG is guaranteed to succeed or fail in the non-ideal scenario where data imbalance persists (i.e., $\eta_l \neq \eta_u$). The theory notes that W2S may fail when $\eta_u \ne \eta_l$. For instance, when $\eta_l \neq \eta_u$, the W2S gain tends to diminish but is not universally negative, suggesting there may be boundary conditions or trade-offs that are not fully explored by the analysis on $(\eta_u - \eta_l)^2$ alone.
3. The detailed excess risk characterizations rely on the ridgeless linear regression setting and a specific, low-intrinsic-dimension kernel assumption.
4. The Enhanced-W2S method introduces two new hyperparameters ($p$ for subset fraction and $q$ for GCE loss). While tuning ranges are provided, the paper does not thoroughly explore the robustness of the optimal $(p, q)$ pair across different $\eta_l, \eta_u$ settings or the effort required to tune them.

Minor: Line 270: The text above Figure 2 should be revised.

## Future Work
Since the paper quantifies the error, I strongly suggest the authors separately quantify the excess risk for the minority group and the majority group in the theoretical analysis. The current theoretical results (Theorems 1 and 2) characterize the average excess risk $ER_{\eta_t}(f)$ across a test distribution $\mathcal{D}(\eta_t)$, but presenting the excess risk specifically for the majority ($\eta_t=0$) and minority ($\eta_t=1$) test groups would provide deeper insight into how spurious correlation affects group performance.

This work touches closely upon the concept of fairness in machine learning, as spurious correlation fundamentally impacts group robustness. The data imbalance investigated here—specifically the role of the minority group fractions $\eta_l$ and $\eta_u$—bears a striking resemblance to the problem of base rate differences discussed in the fairness literature [1]. Given this relationship, I recommend including a discussion on this intersection. Specifically, exploring the connection between the W2S failure mechanism (when $\eta_l \neq \eta_u$) and established fairness criteria could be a valuable direction for future work in W2S.  If you have time, and can derive the difference in errors between the two groups from your current proof to build a similar theory of fairness (maybe in the appendix), this paper would then connect W2S to both robustness and fairness. This would make it much more interesting, and I'd be able to offer you my stronger support. Please don't hesitate to correct me if I've misunderstood anything.

[1] The Implicit Fairness Criterion of Unconstrained Learning. ICML 2019.

**Questions:**

- I infer that achieving parity between the minority group proportions in the labeled ($\eta_l$) and unlabeled ($\eta_u$) datasets is crucial for W2SG. Theoretically, does the primary W2S gain come from the minority or the majority group? In the optimal gain scenario ($\eta_u = \eta_l$), which group primarily contributes to the remaining excess risk?
- Strictly speaking, is data imbalance the sole source of spurious correlation in the setting analyzed by this paper?
- How does group robustness affect fairness?

---

> ### Author Response · Authors · 2025-11-22
> **Response to Reviewer rtLS [1/2]**
>
> We thank the reviewer for their comments and constructive suggestions regarding the improvements and future directions. We also appreciate the reviewer’s recognition of the novelty and importance of studying W2S generalization under spurious correlations.
>
> ## Key Clarifications on the Weaknesses
>
> **[W1. Improved presentation of our theoretical setup through the new Figure 1.]**
> Thanks to the constructive suggestion, we included a new figure, Figure 1, in the revision to visualize our theoretical setup described in Assumptions 1 and 2. Meanwhile, we point the reviewer to lines 183-184 in the revision for the intuitive interpretation of $\Xi$, and lines 236-237 in the revision for the intuitions behind $p_T$ and $p_{T \cap S}$.
>
> **[W2. Precise necessary and sufficient conditions for W2S follow directly from Theorems 1 and 3, but do not provide additional insights beyond Remark 2.]**
> First, we remark that the precise characterizations of the teacher and student generalization errors in Theorems 1 and 2 (or 3) directly provide the necessary and sufficient conditions for W2S to happen, i.e., whenever the W2S gain $\Delta \mathcal R_{\eta_t} > 0$ (for clarification, we revised this definition to eq.(4) and highlighted its implication in the revision). Instead of attributing space to reiterate on such trivial statements that are hard to interpret, we focus on analyzing the different choices of important parameters (e.g.,$\eta_\ell$ and $\eta_u$) to extract interpretable and generic insights from the theory (e.g., the trend of W2S gain as $(\eta_\ell - \eta_u)^2$ increases).
>
> **[W3. The low intrinsic dimension and kernel dynamics of fine-tuning are well-justified by empirical observations.]**
> As explained in Remark 1, we consider the ridgeless regression setting because (i) it provides asymptotically optimal regularization for low-rank features, and (ii) extending the analysis to ridge regression does not alter the key insights. Meanwhile, as highlighted in lines 145-147 of the revision (or lines 126-128 in the submission), our assumptions on the low intrinsic dimension and kernel dynamics of fine-tuning are motivated and well-supported by extensive empirical observations (e.g., [1,4]). Our experimental results further confirm the empirical validity of these theoretical assumptions.
>
> We acknowledge that the assumption about the kernel dynamics of fine-tuning can be considered as a necessary theoretical simplification that often, but not always, holds in practice [4]. The extension of our theory beyond the kernel regime (e.g., taking the gradient descent dynamics of non-linear feature learning into consideration) is an interesting future direction, but not relevant to our main question of interest (i.e., W2S under spurious correlation).
>
> **[W4. Effects of hyperparameters on the proposed Enhanced W2S method.]**
> We thank the reviewer for raising this constructive point.
> 1. For tuning cost, we perform a grid search over $(p, q)$ pairs (see Appendix E.3). For each $(p, q)$ pair, we only retrain on a subset of the data, and using the GCE loss adds almost no extra computation. So each Enhanced-W2S run is cheaper than retraining on the full dataset with standard ERM.
> 2. For robustness to the choice of $(p, q)$, we provide the following additional results. On Waterbirds (Table 1), when $\eta_\ell = 0.5$ and $\eta_u = \eta_o$, the mean relative improvement across model pairs is $6.3$%. Using the second-best and third-best $(p, q)$ pairs yields $5.5$% and $4.5$%, respectively. Similarly, when $\eta_\ell = \eta_o$ and $\eta_u = 0.5$, the best, second-best, and third-best mean improvements are $7.7$%, $6.8$%, and $6.0$%. Since changing $p$ or $q$ within our search grid leads to noticeable shifts in subset size or GCE behavior, these results indicate that our method is reasonably robust to the choice of $(p, q)$.

---

> ### Author Response · Authors · 2025-11-22
> **Response to Reviewer rtLS [2/2]**
>
> ## Response to Future Work
> We thank the reviewer for highlighting that our analysis naturally connects group robustness with fairness. We agree that this relationship broadens the impact of our work, and we added a new section in the appendix to explicitly discuss this connection.
>
> Specifically, we define the absolute difference between the student’s excess risk on
> the majority group and the minority group as a direct metric of group fairness. Note that this definition aligns with the notion of perfect fairness (also called risk parity) in the group fairness literature [2]. Our analysis shows that:
> 1. Low teacher-student similarity ($p_{T \cap S}= 1$) brings robustness of group fairness $\Delta_{\mathrm{grp}}(f_S)$ to teacher bias $\eta_\ell < 0.5$, where W2S is fair if the unlabeled training set is balanced $\eta_u = 0.5$.
> 2. Low teacher-student similarity ($p_{T \cap S}= 1$) induces group fairness of W2S, $\Delta_{\mathrm{grp}}(f_S) \to 0$, as $\nu_z \to 0$.
> 3. For high teacher-student similarity ($p_{T \cap S} \to p_S$), group fairness of the student $\Delta_{\mathrm{grp}}(f_S)$ is influenced by the fairness of the teacher $\eta_\ell$, with the dependence determined by $\mu_T^\top \Xi \mu_S$.
>
> ## Responses to the Questions Section
>
> - Q1: This question can be investigated by plugging $\eta_t = 0$ (majority) and $\eta_t = 1$ (minority) into Theorems 1 and 3 and comparing the W2S gains $\Delta \mathcal R_{0}$ vs. $\Delta \mathcal R_{1}$. In particular:
>   1. “Does the primary W2S gain come from the minority or the majority group?”: In the benign cases for W2S (i.e., $\Xi=0$ and $\nu_z \ll 1$), the minority group contributes more to the W2S gain when the teacher is biased ($\eta_\ell < 0.5$); and the gap between the minority vs. majority W2S gains increases as $\eta_\ell$ decreases. In the extreme case of $\eta_\ell = 0$, the majority W2S gains can be negative (i.e., W2S may not happen in the majority group) if $(\eta_u - \eta_\ell)^2$ is large. The non-negligible $||\Xi||_F^2$ and $\nu_z$ cases suffer from further complications, where the minority vs. majority W2S gains depend additionally on the interaction among $\mu_T, \mu_S, \Xi$.
>   2. “In the optimal gain scenario ($\eta_u=\eta_\ell$), which group primarily contributes to the remaining excess risk?”: We again consider the benign cases for W2S, $\nu_z \ll 1$, to avoid unnecessary complications. When $\eta_u=\eta_\ell < 0.5$, the minority group contributes more to the W2S gain, where the gap between the minority vs. majority W2S gains decreases as $\eta_u$ and $||\Xi \mu_S||_2^2 \le ||\mu_T||_2^2$ increases. When $\eta_u = 0.5$, the two groups contribute equally.
>
> - Q2: Yes, in this work, we focus on spurious correlations caused solely by group imbalance, one of the most common sources of spurious correlation [3], in both the teacher’s and the student’s data.
>
> - Q3: Please see our Response to the Future Work section for a detailed explanation.
>
> We are happy to answer any further questions you may have. If our responses above have addressed your concerns, we would truly appreciate your strong support.
>
> ## References
> [1] Aghajanyan et al. *Intrinsic dimensionality explains the effectiveness of language model fine-tuning.* ACL 2021.
>
> [2] Williamson et al. *Fairness risk measures.* ICML 2019.
>
> [3] Sagawa et al. *Distributionally robust neural networks.* ICLR 2020.
>
> [4] Malladi et al. *A kernel-based view of language model fine-tuning.* ICML 2023.

---

> ### Comment · Reviewer_rtLS · 2025-11-26
>
> Thank you for the further clarifications regarding your theoretical results. Now I understand the point. Furthermore, I especially appreciate the authors for providing the extension to the fairness literature with their theoretical framework in such a short timeframe of the rebuttal. I will certainly give the paper my stronger support. I will increase my score to 8, but the confidence remains since I do not check the proof details.

---

> > ### Author Response · Authors · 2025-11-27
> >
> > Many thanks for your timely response. We are glad to hear that our rebuttal addressed your concerns, and we truly appreciate your constructive feedback on including the appendix on fairness, as well as your strong support.

---

### Official Review · Reviewer_ZDqv · 2025-10-27

**Soundness:** 3
**Presentation:** 3
**Contribution:** 3
**Rating:** 6
**Confidence:** 4

**Summary:**

This paper investigates weak-to-strong generalization in the presence of spurious correlations arising from group imbalance. The authors provide a theoretical analysis demonstrating that the disparity in group imbalance between the weak and strong model training datasets is the key factor controlling the weak-to-strong gain. Their analysis shows that weak-to-strong generalization can fail when the group imbalance distributions differ significantly between the two phases. Furthermore, the paper proposes a novel method called Enhanced-W2S, which is empirically shown to achieve superior weak-to-strong generalization on spurious correlation benchmarks.

**Strengths:**

* This work addresses the role of spurious correlation in weak-to-strong generalization, a topic that has not been sufficiently explored in prior literature.
* The theoretical framework and results are clearly stated, and the paper provides detailed, insightful discussions on these findings.
* Beyond the theoretical analysis, the authors propose a novel method, Enhanced-W2S, as an effective remedy. The effectiveness of this method is well-supported by extensive experiments on various spurious correlation benchmarks and architectures.

**Weaknesses:**

* The theoretical framework, particularly the formal statement of Assumption 2, can be hard to follow at first glance. I believe that adding a more intuitive, illustrative description can significantly improve reader understanding.
* Theoretical discussions are provided before explaining some technical terms (See Questions).
* The connection between theoretical insights and the intuition behind the proposed method is not clearly discussed (See Questions).

**Questions:**

* In main results, the terms $\mathcal{V}_T^{(0)}, \mathcal{V}_T^{(1)},  \mathcal{V}_S^{(0)}, \mathcal{V}_S^{(1)}$ appear. What is the precise meaning of these terms? It seems this term might relate to variance (according to line 228), but there are no explicit discussions on it. It would be helpful if the authors could explicitly discuss the meaning of each term right after the statement of the main theorem.
* In the draft, there is a discussion on why the proposed method can be beneficial in the spurious correlation setting. However, the connection between this intuition and the theoretical insights is vague for me. Could you discuss this point in more detail?

---

> ### Author Response · Authors · 2025-11-22
> **Response to Reviewer ZDqv**
>
> We thank the reviewer for their time and insightful suggestions. We are glad that they find our problem formulation important and our theoretical and experimental findings solid.
>
> ## Key Clarifications on the Weaknesses
>
> **[W1. Improved presentation of our theoretical setup through the new Figure 1.]**
> Thanks to the constructive suggestion regarding presentation, we included a new figure, Figure 1, in the revision to visualize our theoretical setup described in Assumptions 1 and 2.
>
> **[W2 (Q1). Further clarifications on technical terms in the revision.]**
> Thanks again for the constructive suggestion. In the revision, we added intuitive explanations for $\mathcal{V}_T^{(0)}, \mathcal{V}_T^{(1)}, \mathcal{V}_S^{(0)}, \mathcal{S}_T^{(1)}$ in Theorem 2. In particular, $\mathcal{V}_S^{(0)}, \mathcal{V}_T^{(0)}$ represent the variance of the strong student and the weak teacher that arise from label noise; $\mathcal{V}_S^{(1)}, \mathcal{V}_T^{(1)}$ are the variance of $f_S$ and $f_T$ induced by spurious correlations.
>
> **[W3 (Q2) Retraining the strong student on its high-confidence subset of the unlabeled data alters $\eta_u$ and reduces the gap $(\eta_u - \eta_\ell)^2$.]**
> To demonstrate how our theory motivates our Enhanced W2S method, let’s consider using a biased teacher supervisedly fine-tuned on a labeled set with a minority proportion $\eta_\ell = \eta_o \approx 0$ to teach a strong student on a balanced unlabeled set with $\eta_u = 0.5$.
> Recall that our theory suggests that the W2S gain increases as the gap $(\eta_u - \eta_\ell)^2$ decreases. For W2S, the fine-tuned weak teacher is generally considered fixed, whose labeled training data are not available to the strong student (see our responses to Reviewer c6kM, [W1] and [W3.1]). Therefore, $\eta_\ell \approx 0$ is fixed, and to close the gap $(\eta_u - \eta_\ell)^2$, we need to decrease the minority group proportion in the unlabeled set, $\eta_u$, by fine-tuning the strong student on more majority data pseudolabeled by the weak teacher. Now, we empirically observe that the high-confidence subset of the strong student in the unlabeled data is dominated by the majority group (see Figure 7 in Appendix E.3 in the revision, or Figure 6 in the submission), intuitively because the biased weak teacher tends to be more accurate on the majority group. Therefore, retraining the W2S fine-tuned strong student on this high-confidence subset drives $\eta_u$ toward $\eta_\ell$, thereby bringing improvement to W2S.
>
> Both Q1 and Q2 from the Questions section are addressed by **[W2, W3]**, respectively. We are happy to answer any further questions you may have. We hope that the above clarifications address your concerns, and we would truly appreciate a re-evaluation accordingly.

---

> > ### Comment · Reviewer_ZDqv · 2025-11-26
> >
> > Thank you for your response. The response and revisions successfully addressed my concerns. In particular, I believe the added illustration of the theoretical framework will be helpful for readers. I now think this a good paper and have raised my score to 8 accordingly.

---

> > > ### Author Response · Authors · 2025-11-27
> > >
> > > Many thanks for your timely response. We are glad to hear that our rebuttal addressed your concerns, and we truly appreciate your constructive feedback and strong support.

---

### Official Review · Reviewer_caiF · 2025-11-01

**Soundness:** 3
**Presentation:** 3
**Contribution:** 3
**Rating:** 6
**Confidence:** 3

**Summary:**

The paper considers a weak-to-strong generalization setup with ridgeless regression under spurious correlations with teacher and student models having fixed sets of features that entangle the core and group features differently. They show that in the appropriate limit of sample sizes and core feature dimension, weak-to-strong generalization happens (i) whenever the spurious correlations are the same and there is enough teacher labeled samples, (ii) when there is a mismatch of spurious correlations between the teacher and the student the the W2S gain decreases with this difference, and (iii) algorithmic improvement to standard W2S training pipeline that improves the W2S student performance when there are spurious correlations.

**Strengths:**

The main significance of the paper is:
1. Consider regression with spurious correlation in the context of W2S is an interesting way to capture different pre-trainings and capacities of student and teacher models
2. Provides crisp characterization of weak-to-strong generalizations in regression with spurious correlations in the appropriate limit of sample sizes and feature dimensions in terms of mismatch in level of spurious correlation, similarity between student and teacher features, and the number of samples
3. Alternative way of performing W2S training when the spurious correlations don’t align with retraining the student model on high confidence points which improves over the vanilla W2S.

The idea of considering spurious correlation is original. The writing is clear and the mathematical arguments seem sound.

**Weaknesses:**

1. The main theorem, Theorem 3, does not formally establish weak-to-strong generalizations. It only gives a sufficient condition under which weak-to-strong generalization happens in this case in terms of $p_{T S}$. It would be more interesting to find natural choices of $\phi_S$ and $\phi_T$ for which we can understand $p_{T  S}$ and therefore W2S improvement.
2. Here the W2S improvement, like in other ridgeless regression setups, comes “artifically” from the fact that the teacher is trained with too little samples and thus is not optimally. The student is able to  improve over the teacher because it has sufficient samples, and so the effect of student having better features than the teacher is confounded with the effect of the teacher being suboptimal.
3. The paper asks whether W2S still happens in data with spurious correlation, but the definition of what makes a teacher “weak” and a student “strong” (Assumption 2) is based on the spurious correlations, i.e. that the weak model entangles the core and group features and the strong model doesn’t. The result could be further strengthened with a more agnostic way of defining weak and strong which implies Assumption 2. That way, we could use this setup to get insights on the mechanisms underpinning W2S generalization.
4. It is unclear how broadly applicable algorithmic insights are to other non-vision related W2S setups.

**Questions:**

1. Is there a choice of parameters in Assumption 2 (i.e. T and S) under which we can have a theoretical characterization of $p_{T\and S}$?
2. Why is the asymptotic limit taking in this exact form? Is it mostly for the simplicity of analysis of the excess risk in ridgeless regression or is there any intuition behind? Could get a similar result for finite sample size and feature dimensions?
3. Is there a natural agnostic definition of weak and strong that implies Assumption 2?
4. Do you think that the enhanced W2S training would improve performance for the original weak-to-strong setup in Burns et al 2023 (and is it even applicable)?

---

> ### Author Response · Authors · 2025-11-22
> **Response to Reviewer caiF [1/2]**
>
> We thank the reviewer for their constructive comments and feedback. We are glad that they find our setting interesting and recognize the novelty and significance of our work.
>
> ## Key Clarifications on the Weaknesses Section
> **[W1 (Q1). Precise characterizations of the teacher and student generalization errors in Theorems 1 and 2 (or 3) are sufficient to establish W2S, and how to interpret $p_{T \cap S}$ intuitively.]** First, we kindly clarify that in this work, W2S generalization is quantified by the absolute W2S gain $\Delta \mathcal R_{\eta_t}$ (defined in line 221 in the submission or eq.(4) in the revision, a common choice in W2S theory, e.g., [6,7]). In our theoretical setting introduced in Sec. 2.1, the W2S gain is fully and precisely characterized by the teacher and student generalization errors in Theorems 1 and 2 (or 3).
>
> We also highlight that $p_{T \cap S} = 1 + || \Xi ||^2_F$ is concretely defined by our problem setup, whose upper bound $p_{T \cap S} \le p_S \le p_T$ guarantees that the strong student outperforms the weak teacher in the variance induced by label noise. We added further clarification on this in Theorem 2 of the revision to avoid potential confusion. Intuitively, $p_{T \cap S}$ quantifies the effective group feature dimension learned by the strong student from the weak teacher controlled by the teacher-student similarity—less similar teacher-student pairs enjoy lower $p_{T \cap S}$.
>
> **[W2. We focus on W2S in post-training, where the weakness of the pre-trained teacher comes from its inferior sample efficiency.]** We agree with the reviewer that the finite training samples (or, equivalently, the positive aspect ratios $\gamma_z, \nu_z$ in the proportional asymptotic limit) are crucial for our W2S analysis. However, we respectfully point out that __this is a feature of our analysis motivated by practical post-training, instead of an artifact of our analysis__, evident from the following perspectives:
> * W2S is a phenomenon emerging in practice on large pre-trained models [10], motivated by which we formulate W2S in the post-trainign setting, where __downstream samples, especially the labeled ones, are often highly limited and noisy__. In the learning theory language, this means __the estimation error is noticeable for post-training problems__. In contrast, powerful pre-trained models, especially those related to superalignment, are generally considered to be sufficiently expressive for typical downstream tasks (i.e., given infinite samples from the population, both the weak and strong pre-trained models should be able to learn the downstream task). In the learning theory language, this means __the approximation error is negligible__. With noticeable estimation error and negligible approximation error, __the “qualities” of pre-trained features are measured by the sample efficiencies of their corresponding models__ (as highlighted in lines 164-167). The finite numbers of training samples, $n, N$, are incorporated to reflect such sample efficiencies and motivated by the practical setting of post-training.
> * Although finite training samples are essential for our analysis, as elaborated above, we kindly point out a misconception in the review: $N > n$ is not the main reason why W2S happens. First of all, although $n, N$ are both sample sizes, the amounts of information they carry are completely different—$n$ refers to the labeled sample size, whereas $N$ is the unlabeled sample size. Therefore, $N>n$ does not imply that the student receives more samples, since the types of samples received by the teacher and student are distinct. Second, comparing Theorems 1 and 3, we highlight that our theory does not require $N > n$ in order for W2S to happen. In Theorem 2 and Remark 2, we focus on the case $\nu_z \to d_z / N \ll 1$ simply for demonstration purposes since (i) unlabeled data are typically more available in practice, making large $N$ a practical assumption, and (ii) the third term (colored orange) in Theorem 2 (and 3), which is less relevant to our main goal, becomes negligible when $\nu_z \ll 1$.
>
> **[W3 (Q3). Our models are not assumed to be task-dependent. Instead, we separate weak vs. strong models in terms of their *pre-trained representations*, which should depend on both the models and the downstream task.]** We kindly point out a potential misunderstanding that we do not assume the teacher and student *models* depend on the downstream task, which is not reasonable, as the reviewer pointed out. Instead, how we formulate the separation between the weak and strong models in Assumption 2 is through __the pre-trained representations of the teacher and student models on the downstream task__, which ought to depend on both the models and the task. This theoretical setting naturally arises from the post-training nature of W2S highlighted in __[W2]__ above.

---

> > ### Comment · Reviewer_caiF · 2025-11-26
> >
> > **Regarding the bar for claiming to "establish W2S":** I understand that W2S is quantified by $\Delta \mathcal R_{\eta_t}$ and characterized by teacher and student generalization errors. But, I strongly think that this **does not constitute establishing W2S generalization, instead it characterizes the sufficient conditions for W2S.** In order to establish W2S generalization, one needs to show that a particular choice of student and teacher models actually satisfies the conditions needed to apply Theorems 1 and 2.
> >
> > **Regarding the mechanism of W2S:** I agree with the author's point about $n$ and $N$ but I'm still unsure that in the Ridgeless regression example presented in the paper, the W2S improvement does not come from the teacher being suboptimally trained. Can you show that no regularization or early stopping does not help (i.e. it's optimal to take a ridgeless predictor)?
> >
> > **Dependence on downstream task:** Looking at the original paper of Burns et al, the notion of weak and strong is defined by the number of parameters, which is task agnostic. Therefore, there does not have to be dependence on the task. I think a discussion of what this means for different tasks might be useful (like Example 1).

---

> ### Author Response · Authors · 2025-11-22
> **Response to Reviewer caiF [2/2]**
>
> **[W4 (Q4). The enhanced W2S method is motivated by our theory and tailored for W2S under spurious correlation, applicable to W2S on non-vision tasks with spurious correlation.]** We appreciate the reviewer’s constructive suggestion on examining broader applications of our method. Our method is built on two general principles: (i) using confidence-based selection to align $\eta_\ell$ and $\eta_u$ while reducing reliance on spurious features, and (ii) using GCE loss to mitigate pseudolabel noise.  Both ideas naturally extend to non-vision domains.
>
> To show this, we apply our method to MultiNLI, a widely used NLP benchmark with spurious correlations [3]. We use DistilBERT as the teacher and RoBERTa-large as the student, and our Enhanced-W2S yields clear gains over vanilla W2S: when $(\eta_\ell, \eta_u) = (\eta_o, 0.5)$ and when $(\eta_\ell, \eta_u) = (0.5, \eta_o)$, it improves worst group accuracy by $12.43$% and $5.12$%, respectively. For more complex NLP tasks, our ideas can be further adapted by replacing entropy with NLP-specific uncertainty quantification techniques [4] for confidence estimation, or replacing GCE with NLP-specific noise-correction methods [5].
>
> ## Responses to the Questions Section
> * Q1,3,4 are addressed in __[W1, W3, W4]__ above, respectively.
> * Q2: The proportional asymptotic limit is a standard setting in learning theory when random matrix theory tools are applied (see e.g., [8,9]). It brings the __precise characterizations of generalization errors__ (in contrast to upper and lower bounds) in our Theorems 1 and 2 (or 3), by circumventing pessimistic artifacts brought by using concentration inequalities in finite dimensions. Notice that these precise characterizations are crucial for informative and sharp analysis of W2S: since we want to compare the generalization of two models, upper/lower bounds are often not strong enough. It is also worth noting that these results in the proportional asymptotic limit can be extended to incorporate finite-sample cases through standard edge fluctuation analysis (see e.g., [8.9]). We did not include such extensions because they add little insight for our problem of interest.
>
> Once again, we thank the reviewer for their insightful questions and suggestions. If our responses above have addressed your concerns, we would truly appreciate a re-evaluation accordingly.
>
> ## References
> [1] Deng et al. *Robust learning with progressive data expansion against spurious correlation.* NeurIPS 2023.
>
> [2] Yang et al. *Identifying spurious biases early in training through the lens of simplicity bias.* AISTATS 2024.
>
> [3] Sagawa et al. *Distributionally robust neural networks.* ICLR 2020.
>
> [4] Lin et al. *Generating with confidence: Uncertainty quantification for black-box large language models.* TMLR 2024.
>
> [5] Wang et al. *Noise-robust fine-tuning of pretrained language models via external guidance.* EMNLP 2023.
>
> [6] Wu and Sahai. *Provable weak-to-strong generalization via benign overfitting.* ICLR 2025.
>
> [7] Ildiz et al. *High-dimensional analysis of knowledge distillation: Weak-to-strong generalization and scaling laws.* ICLR 2025.
>
> [8] Hastie et al. *Surprises in high-dimensional ridgeless least squares interpolation.* Annals of statistics 50.2 (2022): 949.
>
> [9] Cheng and Montanari. *Dimension free ridge regression.* The Annals of Statistics 52.6 (2024): 2879-2912.
>
> [10] Burns et al. *Weak-to-strong generalization: eliciting strong capabilities with weak supervision.* ICML 2024.

---

> > ### Comment · Reviewer_caiF · 2025-11-26
> >
> > The new example on a language domain is a good addition, and it answers one of my questions.
> >
> > Regarding asymptotic limits, I think it might be useful to add this discussion to the paper.

---

> ### Author Response · Authors · 2025-11-27
>
> We would like to thank the reviewer for their engaging discussion. We appreciate the opportunity to further clarify our work, and we are happy that our responses have addressed some of their questions and concerns. Below, we respond to the additional points raised, with a logically coherent reordering from high-level intuitions to technical details.
>
> **[Q1. Dependence on downstream task.]** We break down this question into two parts:
> 1. **What are pre-trained representations $\varphi_T, \varphi_S$?** A key empirical observation that we leverage to simplify the practical fine-tuning process to an analyzable theoretical model is that fine-tuning often lies in the kernel regime [1]. In the kernel regime, fine-tuning can be well approximated by a kernel regression problem over the neural tangent kernel (NTK), whose associated feature maps $\varphi_T, \varphi_S$ are the gradients of the model $f_T, f_S$ at their respective pre-trained initialization. Notice that $\varphi_T, \varphi_S$ are functions that take data (e.g., images of cows/camels) and output high-dimensional features for the particular inputs. The point we aim to convey in **[W3 (Q3)]** is that **the representation functions $\varphi_T, \varphi_S$ in our setting are independent of the downstream task**. We only make assumptions on **the high-dimensional features that they extract for the downstream samples, which should be task-dependent**. We revised our "weak vs. strong models" paragraph in Sec. 2.1 to further clarify this point.
> 2. **Why Burns et al. (2024) can separate weak vs. strong using "only" parameter counts?** While the original W2S paper, Burns et al. (2024), uses parameter counts to separate weak vs. strong models, this is an intuitive choice of measure conditioned on the fact that their weak teacher, GPT-2, and strong student, GPT-4, are pre-trained models from the same organization, where the strong student is pre-trained with better data, more compute, and likely better strategies. These critical differences co-exist with the intuitive difference in parameter counts, but are not limited to it. In our theoretical framework, we synopsize these differences into the quality of the pre-trained representation functions, $\varphi_T, \varphi_S$, which are independent of the downstream task as explained above.
>
> As mentioned in our response to Reviewer rtLS **[W3]**, we acknowledge that the assumption about the kernel dynamics of fine-tuning can be considered as a necessary theoretical simplification that often, but not always, holds in practice [4]. The extension of our theory beyond the kernel regime (e.g., taking the gradient descent dynamics of non-linear feature learning into consideration) is an exciting future direction, but not relevant to the main question of interest (i.e., W2S under spurious correlation) in this work.
>
> **[Q2. Regarding the bar for claiming to "establish W2S".]** We thank the reviewer for clarifying their definition of "establishing W2S". As clarified in our response to **[Q1]**, we make assumptions directly on the high-dimensional features extracted by the pre-trained representations, $\varphi_T(x) \in \mathbb R^{D_T}, \varphi_S(x) \in \mathbb R^{D_S}$, instead of on the representation functions $\varphi_T, \varphi_S$ and the downstream data $x \sim \mathcal D_x$ separately. This is part of our problem setup resulting from casting W2S as a fine-tuning problem in the kernel regime, which, as acknowledged above, is a necessary theoretical simplification to make the problem analyzable. More importantly, our theory based on this simplification is predictive for real-world tasks; while making separate assumptions on $\varphi_T, \varphi_S$ and $x$, even in the kernel regime, would complicate the analysis significantly, compromising the interpretability without providing additional insights into the key question of W2S under spurious correlation. In light of this and the reviewer's comments, we checked our claims throughout the paper to make sure that there were no overstatements regarding this point.
>
> **[Q3. Regarding the mechanism of W2S.]** As highlighted in Remark 1, motivated by the ubiquitous empirical observations on the low intrinsic dimension of fine-tuning, for features constrained in low-dimensional subspaces (analogous to the latent space features defined in [2], see e.g., [2, Sections 1 and 6.2]), ridgeless regression is nearly optimal at large overparametrization, i.e., $D_T \gg n$ and $D_S \gg N$. Notice that ridgeless regression should not be interpreted as "no-regularization" but rather as a form of fixed, natural regularization (min-norm) induced by the low intrinsic dimension of the features.
>
> [1] Malladi et al. A kernel-based view of language model fine-tuning. ICML 2023.
>
> [2] Hastie et al. Surprises in high-dimensional ridgeless least squares interpolation. Annals of statistics 50.2 (2022): 949.

---

> > ### Comment · Reviewer_caiF · 2025-11-27
> >
> > I thank the authors for their further clarifications.
> >
> > **Regarding separating weak and strong, and more generally, this approach to studying W2S phenomena**: I understand now the interaction between the pre-trained representation and the task. While abstracting away the "weak" and "strong" to the pre-trained representations makes the approach more general, I think the large tradeoff is that it's much harder to understand. This is the main reason why I think the parameter count or similar "more intuitive" definitions are better.
> >
> > **Regarding "establishing W2S"**: I don't object to theoretical simplification to make the problem more analyzable. I believe that the analysis of sufficient conditions for W2S unnecessarily stops at Step 1 of multiple steps that are needed to fully understand W2S. I would still like to see more specific models/settings analyzed as corollaries of the main theorems in the paper.

---

> > > ### Author Response · Authors · 2025-12-03
> > >
> > > We appreciate the follow-up questions and comments. Unfortunately, we are unable to continue the conversation due to the change in the rebuttal policy. We hope our final comment below addresses the remaining concerns.
> > >
> > > Both points raised by the reviewer were regarding the possible extensions/instantiations of our current theoretical setting, i.e., replacing our current assumptions on the instantiations of the teacher and student models on samples, $\varphi_T(x), \varphi_S(x)$, with separated assumptions on the models (feature functions), $\varphi_T, \varphi_S$, and the input, $x$.
> > >
> > > We acknowledge that, as pointed out in our previous response, **it is an exciting future direction to take the training dynamics of the teacher and student models, conditioned on their pre-trained initialization, into consideration, i.e., investigating fine-tuning beyond the kernel regime where $\varphi_T, \varphi_S$ are no longer fixed feature functions**. However, as also pointed out in our previous response, for any fixed feature functions, mathematically contriving $\varphi_T, \varphi_S$ and distribution of $x$ to match our Definition 2 would not qualitatively alter the message in our theory (or the kernel nature of our analysis) on W2S under spurious correlation, but adding unnecessary complications to the results.
> > >
> > > We did not discuss the future direction on **investigating fine-tuning beyond the kernel regime** in our initial submission since it is relatively orthogonal to the focus of this paper (i.e., W2S under spurious correlation). In light of our discussion, we will include a brief comment on this point in our revision.

---

### Official Review · Reviewer_c6kM · 2025-11-04

**Soundness:** 3
**Presentation:** 2
**Contribution:** 2
**Rating:** 4
**Confidence:** 4

**Summary:**

This paper extends the results of Dong et al. (2025) to the setting where there are spurious correlations induced by group imbalance in the training data distribution. The paper derives excess risk bounds for (1) a teacher model trained on a small number of labeled examples (Theorem 1) with a representation that doesn't well-separate the spurious and non-spurious features, and (2) a student model trained on an unlabeled data distribution pseudolabeled by the teacher. The bounds indicate that the weak-to-strong gains are largest when the percentage of the "minority group" data is very close in the teacher and student's training data distributions. Based on this observation, it proposes to do two rounds of weakly-supervised training, where in the second round the student retrains on the subset of examples where it was confident in its label.

**Strengths:**

- Testable theory for the magnitude of weak-to-strong generalization in the presence of group imbalance / spurious correlations
- Theoretical motivation for retraining on subset of data where the student is most confident
- The above approach outperforms "Vanilla" W2S generalization on several datasets and for several teacher/student pairs.

**Weaknesses:**

- The analysis seems to rely on the fact that the teacher model is an ERM on the labeled data. In this case, why doesn’t the student also use the small labeled set instead of (or in addition to) using the pseudolabeled data? How would the student's performance bound compare in that case? I’m sure the analysis can be fixed to incur some additive error term that measures the teacher’s excess risk over the ERM, but the same point applies

- Not enough discussion of (or motivation for---why are we interested in W2S in the context of spurious correlations?) the incremental novelty over Dong et al. The paper seems like a targeted extension of Dong et al. to the spurious-correlation regime, but that regime is not well-motivated.

- The confidence loss / retraining idea is similar to the confidence loss in Burns et al and very similar to the iterative confident retraining in COSINE [1], and the idea of confident data selection in weak supervision is old (see e.g., [2] and many others). The experiments are missing "3rd-party" baselines like the Burns confidence loss, and the paper is missing contextualization of this idea in the long literature on weak supervision, self-training, and co-training.

Minor Presentation comments:
- Assumption 2 takes a long time to parse. A figure here might be helpful?
- Section 2.3 can probably be cut or moved to the appendix…we might not need experimental validation of the theory in the exact setting of the Theorem. It would be more interesting to argue why some of the assumptions of the theorem are practically relevant, or to bring some of the content from Appendix E into the main text

[1] Yu et al. "Fine-Tuning Pre-trained Language Model with Weak Supervision: A Contrastive-Regularized Self-Training Approach"

[2] Zhang et al. "CoTrade: Confident co-training with data editing"

**Questions:**

- If we're doing multiple rounds of training, why not co-train the teacher?
- Why not use the labeled data to train the student?
- "For each experiment on a given dataset, we include all teacher—student pairs whose relative strength (measured by accuracy) remains stable when we vary parameters including $\eta_l$, $\eta_u$, $N$, or $n$” ← What does this mean?

---

> ### Author Response · Authors · 2025-11-22
> **Response to Reviewer c6kM [1/2]**
>
> We thank the reviewer for their time and thoughtful feedback. We are glad that they found our theory solid and our algorithm well-motivated. Meanwhile, we would like to take this opportunity to clarify some potential misunderstandings of the W2S setting and our contributions.
>
> ## Key Clarifications on Weaknesses
>
> __[W1 (Q2). Labeled data for teacher SFT are not available to the student in common W2S settings, and why this setting is reasonable.]__ First, we would like to reiterate the setting of W2S generalization introduced in the original W2S paper [1] (Sec. 3), where the weak teacher is created through supervised fine-tuning (SFT) on ground truth labels (that are only available to the weak teacher but not the strong student). Then, the strong student is trained __only with weak supervision from the fine-tuned teacher__ on a held-out set of samples. Our theoretical setting precisely reproduces this pipeline.
>
> We agree with the reviewer that if ground truth labels for teacher SFT are available to the strong student, leveraging them will enhance the student's performance. However, the purpose of studying W2S generalization [1], especially in theoretical settings like [2,3,4] and ours, is to __develop principled understandings of whether and why weak supervision can bring stronger performance, rather than improving the performance of the strong student__. For this purpose, mixing ground truth labels with weak labels from the teacher is a bad experimental design that fails to isolate the effect of weak supervision. Moreover, __in practical W2S settings, SFT data for the teacher are likely unavailable for the student model__, especially on specialized downstream tasks, e.g., medical diagnosis, where labeled training data are rarely public for privacy considerations.
>
> __[W2.1. Downstream data often contain systematic biases, understanding the effect of which on W2S is crucial for principled design of practical W2S algorithms.]__ We respectfully disagree with the reviewer regarding the lack of motivation for studying W2S under spurious correlation. We kindly point the reviewer to __the Introduction (Sec. 1, lines 43–51) and Conclusion and Discussion (Sec. 4, lines 479–485)__ of our submission, __both of which explicitly highlight the importance of understanding the effect of spurious correlation on W2S, with concrete examples__. In particular, we pointed out in the introduction that data for practical downstream tasks (e.g., medical diagnosis, autonomous driving, etc.) are often limited, noisy, while carrying systematic biases like group imbalance. Therefore, a rigorous understanding of W2S under systematic biases is a critical missing piece in the current W2S theory, and our work made the first step.
>
> __[W2.2. Analyzing W2S under spurious correlation requires fresh theoretical formulation and different mathematical tools, bringing distinct theoretical insights from [4], all beyond incremental contributions.]__ In addition to reiterating our motivation, from the technical perspective, we would like to highlight the substantial difference between the theory in our work and that in [4], clarifying the misconception in the review regarding our contributions. We emphasize that the main novelties of our theoretical results are:
> 1. __a new theoretical formulation of W2S as a post-training/fine-tuning problem__ (i.e., weak vs. strong is encoded in pre-trained representations of different models via different disentanglement of the spurious, group features) __on a downstream regression task with spurious correlation__,
> 2. __precise characterizations of the teacher and student excess risks in the proportional asymptotic limit that leverage tools from random matrix theory__, different from the non-asymptotic analysis in [4], and
> 3. __fresh, testable insights from the rigorous analysis regarding the effect of heterogeneous group imbalance on W2S__, i.e., a large difference in group imbalanceness between the teacher and student training sets, $(\eta_u - \eta_\ell)^2$, hurts W2S.
>
> Each of these contributions is beyond incremental technically and well-motivated by the main question raised in our introduction—“does W2S happen under spurious correlation, which often occurs in practical downstream data?” We agree that, as acknowledged in the problem setup (Sec. 2.1), our theoretical framework is partially inspired by [4] and leverages the same simplifications for fine-tuning dynamics (i.e., kernel regime + low intrinsic dimension). However, in our setting, these are necessary, preliminary assumptions to cast W2S as a post-training problem, instead of the main message of Sec. 2, evident from the organization of Sec. 2.1.

---

> ### Author Response · Authors · 2025-11-22
> **Response to Reviewer c6kM [2/2]**
>
> __[W3.1 (Q1). In W2S, the supervisedly fine-tuned weak teacher is assumed to be fixed and not retrainable, and why this assumption is reasonable.]__ First, we appreciate the related literature pointed out by the reviewer and have integrated it into a new paragraph in the additional related works (Appendix B) on multi-round retraining and confidence-based selection. While we agree that W2S shares a similar high-level flavor as self-training and co-training, we kindly reiterate that their settings and motivations are fundamentally different from those of W2S, as elaborated in the response to __[W1 (Q2)]__ above. In addition to the obvious uniqueness of W2S in the notion of weak teacher vs. strong student, an important difference of the W2S setting is that __the supervisedly fine-tuned weak teacher is generally assumed to be fixed and not retrainable, accessible only through inference [1]__. This assumption is __originally motivated by the explanatory (in contrast to performance-oriented) natural of W2S__ highlighted in __[W1 (Q2)]__ but __also reflects practical user cases__, e.g., one use a closed-source model fine-tuned by some company on potentially biased privated data as a weak teacher to generate weak supervision for fine-tuning a strong student. We included additional clarifications after eq.(2) in the revision to avoid potential misunderstanding.
>
> __[W3.2. Our algorithm is a principled design motivated by the fresh theoretical insights, leveraging widely adopted practical techniques to provide a tailored algorithmic remedy for W2S under spurious correlation.]__ We agree with the reviewer that the concepts of confidence-based selection and retraining are intuitive and have been broadly adopted in many settings different from ours. However, our main algorithmic contribution is a principled combination of these practical ideas, motivated by the fresh insight from our theory and tailored for the challenge of W2S under spurious correlation when $\eta_u \neq \eta_\ell$.
>
> In the revision, we additionally evaluate the auxiliary loss from [1] as a third-party baseline. On Waterbirds, with carefully tuned hyperparameters, Burns’ loss yields $1.10$% and $1.82$% relative improvements in average accuracy across model pairs under the $\eta_u = \eta_o$ and $\eta_\ell = \eta_o$ settings, whereas our Enhanced-W2S method achieves $6.32$% and $7.72$% under the same settings. Moreover, Table 1 shows that Enhanced-W2S improves all model pairs on Waterbirds, while Burns’ loss improves only $7/9$ and $5/9$ pairs in the two settings. These results show that our method is more effective for W2S under spurious correlations.
>
> __[W4. Improved presentation of our theoretical setup through the new Figure 1.]__ Thanks to the constructive suggestions on the presentation, we included a new figure, Figure 1, in the revision to visualize our theoretical setup described in Assumptions 1 and 2.
>
>
> ## Responses to the Questions Section
> * Q1 is addressed in __[W3.1]__, and Q2 is addressed in __[W1]__ above.
> * Q3: For example, in the Enhanced-W2S experiments summarized in Table 1, we vary the combinations of  $(\eta_\ell, \eta_u)$ for each dataset, and for every $(\eta_\ell, \eta_u)$ pair we further change $(N, n)$ to obtain the performance of Enhanced-W2S. If the relative strength of a model pair (measured by accuracy) changes under different $(\eta_\ell, \eta_u)$ or different $(N, n)$ configurations, we consider their strength relationship to be unstable and unreliable for comparison in this experiment, and exclude such pairs from the table.
>
> We are happy to answer any further questions you may have. If our responses above have addressed your concerns, we would truly appreciate a re-evaluation accordingly.
>
>
>  ## References
>
> [1] Burns et al. *Weak-to-strong generalization: eliciting strong capabilities with weak supervision.* ICML 2024.
>
> [2] Lang et al. *Theoretical analysis of weak-to-strong generalization.* NeurIPS 2024.
>
> [3] Ildiz et al. *High-dimensional analysis of knowledge distillation: Weak-to-strong generalization and scaling laws.* ICLR 2025.
>
> [4] Dong et al. *Discrepancies are virtue: Weak-to-strong generalization through the lens of intrinsic dimension.* ICML 2025.

---

### Meta-Review · Area_Chair_9LGm · 2026-01-12

**Summary:**

This paper studies weak-to-strong (W2S) generalization under spurious correlations induced by group imbalance. The authors provide a unified theoretical analysis that characterizes when W2S succeeds or fails as a function of the mismatch between group proportions in the teacher’s labeled data and the student’s pseudo-labeled data. The theory yields precise, testable predictions in a proportional asymptotic regime and explains why W2S gain diminishes as this mismatch increases. Motivated by these insights, the paper further proposes Enhanced-W2S, a simple, group-label-free retraining strategy based on high-confidence data selection, and demonstrates consistent empirical improvements across multiple spurious-correlation benchmarks and teacher–student pairs.

Overall, the reviewers find the paper technically sound, well motivated, and empirically convincing. While some concerns remain regarding the scope of the theoretical assumptions and interpretability, the contribution is clear and timely, and the work makes a meaningful addition to the growing literature on W2S generalization.

**Reviewer Concerns:**

This paper receives mixed reviews. While three reviewers provide generally positive assessments, one reviewer (Reviewer c6kM) raises concerns regarding the W2S setting and its novelty relative to existing work. While Reviewer c6kM was unable to participate in the discussion, I believe most concerns were largely addressed in the rebuttal, as they are primarily regarding clarifications rather than critical weaknesses.

**Reviewer Scores:**

Reviewer ZDqv: Initially positive and, after rebuttal and revisions, explicitly raised the score to 8, indicating that the main concerns were resolved.

Reviewer rtLS: Similarly, began at 6 and raised score to 8 after rebuttal, indicating that the main concerns were resolved and acknowledging the contributions of this work.

Reviewer caiF: was also positive about this work in general.

Reviewer c6kM: Gave the only initially negative score, primarily due to differing views on the W2S setting and novelty. Although he was unable to participate in the discussion, several issues were likely clarified and resolved.

---

### Decision · Program_Chairs · 2026-01-26

Accept (Poster)